JCB Journal of Cell Biology

## TOOLS

# PILS-Nir1 is a sensitive phosphatidic acid biosensor that reveals mechanisms of lipid production

Claire C. Weckerly[1]*, Taylor A. Rahn[2]*, Max Ehrlich[1], Rachel C. Wills[1], Joshua G. Pemberton[3], Michael V. Airola[2], and Gerald R.V. Hammond[1]

Phosphatidic acid (PA) regulates lipid homeostasis and vesicular trafficking, yet high-affinity tools to study PA in live cells are lacking. We identified the lipin-like sequence of Nir1 (PILS-Nir1) as a candidate PA biosensor based on structural analysis of Nir1's LNS2 domain. Using liposome-binding assays and pharmacological and genetic manipulations in HEK293A cells expressing fluorescent PILS-Nir1, we found that while PILS-Nir1 binds PA and PIP$_2$ *in vitro*, only PA is necessary and sufficient for membrane localization in cells. PILS-Nir1 displayed greater sensitivity to organelle-generated PA than Spo20-based probes, enabling visualization of modest PA production by PLD downstream of muscarinic receptors—previously undetectable with existing biosensors. Thus, PILS-Nir1 provides a versatile, sensitive tool for real-time PA dynamics in live cells.

## Introduction

Phosphatidic acid (PA) is a truly versatile lipid, with parallel activities as a vital metabolic intermediate, a second messenger, and a determinant of unique membrane properties (Zhou et al., 2023). PA serves as a precursor for lipid species such as diacylglycerol (DAG), lysophosphatidic acid, and CDP-DAG, each of which is used in its own signaling and metabolic pathways (Thakur et al., 2019). PA also regulates the localization and function of various enzymes such as phosphatidylinositol 4-phosphate 5-kinase (PIP5K) (Cockcroft, 2009), mTOR (Frias et al., 2023), ERK (Zhang et al., 2014), and Hippo (Han et al., 2018). Finally, PA controls membrane architecture by inducing negative membrane curvature (Zhukovsky et al., 2019), thereby playing a role in membrane trafficking (Zeniou-Meyer et al., 2007; Tanguy et al., 2021). Due to these multiple roles, PA and its associated regulatory and effector enzymes have been identified as therapeutic targets in a variety of diseases such as cancers, neurodegenerative diseases, and hypertension (Brown et al., 2017; Bruntz et al., 2014; Castagna et al., 1982; Cooke and Kazanietz, 2022; Dhalla et al., 1997; Fazio et al., 2020; Kato et al., 1992; Sakane et al., 2008; Tappia and Singal, 2009; Thakur et al., 2019). Despite the role of PA in a multitude of cellular functions and diseases, the regulation of PA is not fully understood. This is in part due to a lack of high-affinity tools available to study PA in live cells.

PA is produced at the plasma membrane (PM) through two interrelated pathways: activation of phospholipase C (PLC) and diacylglycerol kinases (DGKs) (Kadamur and Ross, 2013; Shulga et al., 2011) and activation of phospholipase D (PLD) by protein kinase C (PKC) (Selvy et al., 2011). These pathways are thought to be stimulated consecutively as PLC activity increases DAG and intracellular Ca$^{2+}$ levels, which then in turn activate PKC, and PKC subsequently activates PLD. While recent click-chemistry fluorescent lipid reporters have shown PLD activation by PLC signaling, the role of PLD in producing endogenous PA downstream of the PLC pathway is still unclear (Liang et al., 2019). This highlights the need for better tools to study PA production in real time within living cells.

The most robust way to study lipids in live cells is through genetically encoded lipid biosensors (Maekawa and Fairn, 2014; Wills et al., 2018; Hammond et al., 2022). These sensors are fluorescently tagged effector proteins or domains that bind specifically to a lipid of interest and label it in intracellular membranes. Biosensors need to be carefully characterized to avoid misinterpretation of lipid dynamics. Our lab has previously defined three criteria that we consider to be crucial for a lipid biosensor to meet: is the biosensor specific for the lipid of interest? Is the membrane localization of the biosensor dependent on that lipid? And, is the lipid of interest sufficient to localize the biosensor to membranes? (Wills et al., 2018).

A variety of effector PA-binding domains (PABDs) have been characterized with the goal of utilizing them as PA biosensors, including *Saccharomyces cerevisiae* Opi1 (Hofbauer et al., 2018),

[1]Department of Cell Biology, University of Pittsburgh School of Medicine, Pittsburgh, PA, USA;   [2]Department of Biochemistry and Cell Biology, Stony Brook University, Stony Brook, NY, USA;   [3]Section on Molecular Signal Transduction, Program for Developmental Neuroscience, Eunice Kennedy Shriver NICHD, National Institutes of Health, Bethesda, MD, USA.

*C.C. Weckerly and T.A. Rahn contributed equally to this paper.   Correspondence to Gerald R.V. Hammond: ghammond@pitt.edu.



PABD-PDE4A1 (Baillie et al., 2002), PABD-Raf1 (Ghosh et al., 2003), and the N terminus of alpha-synuclein (Yamada et al., 2020). However, these all display somewhat limited PA-membrane binding, respond to additional stimuli such as membrane curvature or $Ca^{2+}$ flux, or have not been characterized in cells with endogenous PA levels (Kassas et al., 2017, 2012).

The most widely used PA biosensors are those that utilize the amphipathic helix of a sporulation-specific soluble N-ethylmaleimide–sensitive factor attachment protein receptor (SNARE) from S. cerevisiae: Spo20p (Fig. 1 I) (Zeniou-Meyer et al., 2007; Bohdanowicz et al., 2013; Zhang et al., 2014). This amphipathic helix, made of residues 51–91 from Spo20 and subsequently referred to as PABD-Spo20, binds PA in vitro and in yeast (Nakanishi et al., 2004). Interestingly, in human cell lines, PABD-Spo20 is highly localized in the nucleus, but it does bind the PM when PA levels are increased (Du and Frohman, 2009; Zeniou-Meyer et al., 2007).

A caveat to PABD-Spo20 is that this sensor also binds phosphatidylinositol 4,5-bisphosphate ($PIP_2$) and phosphatidylinositol 4-phosphate (PI4P) in biochemical assays. It was even suggested to have nonspecific interactions with any negatively charged lipids present in the membrane (Nakanishi et al., 2004; Horchani et al., 2014). This highlights the three major problems in the design of PA biosensors: (1) we have yet to discover a protein sequence or domain structure that is specific for PA binding, (2) amphipathic helices like the PABD tend to indiscriminately interact with membranes, and (3) PA is a negatively charged lipid with a simple structure. Therefore, it can be unclear whether a sensor is specific for PA or whether it has a general affinity for negatively charged membranes.

When looking for a more PA-specific domain to use as a biosensor, we investigated the Nir family of phosphatidylinositol transfer proteins (PITPs). This family of proteins, made up of Nir1, Nir2, and Nir3, localizes to ER-PM membrane contact sites (MCS) to exchange PA and phosphatidylinositol (PI) between the compartments (Cockcroft and Raghu, 2016; Kim et al., 2015). While Nir1 lacks a functional PITP domain, it was initially classified as part of the PITP family based on the homology of its other domains with Nir2 and Nir3. Furthermore, Nir1 has a role in lipid transfer by facilitating Nir2 recruitment to the MCS (Quintanilla et al., 2022).

The Nir proteins all contain a C-terminal lipin/Nde1/Smp2 (LNS2) domain (Fig. 1 I). The AlphaFold predicted structure of this domain shows similarity to the lipin/Pah family of phosphatidic acid phosphatases (PAPs). Lipin/Pah PAPs interact with the membrane through an N-terminal amphipathic helix and catalyze the dephosphorylation of PA through a DxDxT-containing $Mg^{2+}$-binding active site (Khayyo et al., 2020). These features are conserved in the Nir LNS2 domains, except for the catalytic Asp in the DxDxT motif and another $Mg^{2+}$-coordinating residue (Fig. 2 D).

Therefore, the Nirs are suggested to sense PA levels with their LNS2 domain but not dephosphorylate the lipid (Kim et al., 2013). As the LNS2 tertiary structure is unique compared with the helical nature of PABD-Spo20, we investigated Nir1-LNS2 as a putative novel PABD. Due to the differences between the Nir1-LNS2 and a "true" LNS2, we renamed the Nir1 LNS2 domain and the subsequent biosensor made from it the PA-interacting lipin-like sequence of Nir1 or PILS-Nir1.

The literature suggests that the Nir family LNS2 domains are specific for PA in vitro, and bind the PM in a PA-dependent way, both as an isolated domain and in the context of the full-length Nir proteins (Kim et al., 2013; Chang and Liou, 2015; Quintanilla et al., 2022). However, other studies have suggested that the Nir proteins respond to changes in DAG and PA (Kim et al., 2015). Therefore, membrane binding of this domain must be further characterized to determine whether it meets the criteria of a valid PA biosensor.

In this study, we set out to corroborate PILS-Nir1 as a novel and high-affinity PA biosensor. We used pharmacological stimulation of HEK293A cells, liposome-binding assays with the purified PILS-Nir1, and chemically inducible dimerization systems to show that PILS-Nir1 convincingly reports changes in PA levels at the PM and is more sensitive than Spo20-based biosensors both in vitro and in vivo. Furthermore, PILS-Nir1 exhibits properties of a high-quality PA biosensor: it binds PA and $PIP_2$ in liposomes, but its membrane interactions in cells are solely dependent on PA, and PA is sufficient to recruit PILS-Nir1 to cellular membranes. We then use PILS-Nir1 to demonstrate differences in PA production in various organelles and cell models, as well as uncover that endogenous PLD activity contributes to PA levels after PLC activation. Thus, this work defines PILS-Nir1 as a novel tool for the study of PA and reveals aspects of PA regulation that have not been detected by previous biosensors.

## Results

### PILS-Nir1 is highly sensitive to PA

Before setting out to design a novel PA biosensor, we first characterized PABD-Spo20 and other Spo20-based biosensors for use as positive controls. To stop PABD-Spo20 from accumulating in the nucleus, as the small helix is thought to act as a nuclear localization sequence, Zhang et al. (2014) added a nuclear export sequence (NES) to PABD-Spo20, naming this sensor the PA biosensor with superior sensitivity (PASS). PASS shows specificity for PA in vitro and dependency on PA for membrane binding in cells (Zhang et al., 2014). An additional PABD was then added to PASS to increase the avidity by enabling the sensor to bind to two PA molecules. We refer to this sensor as NES-PABDx2-Spo20 (Bohdanowicz et al., 2013).

To test these sensors, we stimulated HEK293A cells expressing various Spo20 biosensors with 100 nM phorbol 12-myristate 13-acetate (PMA). PMA is a phorbol ester that activates PKC, which then activates PLD to hydrolyze phosphatidylcholine (PC) and produce PA at the PM (Castagna et al., 1982; Liang et al., 2019). After imaging the cells, we measured the PM localization of the biosensors by using fluorescent PM markers to create a PM mask: either the $PIP_2$ biosensor iRFP-PH-PLCδ1, a PM-localized fluorophore utilizing the CAAX motif from HRAS (TagBFP-HRAS-CAAX or mCherry-HRAS-CAAX), or CellMask Deep Red (Várnai and Balla, 1998; Idevall-Hagren et al., 2012). We then measured the fluorescence intensity of the biosensors

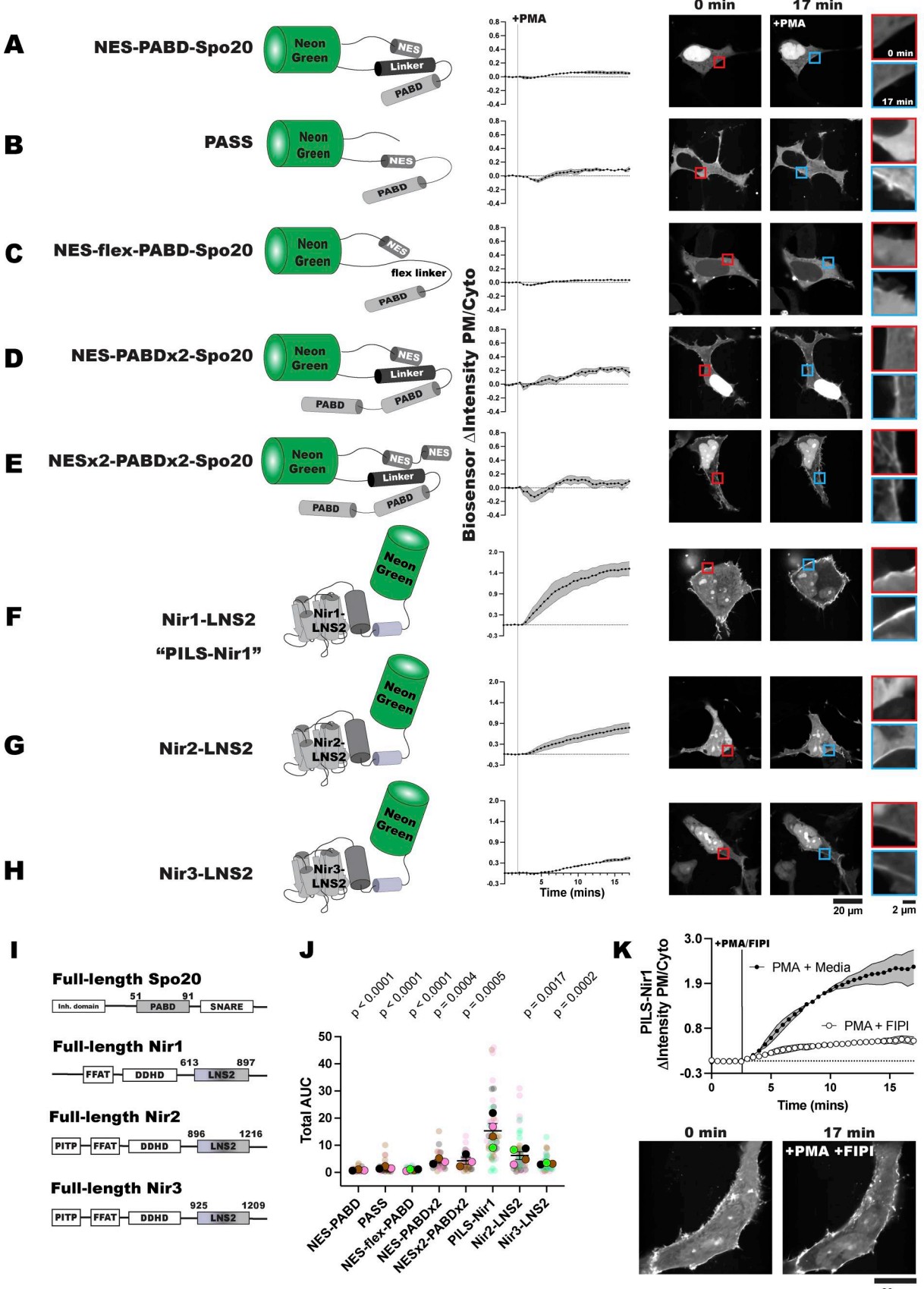

Figure 1. **PILS-Nir1 is highly sensitive to PA. (A–E)** PA biosensors were made from the PABD of Spo20 with added nuclear export sequences (NES) and various linker sequences. Alpha helices are shown by cylinders, while unstructured regions are shown as lines. **(F–H)** Novel biosensors were designed using the

LNS2 domain of the Nir family of proteins. Sensors translocated to the PM after PKC-mediated PLD activation with 100 nM PMA in transfected HEK293A cells. The red inset shows the PM intensity of the sensor before PMA stimulation, and the blue inset shows the PM intensity after PMA stimulation. Data shown are the grand mean of three to four experiments ± SEM. **(I)** Schematics of full-length Spo20 and Nir proteins. **(J)** AUC for the biosensor responses in A–H. The small circles indicate the AUC of individual cells ($n$ = 26–52). The large circles show the average AUC for each experimental replicate ($n$ = 3–4). Cells in each replicate are color-coded accordingly. Statistics were calculated with a post hoc one-way ANOVA using the average AUC of each experimental replicate ($n$ = 3–4), and the P values show the comparison of the respective biosensors to PILS-Nir1 (F = 12.74, P < 0.0001, $R^2$ = 0.8244). **(K)** Stimulating HEK293A cells with 100 nM PMA and 750 nM of the PLD1/2 inhibitor FIPI diminished the PM translocation of PILS-Nir1 seen with PMA and cell media. Data shown are the grand mean of three experiments ± SEM. A total of 27–41 cells were analyzed.

within the membrane mask and calculated the PM/cytoplasmic fluorescence intensity ratio of the biosensors (PM/Cyto), which increases as the biosensor translocates to bind PA at the PM (Wills et al., 2021).

First, we used a NeonGreen (NG)-tagged *S. cerevisiae* PABD-Spo20 with an added NES (Kim et al., 2015), mimicking the design of PASS (Zhang et al., 2014). This biosensor, named NES-PABD-Spo20, only showed minimal PM localization after 15 min

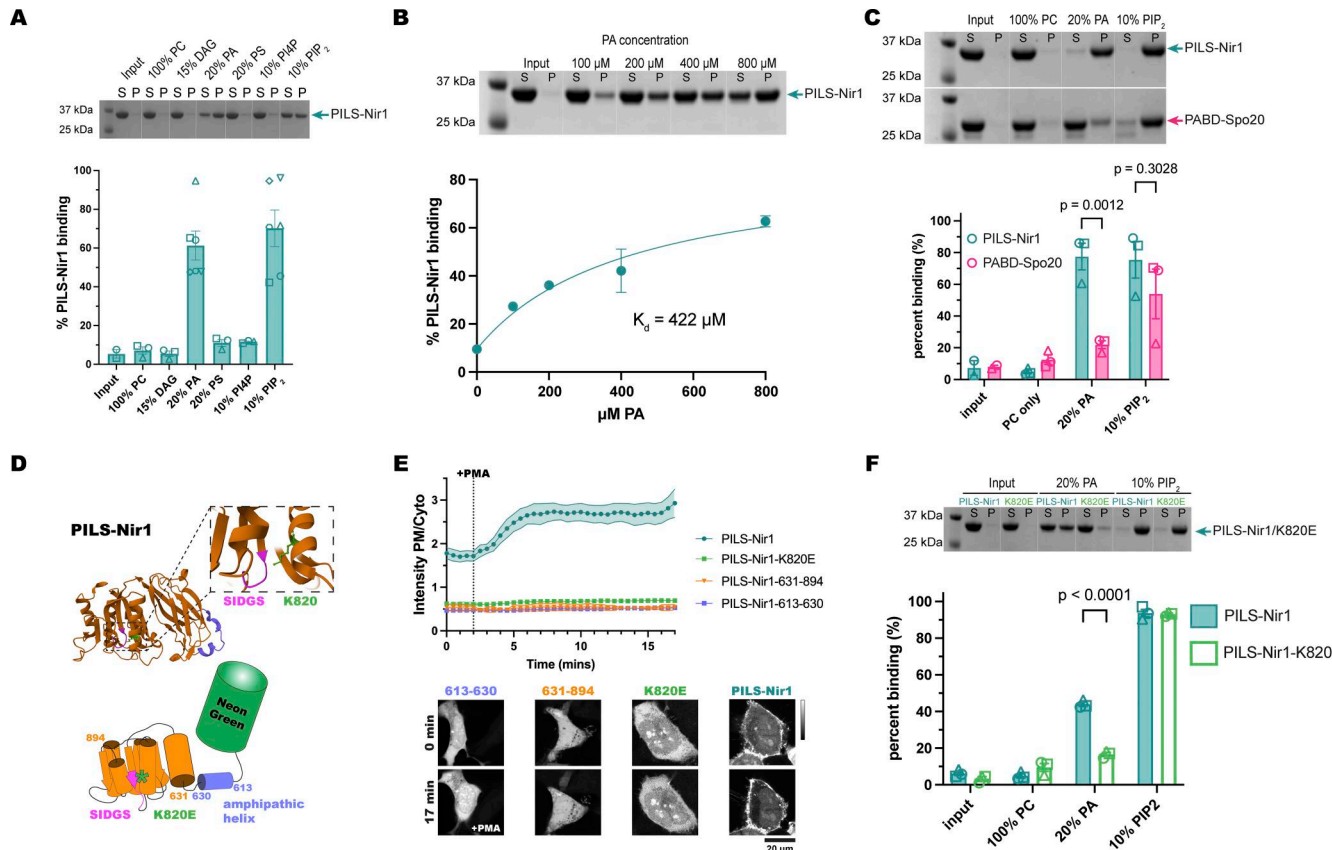

Figure 2. **PILS-Nir1 shows specificity for PA and PIP$_2$ *in vitro*, based on a novel PABD structure. (A)** Representative SDS-PAGE gel and quantification showing PILS-Nir1 binding of various PM lipids in POPC liposomes. Lipids indicated were mixed with POPC to produce a 2 mM solution, and then, 50 µl of the resulting liposome mixture was incubated with 50 µl of protein at ~1 mg/ml to produce 1 mM lipids in the assay. Supernatant (S) and pellet (P) lanes were quantified using ImageJ to determine percent protein bound. The protein-only control pellet was used as a baseline (input). **(B)** Representative SDS-PAGE gel and quantification showing PILS-Nir1–binding affinity for PA. Liposomes were made of 80 mol% PC and 20 mol% PA, and the total concentration of lipids was increased to achieve the indicated [PA]. Supernatant (S) and pellet (P) lanes were quantified using ImageJ to determine percent protein bound. The protein-only control pellet was used as a baseline (input). A nonlinear fit was produced using an equation for a one-site specific binding with background (see Materials and methods). Background was constrained to the minimum value of 9.504. $B_{max}$ was constrained to <100. The $K_d$ is as shown, with a 95% C.I. = 110.1–789.9 µM. Degrees of freedom = 3 and $R^2$ = 0.9715. **(C)** Representative SDS-PAGE gel and quantification of PILS-Nir1 and PABD-Spo20 binding of various PM lipids in POPC liposomes. Statistics were determined using a two-way ANOVA with Šídák's multiple comparisons test. DF = 1, MS = 1,567, F(1,14) = 7.750, P = 0.0146. Solutions were prepared, and binding was quantified as in A. **(D)** Representations of the AlphaFold predicted domain architecture of PILS-Nir1. It includes an amphipathic alpha helix spanning residues 613–630 (purple) and a large, structured domain at residues 631–894 (orange), which contains the SIDGS motif (pink) that is conserved with the lipin/Pah active site. The nearby K820 residue (green) is predicted to help stabilize PA within the domain. **(E)** Isolating either region of the PILS-Nir1 domain or introducing a K820E mutation destroyed the ability of PILS-Nir1 to respond to 100 nM PMA in HEK293A cells. The graph shows the grand means ± SEM of 3–4 experiments (35–42 total cells). **(F)** Representative SDS-PAGE gel and quantification for WT PILS-Nir1 and PILS-Nir1-K820E binding of various PM lipids in POPC liposomes. Statistics were determined using a two-way ANOVA with Šídák's multiple comparisons test. DF = 1, MS = 256, F(1,16) = 45.36, P < 0.0001. Solutions were prepared, and binding was quantified as in A. Source data are available for this figure: SourceData F2.

of PMA stimulation (Fig. 1 A). Unexpectedly, this biosensor had strong localization in the nucleus of HEK293A cells, despite the added NES. We then tested PASS itself against the NES-PABD-Spo20, and while PASS only showed slightly more PM binding after PMA stimulation, the PASS sensor was strongly excluded from the nucleus (Fig. 1 B). We determined that the key difference between NES-PABD-Spo20 and PASS was the location of the NES: in NES-PABD-Spo20, it is N-terminal to the NG, whereas in PASS, it is inserted between the C terminus of NG and the PABD. When looking at the structure of NES-PABD-Spo20 using ColabFold (Mirdita et al., 2022), we observed that the linker of the NES-PABD-Spo20 formed an alpha helix at the C terminus of NG (represented by cylinders in Fig. 1). We hypothesized this helix may have blocked nuclear exporters from accessing the NES and that the structure of PASS avoided this hindrance by having both the PABD and the NES on the C-terminal end of NG (Fig. 1, A and B).

To test this idea, we redesigned the sensor to replace the helical linker with a flexible Ser/Gly-rich linker, naming this biosensor NES-flex-PABD-Spo20. The flexible linker was sufficient to stop nuclear localization of the sensor, although NES-flex-PABD-Spo20 still only showed slight responsiveness to PMA (Fig. 1 C). This suggests that the placement of the NES within a biosensor's structure is important for its efficacy. However, regardless of PABD-Spo20's basal localization, biosensors utilizing this domain are not very sensitive to PA levels at the PM.

To increase the PA-binding ability of the PABD-Spo20 biosensors, we replicated the design for a biosensor with tandem PABD domains (NES-PABDx2-Spo20) (Bohdanowicz et al., 2013). NES-PABDx2-Spo20 showed strong nuclear localization in our HEK293A cells, presumably due to steric hindrance by the same helical linker. However, the tandem PABDs did increase the response of NES-PABDx2-Spo20 to PMA (Fig. 1 D).

Finally, to try and make the NES in the tandem biosensor more accessible, we added an additional NES on the N terminus of the NG, naming this sensor NESx2-PABDx2-Spo20. The addition of this second nuclear export sequence did lower the amount of sensor localized in the nucleus of unstimulated cells. Although NESx2-PABDx2-Spo20 did not bind the PM, NES-PABDx2-Spo20 did after PMA stimulation (Fig. 1 E). It should be noted that of the Spo20 biosensors, NES-PABDx2-Spo20 had a larger response to PMA than PASS did and was thus the biosensor we used throughout the rest of this study (Fig. 1 J and Table S1).

To develop a PA biosensor that shows robust PM localization when PA is produced, we examined the LNS2 domain of *H. sapiens* Nir1. We defined this domain as residues 613–897, based on its structural prediction in AlphaFold (UniProt: Q9BZ71), which showed similarities to the lipin/Pah PAPs (Fig. 1 I and Fig. 2 D). This biosensor was named PILS-Nir1 as it utilizes the PA-interacting lipin-like sequence of Nir1.

In comparison with the Spo20-based sensors, NG-tagged PILS-Nir1 showed a fairly even distribution between the PM, the cytoplasm, and the nucleus in unstimulated HEK293A cells. After PMA stimulation, PILS-Nir1 noticeably translocated to the PM, more so than any of the Spo20-based biosensors (Fig. 1 F).

This suggests that PILS-Nir1 could serve as a highly sensitive PA biosensor and produce nicer images since its fluorescence does not saturate the nucleus.

We then tested the LNS2 domains of the other two Nir family members, Nir2 and Nir3, to determine their sensitivity to PA. The boundaries of the Nir2-LNS2 (UniProt: O00562) and Nir3-LNS2 (UniProt: Q9BZ72) were also defined using AlphaFold predictions of these domains. Previous definitions of the Nir2-LNS2 domain considered the domain smaller than we do here (Kim et al., 2013; Kim et al., 2015). However, according to AlphaFold, the boundaries set previously excluded a large N-terminal immunoglobulin domain that is conserved in the lipin/Pah PAPs, as well as disrupted the domain fold that is homologous to the lipin active site. Therefore, we set boundaries in our constructs to include the entire predicted LNS2 fold.

The Nir2-LNS2 and Nir3-LNS2 sensors did not have as strong of a response to PMA as PILS-Nir1 did (Fig. 1, G and H). When looking at the total area under the curve (AUC) for the Spo20 and Nir biosensors tested, we observed that all of the sensors responded to PMA significantly less than PILS-Nir1 did (Fig. 1 J and Table S1).

Next, we confirmed that the PM binding of PILS-Nir1 after PMA stimulation was dependent on PLD activation and an increase in PA. To do this, we simultaneously stimulated the cells with PMA and the PLD1/PLD2 inhibitor 5-fluoro-2-indolyl deschlorohalopemide (FIPI) (Su et al., 2009; Liang et al., 2019). Treatment with FIPI significantly reduced the translocation of PILS-Nir1 to the PM, demonstrating that PA produced by PLD was necessary for the PM localization of PILS-Nir1 (Fig. 1 K).

Altogether, this suggests that PILS-Nir1 is more sensitive to PA production by PLD than the Spo20 biosensors or the LNS2 domains from Nir2 and Nir3. Furthermore, PILS-Nir1 avoids the strong nuclear localization of the PABD-Spo20 helix. Therefore, we went on to further characterize the membrane binding of this domain to validate its use as a high-affinity PA biosensor.

## PILS-Nir1 shows specificity for PA and PIP$_2$ *in vitro*, based on a novel PABD structure

To experimentally probe the lipid-binding specificity of the PILS-Nir1 domain, we purified a recombinant 6xHis-tagged PILS-Nir1 protein (residues 604–912) from *E. coli* and performed liposome cosedimentation to monitor membrane recruitment. Liposomes were made with palmitoyl-oleoyl (PO) phospholipids to best represent the lipid composition of cellular membranes. In line with prior results, we observed no binding to liposomes only containing PC (Kim et al., 2013; Kim et al., 2015; Chang and Liou, 2015). Using this same PC background, we tested the efficacy of the PM lipids DAG, PA, phosphatidylserine (PS), PI4P, and PIP$_2$ in recruiting PILS-Nir1 to membranes. While PI serves as a substrate for PI4P and PIP$_2$ synthesis (collectively referred to as the phosphatidylinositol phosphates (PIPs)) at the PM, levels of PI at the PM are very low compared with the PIPs, and therefore, PI itself was not tested (Zewe et al., 2020; Pemberton et al., 2020).

We saw that PILS-Nir1 was specifically recruited to liposomes enriched with PA or PIP$_2$, but not to liposomes enriched with DAG or other anionic lipids such as PS or PI4P (Fig. 2 A). Overall,

this suggests that PILS-Nir1 binds to both PA and PIP$_2$ *in vitro* but does not generally bind all anionic lipids.

In addition, we found that PILS-Nir1 bound PA-rich liposomes in a concentration-dependent manner. By fitting a binding curve to these data, we determined that the interaction of PILS-Nir1 with PA had a K$_d$ value of 422 µM (Fig. 2 B). To confirm that the PILS-Nir1 associated with liposomes was folded properly and not aggregated protein that had been pelleted, we performed a liposome flotation assay. This assay showed that little PILS-Nir1 was aggregated after incubation with PA or PIP$_2$-containing liposomes (Fig S1, A and B). Circular dichroism (CD) analysis also showed that incubation of PILS-Nir1 with PA-containing liposomes did not change the CD spectra of PILS-Nir1. The spectra of PILS-Nir1 with and without PA liposomes both showed characteristic features of secondary structures, suggesting that membranes do not induce unfolding of PILS-Nir1 (Fig. S1 C).

We then tested PABD-Spo20's binding to PC, PA, and PIP$_2$ to determine the selectivity of PABD-Spo20 compared with PILS-Nir1. This experiment showed that PILS-Nir1 bound PA significantly better than PABD-Spo20, although both sensors showed high PIP$_2$ binding as well (Fig. 2 C).

We next determined how PILS-Nir1 binds to PA at the structural level. Sequence homology of PILS-Nir1 together with AlphaFold structural predictions showed a high degree of similarity to the lipin family of enzymes, minus key residues necessary for Mg$^{2+}$ binding and catalysis. The lipin catalytic motif DxDxT is partially conserved in PILS-Nir1 as a SIDGS motif spanning residues 742–746. We looked for positively charged residues nearby that could bind to the PA in the membrane and stabilize its position in the SIDGS site. The active site of the lipins has a nearby Lys residue, which was predicted to perform this role (Khayyo et al., 2020). AlphaFold analysis of PILS-Nir1 showed that the residue K820 similarly projects toward the SIDGS site where it would be able to contact the negatively charged PA (Fig. 2 D).

The conservation of these features between the lipins and PILS-Nir1 suggests that PA binds this positively charged residue near the SIDGS pocket within PILS-Nir1 (Kim et al., 2013; Khayyo et al., 2020). However, for efficient catalytic activity, the lipins also require an N-terminal amphipathic helix for membrane interaction. This helix is made up of residues 1–18 in *Tetrahymena thermophila* Pah2 (Khayyo et al., 2020), and residues 613–630 in the N terminus of PILS-Nir1 are predicted to form a similar amphipathic helix (Fig. 2 D).

We tested which of these features, residue K820, the amphipathic helix, and the SIDGS-containing domain, were necessary for PILS-Nir1 interaction with PA at the PM. To do this, we made two truncations of the PILS-Nir1 construct: PILS-Nir1-613-630 is the isolated amphipathic helix, while PILS-Nir1-631-894 is the rest of the domain excluding the helix but including the SIDGS motif. Surprisingly, neither truncated construct responded to PMA by binding the PM, and they even showed reduced basal PM localization (Fig. 2 E). This suggests that the amphipathic helix and the SIDGS-containing domain may both interact at the membrane for binding.

We probed into the suspected PA-binding residue K820 by mutating it into a negatively charged Glu residue, which should disrupt its interaction with the negatively charged PA. The K820E mutation completely ablated PILS-Nir1 localization at the PM under basal conditions, recruitment to the PM after PMA stimulation, and association with PA liposomes (Fig. 2, E and F). Interestingly, the K820E mutation did not alter PILS-Nir1 binding to PIP$_2$ *in vitro*, demonstrating the specificity of this site for PA and suggesting that the PIP$_2$ binding seen in the wild-type (WT) construct is simply due to an electrostatic interaction.

Altogether, our data suggest that PILS-Nir1 requires both the larger SIDGS-containing domain and the amphipathic helix for sustained binding to membrane-embedded PA, but that the PA specifically interacts with K820 near the SIDGS motif. Therefore, PILS-Nir1 demonstrates a novel PABD with a tertiary structure beyond the simple amphipathic helix found in Spo20.

Since the truncated and mutated PILS-Nir1 constructs showed reduced basal PM localization, we wanted to further characterize the basal localization of the WT PILS-Nir1. PILS-Nir1 localization varies between resting cells, but analysis of all the cells used throughout this study determined that the basal PM/Cyto ratio of the WT PILS-Nir1 is 1.141 ± 0.097 (mean ± SEM), which suggests that at resting conditions, PILS-Nir1 is slightly enriched at the PM (Fig. S2, A and D). When we did the same analysis for all the cells where we expressed NES-PABDx2-Spo20 or PASS, we observed that NES-PABDx2-Spo20 had a similar basal PM/Cyto ratio and spread of data to that of PILS-Nir1. PASS had a lower ratio than NES-PABDx2-Spo20 and PILS-Nir1, presumably due to its single PABD limiting its affinity for PA (Fig. S2, B, C, and E).

As the K820E mutation disrupted PILS-Nir1 PM association at rest, this suggests that the spread in the basal localization of these sensors reflects variable PA levels in the PM at resting conditions. Mass spectrometry data have estimated that PA comprises 2 mol% of the inner leaflet of the PM in resting red blood cells (Lorent et al., 2020). Furthermore, FRET-based imaging of PA has indicated that there are detectable levels of PA under basal conditions, and this approach also showed that there was variability in basal PA levels within individual cells of a population (Nishioka et al., 2010). Overall, our data suggest that the high affinity of PILS-Nir1 for PA is reflected in both its basal association with the PM and its response to stimulations such as PMA.

## Polyanionic lipids do not affect the association of PILS-Nir1 with the PM, but do affect NES-PABDx2-Spo20 membrane binding

Because PILS-Nir1 and PABD-Spo20 bound to PIP$_2$ *in vitro*, we investigated whether PIP$_2$ mediates the interaction of PILS-Nir1 or NES-PABDx2-Spo20 with the PM in HEK293A cells. To do this, we utilized a chemically inducible dimerization system with PIP phosphatases linked to FK506-binding protein from the mTOR complex (FKBP) and a PM-anchored FKBP rapamycin-binding domain (FRB). In short, cells expressing the system were stimulated with rapamycin. Rapamycin induced dimerization of the FKBP and FRB, thereby acutely localizing the phosphatases at the PM where they degraded PIPs. Then, we

determined the effects of the loss of specific PIPs on PILS-Nir1 and NES-PABDx2-Spo20 membrane binding using total internal reflection fluorescence (TIRF) microscopy. TIRF microscopy was used to selectively excite fluorophores near the bottom of the cell. The fluorescence at a given time ($F_t$) was divided by the fluorescence before rapamycin stimulation ($F_{pre}$) so that as biosensors moved off of the PM, the fluorescence ratio decreased.

To deplete both $PIP_2$ and PI4P, we used a chimeric construct Pseudojanin (PJ), consisting of the inositol polyphosphate 5-phosphatase E (INPP5E) and the *S. cerevisiae* Sac1 phosphatase (Hammond et al., 2012). PJ depletes $PIP_2$ sequentially, as the INPP5E domain dephosphorylates $PIP_2$ to produce PI4P, and then, the Sac1 domain dephosphorylates PI4P to produce PI. Then, as a negative control, we expressed a doubly catalytically dead mutant of PJ, referred to as PJ-Dead.

When PJ-Dead was recruited to the PM, we confirmed that $PIP_2$ and PI4P levels remained unaltered by seeing a stable association of the $PIP_2$ biosensor Tubby(c) with the PM. Additionally, we observed no loss of the PM localization of PILS-Nir1 or NES-PABDx2-Spo20 with PJ-Dead recruitment (Fig. 3 A). When the active PJ was recruited in HEK293A cells, we saw that PILS-Nir1 was able to remain associated with the PM, but that NES-PABDx2-Spo20 moved off the PM to a similar extent that the $PIP_2$ biosensor Tubby(c) moved off the PM (Fig. 3 B).

Since PJ depletes both $PIP_2$ and PI4P, we examined whether either of these lipids specifically contribute to PILS-Nir1 or NES-PABDx2-Spo20 membrane binding. FKBP-INPP5E was used to deplete $PIP_2$, but not PI4P at the PM, as seen by the significant loss of PM-localized Tubby(c). FKBP-PJ-Sac1, an FKBP-PJ construct that has a catalytically dead INPP5E domain, but an active Sac1 domain, was used to deplete PI4P without altering $PIP_2$ levels, as seen by removal of the PI4P biosensor P4Mx1 from the PM. The association of PILS-Nir1 was unaffected by either FKBP-INPP5E or FKBP-PJ-Sac1 recruitment (Fig. 3, C and D). However, there was a slight loss of NES-PABDx2-Spo20 at the PM upon FKBP-PJ-Sac1 degradation of PI4P (Fig. 3 D). These data suggest that PILS-Nir1 is specific for PA in cells even though it shows association with the anionic lipid $PIP_2$ *in vitro*. In contrast, decreasing the anionic charge of the membrane through depletion of PIPs does affect NES-PABDx2-Spo20's ability to associate with the PM. This is not too surprising given the previously reported interactions of Spo20 with PIPs (Nakanishi et al., 2004; Horchani et al., 2014).

## PA alone is sufficient for PILS-Nir1 membrane binding

An often-overlooked criterion for a lipid biosensor is showing that the lipid of interest is sufficient to recruit the sensor to membranes. Because we saw PILS-Nir1 to bind $PIP_2$ *in vitro*, but not be affected by depletion of $PIP_2$ *in vivo*, we first tested whether the PIPs were sufficient for PILS-Nir1 membrane binding. To do this, we designed a knock-sideways system to produce PIPs at the mitochondrial membrane and look at PILS-Nir1's interaction with lipids outside the context of the PM (Robinson et al., 2010). The mitochondrial membrane was chosen specifically for this as it is more isolated from the PM than organelles like the ER or Golgi with which the PM exchanges lipids by vesicular traffic.

To make PIPs at the mitochondria, we expressed a mito-FRB in cells. Then, we utilized the overexpression of an FKBP-tagged phosphatidylinositol 4-kinase (FKBP-PI4K) or the co-expression of FKBP-PI4K and an FKBP-tagged phosphatidylinositol 4-phosphate 5-kinase (FKBP-PIP5K). The FKBP-PIP5K was designed by adding point mutations in the full-length *H. sapiens* PIP5K1γ (D101R and R304D) that swap the charge of these positions to stop dimerization with endogenous PIP5Ks (Hu et al., 2015). Additionally, point mutations in the C-terminal domain (R445E and K446E) were used to stop the constitutive PM association of the enzyme so that we could use the FKBP/FRB system to acutely localize it to mitochondria (Suh et al., 2006). Once at the mitochondria, the FKBP-PI4K converts PI in the membrane into PI4P (Zewe et al., 2020). In cells that co-express the kinases, the PI4P made by FKBP-PI4K is further converted into $PIP_2$ by FKBP-PIP5K (Fig. 4 A).

We validated the efficacy of this system by using PH-PLCδ1 to monitor $PIP_2$ production at the mitochondria (Idevall-Hagren et al., 2012; Várnai and Balla, 1998). Rapamycin induced robust recruitment of the FKBP constructs to the mitochondria, which was then followed by PH-PLCδ1 translocation as $PIP_2$ was produced (Fig. 4 B). Upon production of PI4P or $PIP_2$ at mitochondria, we did not see any localization of PILS-Nir1 or NES-PABDx2-Spo20 at this organelle (Fig. 4 C). Overall, this suggests that PI4P and $PIP_2$ are not sufficient to recruit these PA biosensors.

Next, to test whether PA was sufficient to induce membrane binding of PILS-Nir1, we used a different chemically inducible dimerization system to produce DAG and PA at mitochondria. We used an FKBP-tagged *B. cereus* phosphatidylinositol PLC (PI-PLC), which uses PI as a substrate to produce DAG (Pemberton et al., 2020). This construct was then co-expressed with an FKBP-tagged catalytic fragment of DGKα to subsequently produce PA from the DAG (Fig. 4 D). Both FKBP constructs showed dimerization with mito-FRB leading to mitochondrial localization after rapamycin addition (Fig. 4 E). However, only cells co-expressing FKBP-PI-PLC and FKBP-DGKα showed any mitochondrial localization of PILS-Nir1 and NES-PABDx2-Spo20 (Fig. 4 F). This demonstrates that PA is sufficient to recruit these biosensors to membranes, while DAG, PI4P, and $PIP_2$ are not, thus substantiating that PILS-Nir1, and NES-PABDx2-Spo20 as well, fulfills this criterion for being valid PA biosensors.

## PILS-Nir1 detects PA produced downstream of PLC

We have shown that PILS-Nir1 responds to PLD activation after PMA addition and that PA is sufficient for PILS-Nir1 membrane binding, so next we investigated whether PILS-Nir1 is a useful probe to measure PA production downstream of PLC and DGK activation. To do this, we used carbachol (CCh) to stimulate the cholinergic receptor muscarinic 3 (CHRM3; referred to as M3) in HEK293A cells and activate PLC. The receptor antagonist atropine was then used to turn off signaling so that PA levels returned to baseline.

To maximize PLC activation, we overexpressed the M3 receptor in HEK293A cells. Interestingly, we saw that when cells overexpressed this receptor, the basal PM localization of PILS-Nir1 and NES-PABDx2-Spo20 was significantly

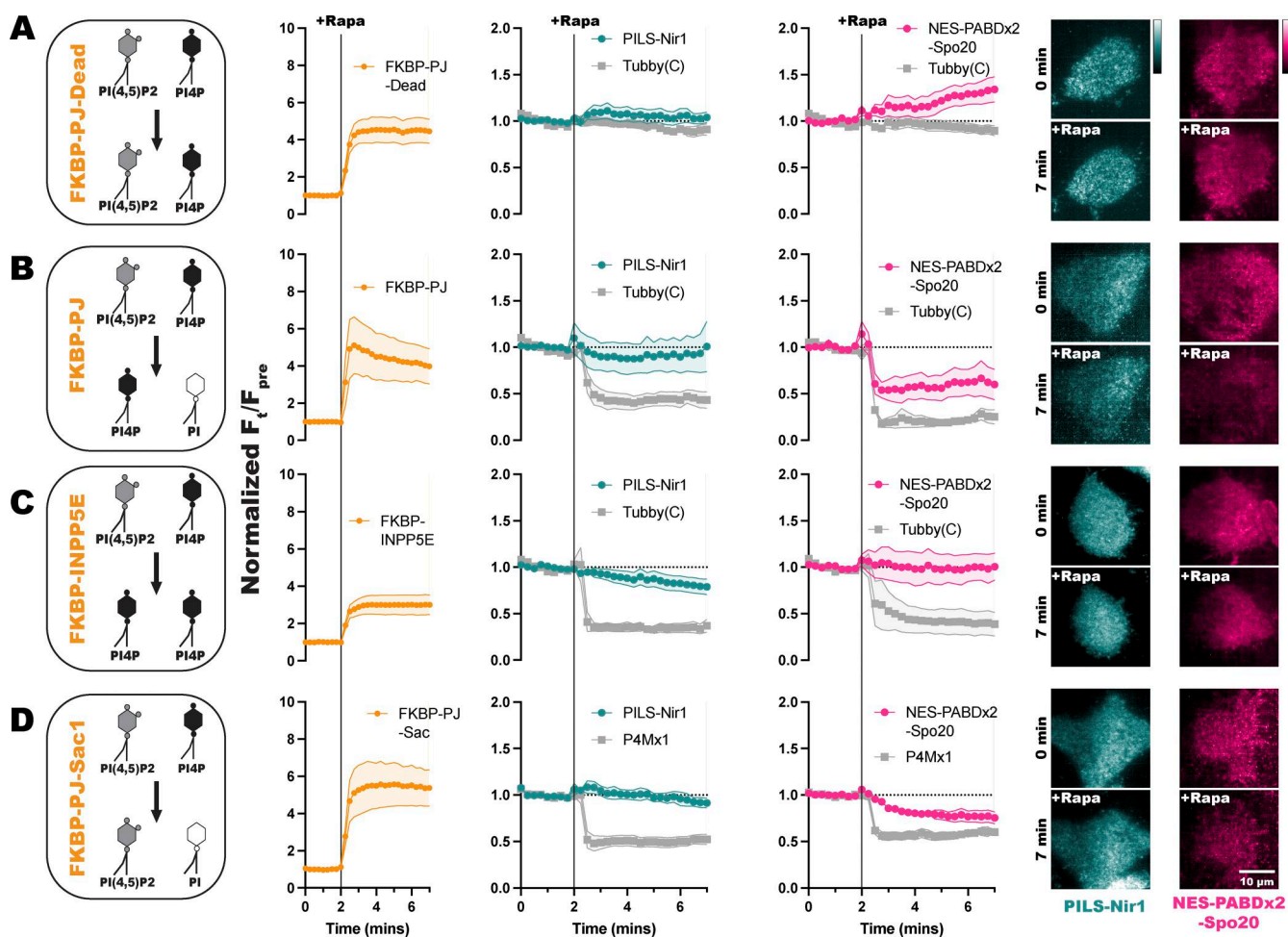

**Figure 3. Polyanionic lipids do not affect the association of PILS-Nir1 with the PM, but do affect NES-PABDx2-Spo20 membrane binding.** FKBP-tagged PIP phosphatases were recruited to the PM by 1 μM rapamycin (Rapa) inducing dimerization between the FKBP fragment and PM-localized FRB. Quantification of the recruitment of the FKBP-tagged constructs is shown in orange. The resulting depletion of PIPs is shown by the Tubby(C) biosensor for $PIP_2$ or the P4Mx1 biosensor for PI4P in gray. **(A)** FKBP-PJ-Dead does not affect $PIP_2$ or PI4P levels. **(B)** FKBP-PJ depletes both $PIP_2$ and PI4P at the PM. **(C)** FKBP-INPP5E depletes $PIP_2$ at the PM but does not reduce PI4P levels. **(D)** FKBP-PJ-Sac1 only depletes PI4P at the PM. Association of PILS-Nir1 (teal) or NES-PABDx2-Spo20 (pink) with the PM during recruitment of these phosphatases was determined using TIRF microscopy to analyze the fluorescence intensity at the basal membrane of the cells. The fluorescence at a given time ($F_t$) was divided by the fluorescence before rapamycin stimulation ($F_{pre}$). All xy graphs show the grand means of three to five experiments ± SEM. Total cells analyzed = 7–16. Representative TIRF images of the PILS-Nir1 and NES-PABDx2-Spo20 biosensors during rapamycin recruitment of the PIP phosphatases are shown to the right.

elevated compared with WT cells (Fig. 5 A). This suggests that there is some basal activity of the overexpressed receptors that these sensors respond to even without exogenous agonist addition.

A biosensor made up of the tandem C1 domains of protein kinase D (C1ab-Prkd1), which senses DAG, was used to directly monitor PLC output (Kim et al., 2011). However, we did not see any difference in the localization of C1ab-Prkd1 in the M3-overexpressing cells versus the WT cells (Fig. 5 A). We suspect that any increase in DAG is quickly converted to PA by the DGKs, as others have seen DAG clearance around 10 min with receptor overexpression (Kim et al., 2015).

We next validated that the addition of atropine was able to halt PLC activity by using the C1ab-Prkd1 biosensor. In control cells treated with CCh for 2 min and then vehicle, DAG levels remained elevated after 15 min. However, when treated with

CCh and then atropine, the DAG levels quickly returned to baseline (Fig. 5 B).

Similarly, PILS-Nir1 remained at the PM when cells were stimulated with CCh and then vehicle, presumably due to continued elevation of PA. However, in cells treated with CCh and then atropine, PILS-Nir1 localized to the PM after CCh was added but then returned to the cytoplasm over the 15-min treatment with atropine as PA levels declined (Fig. 5 C). Additionally, we observed that the PM accumulation of PILS-Nir1 after CCh addition lagged behind the accumulation of DAG, which is consistent with the conversion of DAG to PA by DGKs (Fig. 5 D). Overall, this experiment shows that PILS-Nir1 binding to the PM follows the expected kinetic profile of DGK-produced PA.

Since DAG is a small lipid that could potentially fit inside the PILS-Nir1 domain, we wanted to ensure that the observed PILS-

Figure 4. **PA alone is sufficient for PILS-Nir1 membrane binding. (A)** FKBP-PI4K and FKBP-PIP5K were co-expressed in HEK293A cells to convert PI into PI4P and then PIP$_2$ after the dimerization of the FKBP fragment and mitochondrial-localized FRB with 1 µM rapamycin. Cells expressing FKBP-PI4K alone were used to determine any effects of PI4P on PILS-Nir1 and NES-PABDx2-Spo20 membrane binding. **(B)** Quantification and representative images of FKBP recruitment and PIP$_2$ production as monitored by the PIP$_2$ biosensor PH-PLC δ1. **(C)** Localization of PILS-Nir1 and NES-PABDx2-Spo20 was unchanged upon ectopic PI4P and PIP$_2$ production. **(D)** FKBP-PI-PLC and FKBP-DGKα were co-expressed in cells and recruited to mitochondria to produce DAG from PI and subsequently produce PA from DAG. Control cells expressed FKBP-PI-PLC alone to look at the effects of DAG production on the PA biosensors. **(E)** Quantification and representative images of FKBP recruitment. **(F)** PILS-Nir1 and NES-PABDx2-Spo20 were only recruited to mitochondria where PA was produced by FKBP-DGKα. All experiments were performed three to four independent times, with the xy graphs showing the grand means of the experiments ± SEM. 33–45 total cells were analyzed. Note the PILS-Nir1–expressing cell shown in F is the same as shown in E.

Nir1 response to CCh was not due to direct DAG binding. We stimulated cells with DAG analogs and compared C1ab-Prkd1 and PILS-Nir1 localization 30 s after stimulation. We used 1,2-dioctanoyl-*sn*-glycerol (DiC8), 1-oleoyl-2-acetyl-*sn*-glycerol, phorbol 12,13-dibutyrate, or PMA, which are all analogs of endogenous DAG. As expected, the C1ab-Prkd1 biosensor robustly localized to the PM after 30 s of stimulation (Fig. 5 E), since it bound directly to the DAG analogs (Chen et al., 2008). However, none of the DAG analogs caused a large change in the localization of PILS-Nir1 in this time frame (Fig. 5 F). We did see some slight PM localization of PILS-Nir1 after DiC8 stimulation; however, these DAG analogs can activate PKC and subsequently PLD to produce PA (Selvy et al., 2011), which could cause the translocation of PILS-Nir1 seen.

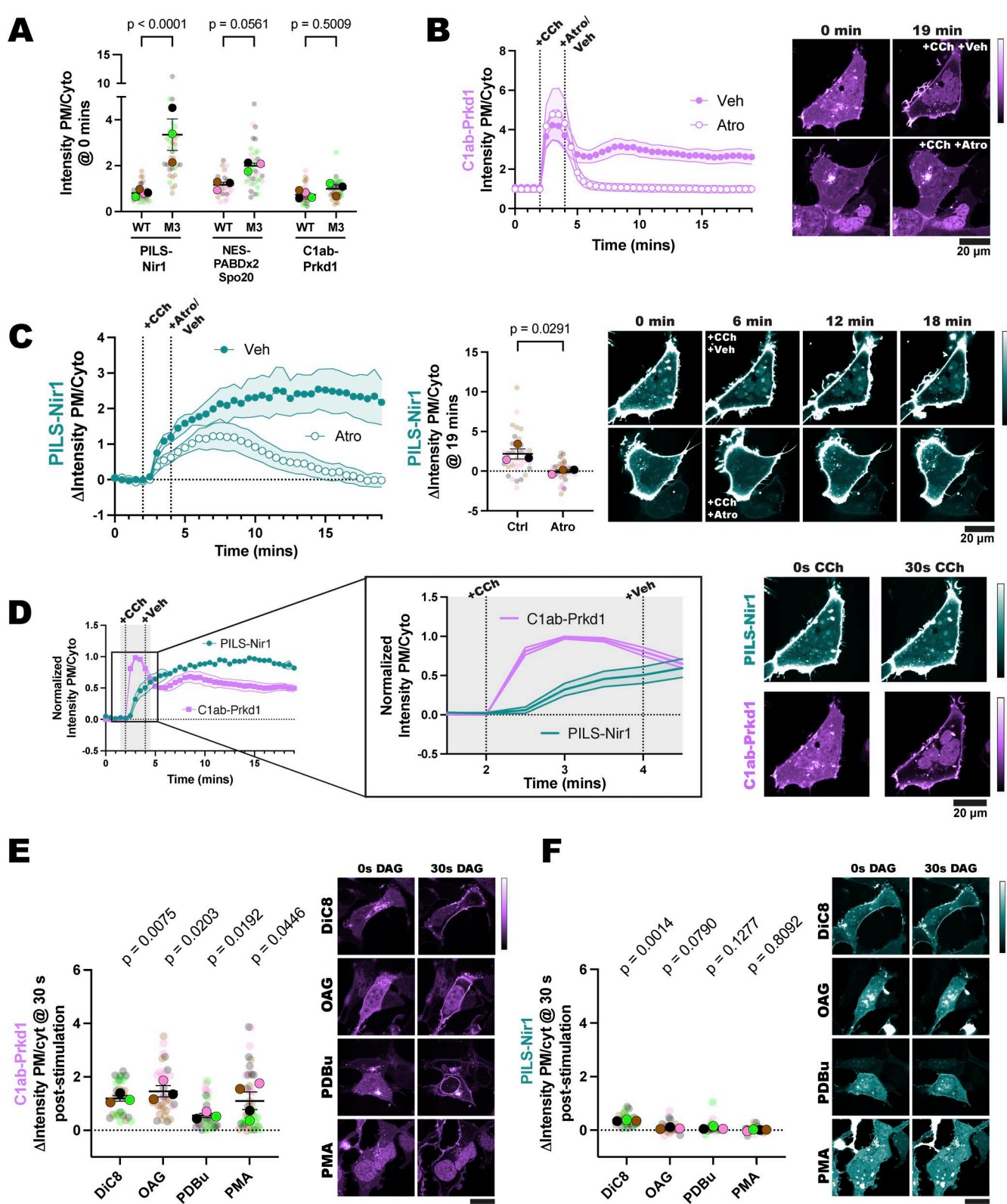

Figure 5. **PILS-Nir1 detects PA produced downstream of PLC. (A)** Overexpression of the M3 receptor in HEK293A cells elevated PILS-Nir1 and NES-PABDx2-Spo20 PM localization even without agonist addition. The DAG biosensor C1ab-Prkd1 localization was unchanged. Data were taken from time point 0 of experiments in 1D, 1F, 5B, and 5C. 30–46 cells (small symbols) were analyzed from 3 to 4 individual experiments (large symbols). Statistics were calculated using an ordinary two-way ANOVA on the average of each experimental replicate (large symbols, $n = 3$–4). DF = 1, MS = 7.185, $F_{(1, 14)} = 30.37$, $P < 0.0001$. **(B)** C1ab-Prkd1 showed the activation and subsequent attenuation of PLC signaling upon addition of 10 µM CCh for 2 min and then 5 µM atropine (Atro) for 15 min. Control cells were treated with cell media (veh) after CCh stimulation. **(C)** PILS-Nir1 response to CCh and then atropine treatment. Cells were treated as in

B. The scatter plot shows the change in intensity ratio of PILS-Nir1 at the final time point. Small symbols represent individual cells (*n* = 38–46), which are color-coded according to their experimental replicate as shown by the large symbols (*n* = 3–4). Statistics were calculated using an unpaired *t* test on the average AUC for each replicate (*n* = 3–4). A two-tailed P value was used, t = 3.328, df = 4. **(D)** PILS-Nir1 translocation to the PM (data replicated from 5C) lagged behind C1ab-Prkd1 (data replicated from 5B) in response to CCh addition. The data have been normalized so that the maximum value is 1. All xy graphs show the grand means of each experimental replicate ± SEM. **(E and F)** 30 s after HEK293A cells were stimulated with 150 μM DiC8, 150 μM OAG, 5 μM PDBu, or 100 nM PMA, C1ab-Prkd1 translocated to the PM (E), but PILS-Nir1 did not bind the DAG analogs at the PM (F). The small circles indicate the change in the intensity ratio of individual cells (*n* = 30–46) 30 s after stimulation. The large circles show the average change in intensity ratio for each experimental replicate (*n* = 3–4). Cells in each replicate are color-coded accordingly. Statistics were calculated using a one-sample t and a Wilcoxon test with 0 as the hypothetical value. Statistics used the average change in ratio of each experimental replicate (*n* = 3–4). PILS-Nir1-DiC8: t = 26.25, df = 2; PILS-Nir1-OAG: t = 3.343, df = 2; PILS-Nir1-PDBu: t = 2.523, df = 2; PILS-Nir1-PMA: t = 0.2635, df = 3. C1ab-Prkd1-DiC8: t = 11.49, df = 2; C1ab-Prkd1-OAG: t = 6.916, df = 2; C1ab-Prkd1-PDBu: t = 7.118, df = 2; C1ab-Prkd1-PMA: t = 3.334, df = 3. PMA data are duplicated from Fig. 1 F.

## PILS-Nir1 is a high-affinity PA biosensor that can be used to study endogenous PA signaling in a variety of contexts

So far, we have confirmed that PILS-Nir1 is a promising PA biosensor: the purified protein binds PA in artificial membranes, its membrane interactions in cells depend on PA, and PA is sufficient for its membrane localization. Next, we wanted to show that PILS-Nir1 is a high-affinity biosensor, and we wanted to demonstrate the applications of PILS-Nir1.

In Fig. 1, we saw that PILS-Nir1 bound PMA-stimulated PA at the PM with higher affinity than NES-PABDx2-Spo20 did. Then in Fig. 4, we saw that PILS-Nir1 was recruited to the mitochondria more robustly than NES-PABDx2-Spo20 after PA was produced by FKBP-PI-PLC and FKBP-DGKα. These data have been replicated in Fig. 6, A and B to facilitate comparison between the biosensors.

To see whether PILS-Nir1 showed higher affinity for PA in other organelles, we modified the FKBP-PI-PLC and FKBP-DGKα system by using FRB fragments that are targeted to other organelles: Golgi-FRB, Rab5-FRB, and ER-FRB. After recruitment of the FKBP constructs to these organelles, we saw that PILS-Nir1 responded to PA production in the Golgi and Rab5 endosomes with higher affinity than NES-PABDx2-Spo20 did (Fig. 6, C and D). When it came to PA produced at the ER, both biosensors responded only transiently, presumably due to the quick metabolism of PA in this compartment. However, we did see that NES-PABDx2-Spo20 showed a higher peak response in this organelle. It seems that the localization of this sensor in the nucleus helped it to respond to PA made in the ER that is continuous with the nuclear envelope, while PILS-Nir1 tended to label ER structures that were more distal (Fig. 6 E). Altogether, these data suggest that PILS-Nir1 can serve as a high-affinity PA biosensor at various cellular locations, although NES-PABDx2-Spo20 has some advantages when it comes to PA production in specific regions of the ER.

Next, we validated that PILS-Nir1 can be utilized in various model cell lines to show PA levels with high affinity. We expressed PILS-Nir1 and NES-PABDx2-Spo20 in African green monkey kidney cells (Cos7) and HeLa cells. We then stimulated the Cos7 cells with ATP and the HeLa cells with histamine to activate the cells' native PLC-coupled purinergic and histamine receptors, respectively. In Cos7 cells, NES-PABDx2-Spo20 responded to ATP just as robustly as PILS-Nir1 did; however, PILS-Nir1 showed less nuclear localization than NES-PABDx2-Spo20, which made it easier to image cells expressing PILS-Nir1 (Fig. 6 F). In HeLa cells, we saw that PILS-Nir1 showed much

greater PM binding upon PLC activation, as well as less nuclear localization when compared to NES-PABDx2-Spo20 (Fig. 6 G). This confirms that PILS-Nir1 still has some advantages over NES-PABDx2-Spo20 within a variety of cell lines.

As PILS-Nir1 shows high affinity for PA across cell lines, this brings up the concern that the use of PILS-Nir1 will sequester PA and inhibit endogenous signaling pathways that depend on PA, an effect that has been seen with other biosensors for low abundance lipids (Holmes et al., 2025). To determine whether this is the case, we used Nir2 MCS formation as a model PA-dependent event. Full-length Nir2 is localized to the ER by interaction of its FFAT motif with the VAPA/B proteins. Then when PA is produced at the PM, the PILS of Nir2 binds the PA, bridging the ER and the PM and forming an MCS (Cockcroft and Raghu, 2016). We can observe the formation of the MCS using TIRF microscopy. In TIRF, Nir2 localized on the ER can be seen as a hazy network, and then when Nir2 moves to the PM, the MCS appears as bright distinct puncta. This setup also avoids artifacts of Nir2 overexpression as only Nir2 interacting with endogenous VAPA/B is able to form MCS.

We co-expressed a GFP-tagged Nir2 and either iRFP-PILS-Nir1 or iRFP-Tubby(c), a PIP$_2$ biosensor that is not expected to affect MCS formation. It should be noted that although we have used the NG-tagged PILS-Nir1 throughout this work, iRFP and mCherry-tagged PILS-Nir1 sensors have behaved the same as the NG-tagged version in the experiments where we utilized them (data not shown).

We saw that there was no significant difference in Nir2 MCS formation after CCh stimulation in cells that were expressing iRFP-PILS-Nir1 compared with control cells expressing iRFP-Tubby(c) (Fig. 6 H). It is suggested that cellular homeostasis may compensate for the amount of bound lipid by increasing synthesis of free lipid, as this has been seen with the PIP$_2$ biosensor PH-PLCδ1 (Traynor-Kaplan et al., 2017). While PA has a plethora of cellular functions, the fact that PILS-Nir1 expression does not disrupt MCS formation shows promise that the high affinity of PILS-Nir1 will not inhibit downstream PA signaling.

## PILS-Nir1 reveals that PLD contributes to PA production downstream of PLC

The novelty of PILS-Nir1 is its high-affinity interaction with PA, and so we hypothesized that this high affinity would allow us to visualize subtle changes in PA levels that cannot be seen with Spo20-based biosensors. Therefore, we utilized PILS-Nir1 to determine how PA is produced downstream of M3 activation.

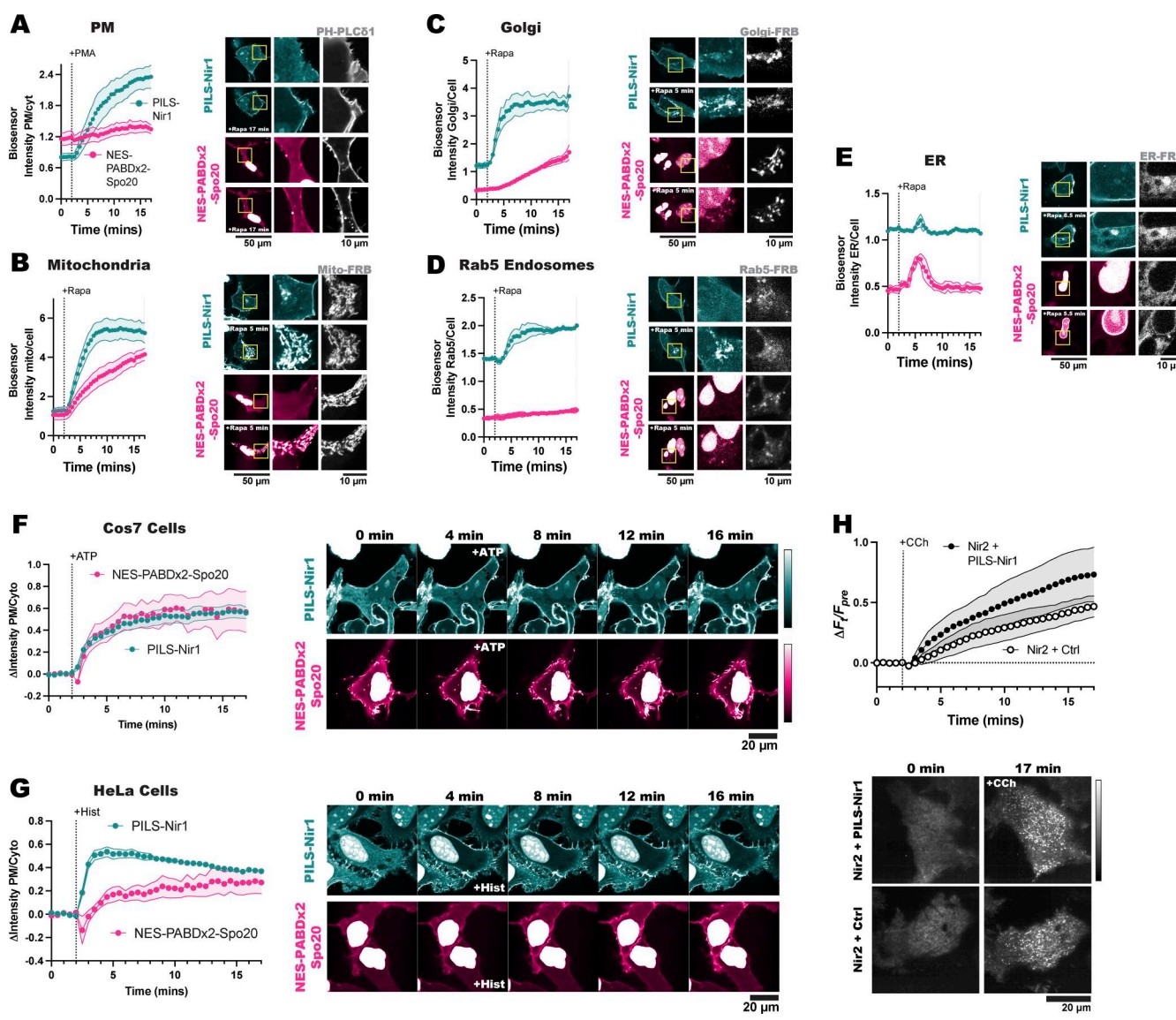

Figure 6. **PILS-Nir1 is a high-affinity PA biosensor that can be used to study endogenous PA signaling in a variety of contexts. (A–E)** Comparison of PILS-Nir1 and NES-PABDx2-Spo20 responses at the PM after stimulation by 100 nM PMA (A), or at the mitochondria (B), Golgi (C), Rab5 endosomes (D), or ER (E) after recruitment of FKBP-PI-PLC and FKBP-DGKα with 1 µM rapamycin. Organelle markers are shown in gray. Graphs show the grand means ± SEM of 3–4 experiments with 30–54 cells. Data in A are replicated from Fig. 1, D and F. Data in B are replicated from Fig. 4 F. **(F)** Cos7 cells transfected with PILS-Nir1 and NES-PABDx2-Spo20 showed the biosensors' response to 100 µM ATP. **(G)** HeLa cells were transfected with the biosensors to compare the response to treatment with 100 µM histamine. Graphs show grand means ± SEM for three to four experiments. A total of 33–57 cells were analyzed. **(H)** Nir2 MCS formation was quantified as the change in fluorescence at a given time ($F_t$) divided by the fluorescence before 100-µM CCh stimulation ($F_{pre}$). GFP-Nir2 was co-expressed with either iRFP-PILS-Nir1 or a control biosensor iRFP-Tubby(c), which binds PIP$_2$. The graph shows the grand means ± SEM for 4–5 experimental replicates ($n$ = 44–48 cells).

Stimulation of this receptor with CCh activates PLC and DGK to produce PA but is also thought to activate PKC, which then activates PLD (Shulga et al., 2011; Liang et al., 2019). We investigated the specific role of PLD in M3 signaling and whether the PILS-Nir1 biosensor could detect PLD's contribution to PA levels. To do this, we pretreated HEK293A cells with the PLD1/2 inhibitor FIPI, then treated with CCh. We did not see effects of FIPI alone on PILS-Nir1 localization before CCh addition. However, we did see a reduced PILS-Nir1 response to CCh when cells were pretreated with FIPI (Fig. 7 A). This suggests that PLD is making a small contribution to PA levels downstream of PLC. However,

when using NES-PABDx2-Spo20, we did not see any difference in the response of the sensor to CCh when FIPI was used (Fig. 7 B). It should also be noted that PILS-Nir1 was more responsive to PLC activation by CCh than NES-PABDx2-Spo20 is, just as we saw with PLD activation downstream of PMA. This suggests that the high affinity of PILS-Nir1 is indeed necessary to deconvolve PLD activity from that of DGK activity downstream of PLC. Overall, PA is a lipid with complex regulatory mechanisms that only a high-affinity sensor such as PILS-Nir1 can untangle. Therefore, we anticipate PILS-Nir1 will greatly impact future experiments dissecting PA regulation.

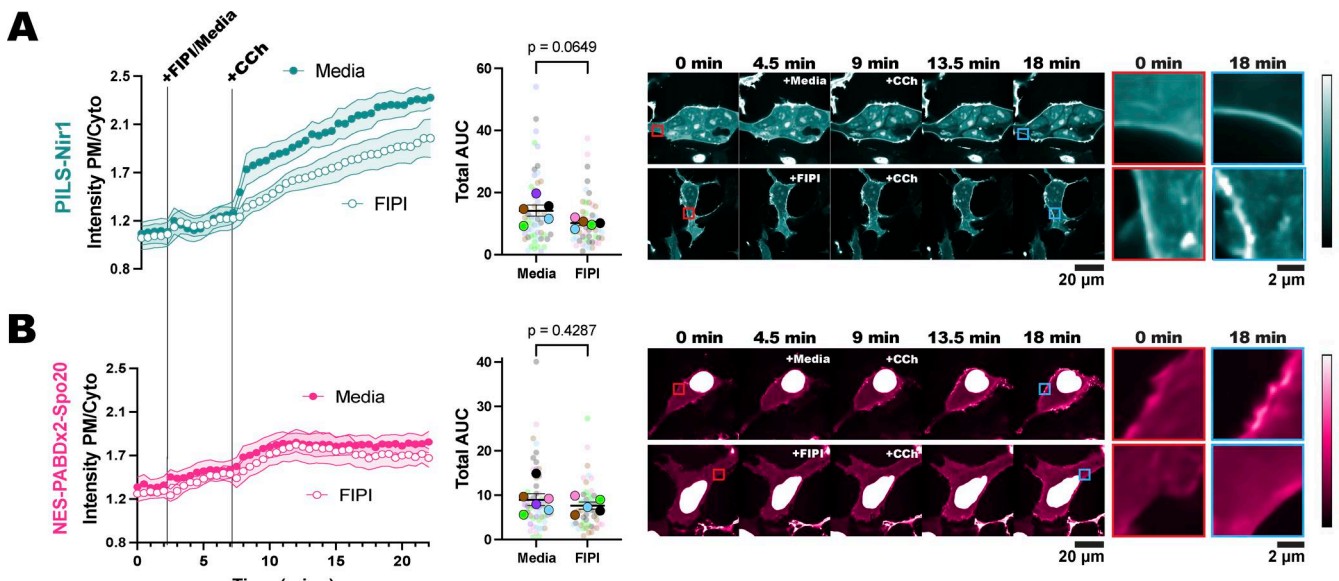

**Figure 7. PILS-Nir1 reveals that PLD contributes to PA production downstream of PLC. (A and B)** HEK293A cells expressing either PILS-Nir1 (A) or NES-PABDx2-Spo20 (B) were treated with 750 nM FIPI for 5 min to inhibit PLD activity or cell media as a control, and then, 5 μM CCh was added for 15 min to induce M3 receptor signaling. The red insets show the biosensors at a region of the PM before stimulation, while the blue insets show the same region after stimulation. The xy graphs show the grand means of five to six experiments ± SEM. The scatter plots show the AUC of individual cells' responses as the small symbols (*n* = 48–60) and average AUC of the experimental replicates as the large symbols (*n* = 5–6). Cells are color-coded according to their experimental replicate. Statistics were calculated using Student's *t* test on the average replicate AUC (*n* = 5–6). For PILS-Nir1, t = 2.139, df = 8. For NES-PABDx2-Spo20, t = 0.8288, df = 9. In both tests, a two-tailed P value was used.

## Discussion

In this study, we set out to validate the PILS-Nir1 domain as a novel PA biosensor by characterizing its membrane interactions both *in vitro* and in cells. We saw that PILS-Nir1 offers several advantages over the current PA biosensors based on the Spo20 PABD. Namely, PILS-Nir1 has a robust response to PLD's indirect activation by PMA, indicating a high-affinity interaction with PA, and PILS-Nir1 provides clearer confocal images by avoiding heavy nuclear localization (Fig. 1). We then characterized PILS-Nir1 lipid binding using liposomes and determined that PILS-Nir1 binds PA *in vitro* better than PABD-Spo20 does, due to a novel PABD domain structure similar to the lipin/Pah family. However, PILS-Nir1 and PABD-Spo20 both show *in vitro* PIP$_2$ binding as well (Fig. 2). Although, in live cells, PILS-Nir1 is not affected by loss of PM PIPs, only PA is sufficient to recruit PILS-Nir1 to membranes (Figs. 3 and 4). Next, we showed that PILS-Nir1 detects PA made downstream of PLC and that the membrane binding of PILS-Nir1 depends on the presence of PA (Fig. 5). Altogether, these data show that PILS-Nir1 meets the criteria for a valid biosensor. We then demonstrated that PILS-Nir1 can be used in a variety of organelles and cell types and does not seem to disrupt downstream PA signaling (Fig. 6). Importantly, PILS-Nir1 can be used to detect subtle contributions of PLD to the pool of PA that has been difficult to image with previous tools (Fig. 7). Overall, we have characterized PILS-Nir1 as a novel and high-affinity PA biosensor that can be applicable in diverse studies of the PA pathway.

The lack of PILS-Nir1 binding to DAG-rich liposomes (Fig. 2), DAG produced at the mitochondria (Fig. 4), and DAG analogs (Fig. 5) shows that the Nir family LNS2 domain only binds to PA rather than to PA and DAG as has been reported previously (Kim et al., 2015). In this study, we redefine the boundaries of the Nir family LNS2 domain based on the structure of the lipin/Pah family domains and the AlphaFold prediction for Nir1. The new boundaries include the entire fold that is conserved between the lipin/Pah family and the Nir family (Fig. 2 D). Therefore, we suspect that the extended boundaries of the LNS2 domain in our work explain the differences in our data and the published literature regarding DAG binding. Importantly, the data obtained with our amended LNS2 (a.k.a. PILS) suggest that within the context of the lipid transfer cycle and MCS formation, the Nir family of PITPs translocate to the PM solely based on PA. This information will be important as the field continues to determine the exact mechanism of the Nir PITPs in lipid homeostasis.

As far as the use of PILS-Nir1 as a biosensor, one caveat is the discrepancy in its specificity: *in vitro*, PA and PIP$_2$ are sufficient to recruit PILS-Nir1 to PC liposomes (Fig. 2), but *in vivo*, only PA is sufficient for mitochondrial recruitment (Fig. 4).

One reason for this discrepancy could be differences in the negative charge of cellular membranes versus that of liposomes. To interact with membranes, PILS-Nir1 requires both K820 near the SIDGS pocket, which is specific for PA binding, and an N-terminal amphipathic helix, which is thought to generally interact with negatively charged lipids (Kim et al., 2013; Khayyo et al., 2020) (Fig. 2). The negative charge of PIP$_2$ and therefore its ability to recruit the N-terminal helix of PILS-Nir1 depend on its protonation state (Kooijman et al., 2009). PIP$_2$ can hydrogen-bond with other lipids such as PI, with itself inside of PIP$_2$-rich domains, or even with neighboring proteins, all of which would attenuate its charge. Phosphatidylethanolamine, on the other

hand, increases PIP$_2$ ionization and its negative charge (Graber et al., 2012; Borges-Araújo and Fernandes, 2020). Therefore, the degree of charge on PIP$_2$ is greatly dependent on the local protein and lipid environment. Due to this, PIP$_2$ molecules in cellular membranes may possess less charge compared with PIP$_2$ molecules in liposomes. Thus, in cells, the N-terminal helix's interaction with PIP$_2$ may be very weak compared with the interaction of K820 with PA so that only PA levels influence membrane binding.

Differences in biosensor specificity *in vitro* and *in vivo* have been seen for other biosensors as well. For example, the PH domains of OSBP and FAPP1 bound to liposomes in a PI4P- and PIP$_2$-dependent manner. However, membrane interactions of these probes in yeast were only dependent on PI4P production (Levine and Munro, 2002). Even with these discrepancies, the biosensors have still proved to be useful in studies of PI4P-dependent processes (Szentpetery et al., 2010). Similarly, we believe that PILS-Nir1 will serve as a valid PA biosensor for future studies in live cells.

PILS-Nir1's major use as a PA biosensor stems from its highly sensitive membrane recruitment by PA. The widely used Spo20 biosensors have lower responsiveness to PA production as compared to PILS-Nir1 (Fig. 6, A–E), and the high affinity of PILS-Nir1 allows us to now more easily sense subtle changes in the pool of PA (Fig. 7). Previous studies have used PABD-Spo20 to successfully show the specific activity of PLD1 during exocytosis by using PLD1 siRNA (Zeniou-Meyer et al., 2007). Other groups have used FIPI to look at the effect of PLD on basal PA levels and saw that the effects of FIPI on NES-PABDx2-Spo20 varied depending on the cell type used (Bohdanowicz et al., 2013).

Looking at PLD activation specifically downstream of muscarinic receptors has remained difficult until a recent study used click chemistry to repurpose the transphosphatidylation reaction catalyzed by PLD to create clickable lipids that can incorporate fluorescent reporters, a technique referred to as real-time IMPACT (Liang et al., 2019). This study corroborated our results, showing that PLD is activated at the PM by stimulation of the muscarinic M1 receptor.

However, real-time IMPACT does not directly report on PA levels as it creates a bio-orthogonal fluorescent lipid. Instead, it offers several advantages such as being able to interrogate lipid trafficking over time. Since the resulting phosphatidyl alcohols are not rapidly metabolized *via* the same pathways as PA, the fate of these lipids can be continuously monitored. Thus, these PLD-produced fluorescent lipids were determined to traffic from the PM to the ER with a half-life of around 104 s, which we are not able to directly observe using PILS-Nir1 or the Spo20-based biosensors (Liang et al., 2019).

However, we see Nir2 MCS formation occurs in cells expressing the biosensors (Fig. 6 H), which is thought to mediate PA trafficking to the ER (Kim et al., 2015), Additionally, we see a quick loss of biosensor signal in the ER after PA is produced by the chemically inducible dimerization system (Fig. 6 E), so we believe trafficking to and metabolism in the ER are still occurring in our system even if the PILS-Nir1 probe cannot be used to directly visualize the intermembrane transfer.

Overall, we want to emphasize that PILS-Nir1 should not replace current tools such as real-time IMPACT or Spo20 biosensors such as PASS. In this work, we have further characterized the Spo20 PABD by demonstrating that PA is sufficient to recruit NES-PABDx2-Spo20 to membranes (Fig. 4). Therefore, our data support Spo20 biosensors as valid and robustly characterized options for low-affinity PA biosensors. There are various situations where it is particularly useful to have both a low-affinity and a high-affinity lipid biosensor. For instance, high-affinity biosensors are not very effective at quantifying increases in a lipid as the sensor can already be saturated on the membrane. Conversely, low-affinity biosensors struggle to show decreases in a lipid since there is already so much noise in the cytosol (Wills et al., 2018). A recent paper demonstrated the usefulness of having multiple biosensors, describing both a high-affinity cholesterol biosensor and its low-affinity counterpart. The authors used these sensors in parallel to successfully detect decreases and increases in accessible cholesterol (Koh et al., 2023). Therefore, we introduce PILS-Nir1 as a novel tool in the study of PA to be used in combination with existing tools to aid in our understanding of this important lipid.

## Materials and methods
### Protein overexpression and purification
The full-length Nir1 gene (accession code: NC_000017.11) was codon-optimized for expression in *E. coli* and gene synthesized (Twist Bioscience) in the pET28 plasmid. DNA oligo primers were synthesized (Integrated DNA Technologies) for PILS-Nir1 using residues 609–912 and inserted into the pTHT vector, a modified pET-28 plasmid containing a TEV-cleavable N-terminal 6xHis-tag. The construct was verified with direct sequencing. The PILS-Nir1-K820E mutant was generated using Q5 Site-Directed Mutagenesis Kit (New England Biolabs) in the pTHT plasmid and was verified with direct sequencing. A verified plasmid containing the PABD of Spo20 (residues 51–91) in tandem with an N-terminal glutathione S-transferase (GST) purification tag was gifted to us graciously from Dr. Aaron Neiman at Stony Brook University (Neiman et al., 2000).

All plasmids were transformed into competent BL21 (DE3) RIPL cells (Cat. No. 230280; Agilent Technologies) for protein overexpression. Cells were grown at 37°C to an OD600 of 1.5, and cooled at 10°C for 1 h before inducing protein expression with 100 µM isopropyl β-D-1-thiogalactopyranoside at 15°C for overnight growth. Cell pellets were harvested and lysed *via* sonication in buffer comprised of 50 mM Tris, pH 7.5, 500 mM NaCl, 5% glycerol, 1% Triton X-100, and 2 mM beta-mercaptoethanol (βME), and lysates were centrifuged at 82,000 *g* at 4°C for 1 h. The protein-rich supernatant was collected, and the 6xHis-tagged proteins were isolated using Ni-NTA gravity flow affinity chromatography and eluted with buffer comprised of 50 mM Tris, pH 7.5, 500 mM NaCl, 300 mM imidazole, pH 7.5, and 5 mM βME. Spo20-GST was captured from supernatant using glutathione resin gravity flow chromatography and was eluted in buffer containing freshly prepared 10 mM reduced glutathione, 50 mM Tris buffer, pH 7.5, and 500 mM NaCl. All proteins were applied to a Superdex 75 26/60 HiLoad column

(GE Healthcare) equilibrated with buffer comprised of 20 mM Tris, pH 7.5, 150 mM NaCl, 10 mM βME, and 1 mM dithiothreitol. The purified proteins were concentrated to 1–5 mg/ml, flash-frozen in liquid nitrogen in 30 µl aliquots, and stored at –80°C.

## Liposome Cosedimentation

PO phospholipids (Avanti Polar Lipids) dissolved in a chloroform: methanol solution were dried under nitrogen gas and resuspended in Buffer A comprised of 150 mM NaCl and 20 mM Tris, pH 7.5, to generate a 2 mM solubilized lipid mixture. Solubilized PO lipids underwent five freeze/thaw cycles with liquid nitrogen and were subsequently sonicated for 2 min in a water bath. 50 µl of pure protein at ~1 mg/ml was incubated with 50 µl of liposome mixture for 30 min at room temperature. Reactions were then centrifuged in a vacuum for 1 h at 100,000 $g$ at 4°C. 75 µl of the supernatant (S) was collected. The liposome pellet (P) was resuspended in 100 µl of Buffer A, and 75 µl was collected for samples to be resolved *via* SDS-PAGE. ImageJ software was used to quantify pixel intensity of the S and P fraction gel bands for each condition, and percent protein bound was found using the following equation:

$$\left( \frac{P}{P + S} \right) \times 100$$

The $K_d$ of PILS-Nir1 binding to PA was calculated with the following equation for a one-site specific binding with background:

$$Y = BG + \left( \frac{(B_{max} - BG) \times X}{K_d + X} \right)$$

Where X = % PILS-Nir1 in the pellet, BG = background, and $B_{max}$ = maximum % bound.

## Cell culture and transfection

HEK293A cells (R70507; Invitrogen), Cos7 cells (CRL-1651; ATCC), and HeLa cells (CCL-2; ATCC) were maintained in complete Dulbecco's modified Eagle's medium (DMEM) comprised of low-glucose DMEM (10567022; Life Technologies), 10% heat-inactivated fetal bovine serum (HI-FBS) (10438-034; Life Technologies), 1% 10,000 units/ml streptomycin + penicillin (15140122; Life Technologies), and 0.1% chemically defined lipid supplement (11905031; Life Technologies). Cells were grown at 37°C and 5% $CO_2$. The lines were passaged with a 1:5 dilution twice per week after rinsing with PBS and dissociating in TrypLE (12604039; Life Technologies).

For imaging, cells were seeded onto coated 35-mm glass-bottom dishes with a 20-mm glass aperture (D35-20-1.5-N; Thermo Fisher Scientific). HEK293A cells were seeded onto dishes that had been coated with 10 µg ECL cell attachment matrix (08-110; Sigma-Aldrich) diluted in 0.5 ml DMEM per dish or Stem Cell Qualified ECM gel (CC131; Sigma-Aldrich) diluted 1:80 in 0.5 ml DMEM per dish. HeLa and COS-7 cells were seeded onto dishes coated with 5 µg fibronectin (33016-015; Life Technologies) in 0.5 ml diH2O per dish. The volume of cells seeded was calculated so that cells would be 90–100% confluent on the

day of confocal imaging and 40–50% confluent on the day of TIRF imaging.

After allowing cells to adhere and spread on the dish for 2+ h, plasmids were transfected into the cells using Lipofectamine 2000 (11668019; Life Technologies). 1 µg of DNA was complexed with 3 µg of Lipofectamine 2000. This mixture was diluted to 200 µl in Opti-MEM (51985091; Life Technologies) and incubated for 5 min up to 2 h at room temperature before being added to the cells. HEK293A and HeLa cells were treated with the DNA and Lipofectamine solution for 3–4 h before the solution was removed, and the cells were placed in 1.6–2 ml of imaging media for imaging the next day. For Cos7 cells, the DNA and Lipofectamine solution were left on the cells for 12–16 h before being replaced with the appropriate imaging media.

The imaging media, complete HEPES-buffered imaging media (CHIM), were made of FluoroBrite media (A1896702; Life Technologies), 10% HI-FBS, 1% GlutaMAX (35050061; Life Technologies), 25 mM Na-HEPES, pH 7.4 (EM-5320; VWR), and 0.1% chemically defined lipid supplement. In some experiments, serum-free CHIM + 0.1% BSA (SF-CHIM + 0.1% BSA) was used for imaging. This medium was made using the same recipe as CHIM, excluding the HI-FBS and supplementing with 0.1% bovine albumin fraction V solution (BSA; 15260-037; Life Technologies). CHIM was used for the experiments in Fig. 3, Fig. 4, Fig. 5, A–D, and Fig. 6, while SF-CHIM + 0.1% BSA was used in Fig. 1; Fig. 2; Fig. 5, E and F; and Fig. 7.

## Confocal microscopy

The transfected cells were imaged on a Nikon A1R-HD resonant laser scanning confocal microscope, using an inverted TiE microscope stand. The entire stage was enclosed in a chamber maintained at 37°C (Tokai Hit). Resonant mode was used with a 100× 1.45 NA plan-apochromatic oil immersion objective. A dual fiber-coupled LUN-V laser launch was used to excite fluorophores. One line scan used 488- and 640-nm lasers to co-excite green (NG or EGFP) and far-red (iRFP) fluorescence. A second line scan used 561- and 405-nm lasers to co-excite red (mCherry or mRFP) and blue (BFP) fluorescence to avoid crosstalk. Emission was collected using individual filters for blue (425–475 nm), green (500–550 nm), yellow/orange (570–620 nm), and far-red (663–737 nm). The pinhole used was calculated to be 1.2× the Airy disk size of the longest wavelength channel used in the experiment. To decrease noise in the images, 8× frame averages were taken, and in some experiments, Nikon Elements denoising software was used.

To stain the PM with CellMask Deep Red (C10046; Life Technologies), a 2.5 ng/µl solution was made up in the appropriate cell imaging media. Cells were incubated with 500 µl of the CellMask solution for 5 min. The cells were then washed once with imaging media, and 1.6–2 ml of imaging media was added for imaging.

Imaging was performed for the time courses as indicated in the figures/legends. 10–15 fields of cells were selected and imaged every 30 s. Stimulations were added after the time point indicated in the figures/legends. Cell stimulations were created by diluting the reagents in the appropriate imaging media as outlined in Table 1. The stimulations were made at a

JCB

Table 1. **Reagents used for cell stimulation throughout this study**

| Reagent | Manufacturer | Catalog number | Stock solution | Storage temperature | Concentration added to cells (diluted in cell media) |
|---|---|---|---|---|---|
| PMA | MilliporeSigma | P8139 | 437 µM in DMSO | −20°C | 100 nM |
| FIPI | MilliporeSigma | 528245 | 750 µM in DMSO | −20°C | 750 nM |
| DiC8 | EMD Millipore | 317505 | 30 mM in methanol, dried. Resuspended in 50 µl methanol before use | −80°C | 150 µM |
| OAG | EMD Millipore | 495414 | 30 mM in methanol, dried. Resuspended in 20 µl DMSO before use | −80°C | 150 µM |
| PDBu | Sigma-Aldrich | P1269 | 10 mM in DMSO | −80°C | 5 µM |
| Rapamycin | Sigma-Aldrich | 553210 | 1 mM in DMSO | −20°C | 1 µM |
| CCh | Thermo Fisher Scientific | AC10824 | 50 mM in dH$_2$O | −20°C | 5 µM (Fig. 7), 10 M (Fig. 5), or 100 µM (Fig. 6) |
| Atropine | Sigma-Aldrich | A0257 | 25 mM in dH$_2$O | 4°C | 5 µM |
| ATP | Sigma-Aldrich | 10127523001 | 100 mM in 100 mM MgCl$_2$ + 200 mM Tris base | −20°C | 100 µM |
| Histamine | Thermo Fisher Scientific | AC15062 | 100 mM in dH$_2$O | −20°C | 100 µM |

5× concentration, and then, 500 µl of stimulation was added to the 2 ml of imaging media in the dish to produce the final concentration described in Table 1. For experiments that used two consecutive stimulations, cells were imaged in 1.6 ml media and 400 µl of the first stimulation was added and then 500 µl of the second stimulation was added.

## Confocal image quantification

Confocal image analysis was done using FIJI and custom macros (Schindelin et al., 2012). Images were imported into FIJI and then displayed as a montage of all xy positions for each specific channel. ROIs were then drawn in the background, around the cell, and, in some experiments, within the cytosol of each cell. Cells that moved too much during imaging were excluded from analysis.

The signal of constructs at the PM or at specific organelles was quantified by using a PM marker (iRFP-PH-PLCδ1, BFP-HRAS-CAAX, mCh-HRAS-CAAX, PM-FRB, or CellMask deep red) or organelle marker (mito-FRB, ER-FRB, Golgi-FRB, Rab5-FRB) to create a binary mask at the relevant organelle.

To create these masks, the PM, mito, and Golgi images were filtered with a Gaussian blur filter at 1×, 2×, 3×, and 4× the airy disk size of the marker fluorophore. The ER images were filtered at 1× and 2× the Airy disk size, and the Rab5 images were filtered at 1×, 2×, and 3× the Airy disk size. Wavelets were then generated by subtracting each filtered image from the image filtered at the next smaller length scale. The wavelets were multiplied, and a threshold of 0.5× standard deviations of the original image was applied. The mask then underwent a 1- or 2-pixel dilation cycle to ensure that the whole area of the relevant organelle was included.

The resulting mask was then used to measure the normalized intensity of a given construct over time at these membranes. Then, the intensity of the construct within the mask was divided by the intensity within either a cytoplasmic or whole-cell ROI to

create the reported ratios. Background fluorescence was subtracted using the pixel intensity within the background ROI. Further details on this analysis protocol can be found in Wills et al. (2021).

## TIRF microscopy

Transfected cells were imaged on a Nikon motorized TIRF illuminator mounted on a Nikon TiE inverted microscope stand, using a 100× 1.45 NA plan-apochromatic objective. The entire stage was enclosed in a chamber maintained at 37°C (Tokai Hit). An Oxxius L4C laser launch was used to excite the following fluorophores: 488 nm for EGFP/NG, 561 nm for mCherry, and 638 nm for iRFP. Single-pass chroma filters were used to collect yellow/orange (570–620 nm) and green (505–550 nm) emission, a dual-pass green/far-red filter was used to collect far-red emission (650–850 nm), and a dual-pass blue/yellow/orange filter was used to collect blue emission (420–480 nm). To image the time lapse, 10–15 individual fields were marked and imaged every 15–30 s using a Hamamatsu ORCA-Fusion BT sCMOS camera. The fields were imaged using an exposure time of 50–100 ms and 2 × 2 pixel binning.

Stimulations were added after 2 min of baseline imaging, as indicated in the figure legends. Cell stimulations were created by diluting the reagents in the appropriate imaging media to create a 5× solution. Then, 500 µl of stimulation was added to the 2 ml of imaging media in the dish to produce the final concentration described in Table 1.

## TIRF microscopy image quantification

TIRF microscopy image analysis was done using FIJI and custom-written macros (Schindelin et al., 2012). Images were imported into FIJI and then displayed as montages of each position in each channel. ROIs around the cell footprint were drawn using a minimum intensity projection to account for any movement of

Table 2.  **Plasmids used in this study**

| Shorthand name | Sequence | Reference |
|---|---|---|
| NES-PABD-Spo20 | *X. laevis* map2k1.L[32-44]-APVAT-EGFP-GLRSRASI-*S. cerevisiae* Spo20p[51-91] | Zeniou-Meyer et al. (2007), Zhang et al. (2014) |
| PASS | EGFP-SGLRSRA-*M. musculus* PKIa[34-51]-SR-*S. cerevisiae* Spo20p[51-91] | Zhang et al. (2014), Zeniou-Meyer et al. (2007) |
| NES-flex-PABD-Spo20 | *X. laevis* map2k1.L[32-44]-APVAT-EGFP-SGGGSGGS-*S. cerevisiae* Spo20p[51-91] | This study |
| NES-PABDx2-Spo20 | *X. laevis* map2k1.L[32-44]-APVAT-EGFP-SGLRSRA-*S. cerevisiae* Spo20p[51-91]- *S. cerevisiae* Spo20p[51-91] | Bohdanowicz et al. (2013) |
| NESx2-PABDx2-Spo20 | *X. laevis* map2k1.L[32-44]-AGGSG-*X. laevis* map2k1.L[32-44]-APVAT-EGFP-SGLRSRA-*S. cerevisiae* Spo20p[51-91]- *S. cerevisiae* Spo20p[51-91] | This study |
| PILS-Nir1 (Nir1-LNS2) | NeonGreen-GGSGGM-(PITPNM3) Nir1[613-897] | This study |
| Nir2-LNS2 | NeonGreen-GGSGGM-(PITPNM1) Nir2[896-1216] | This study |
| Nir3-LNS2 | pcDNA3.1-mEGFP-GGGGSHM-(PITPNM2 isoform 6) Nir3[925-1209] | This study |
| PILS-Nir1-613-630 | (PITPNM3) Nir1[613-630]-GGSGG-NeonGreen | This study |
| PILS-Nir1-631-894 | NeonGreen-GGSGG-(PITPNM3) Nir1[631-894] | This study |
| PILS-Nir1-LNS2-K820E | NeonGreen-GGSGGM-(PITPNM3) Nir1[613-897, K820E] | This study |
| C1ab-Prkd1 | *X. laevis* map2k1.L[32-44]-APVAT-mCherry-SGLRSRAQASNSTS-*M. musculus* Prkd1[138-343] | Kim et al. (2011) |
| FKBP-PJ-dead | mCherry-SGLRSRSAAAGAGGAARAALG-FKBP1A[3-109]-SAGGSAGGSAGGSAGGSAGGPRAQASRSLDA-*S. cerevisiae* Sac1[2-517, C392S]-GGTARGAAAGAGGAGR-INPP5E[214-644, D556A,C641A] | Hammond et al. (2012) |
| FKBP-PJ | mCherry-SGLRSRSAAAGAGGAARAALG-FKBP1A[3-109]-SAGGSAGGSAGGSAGGSAGGPRAQASRSLDA-*S. cerevisiae* Sac1[2-517]-GGTARGAAAGAGGAGR-INPP5E[214-644, C641A] | Hammond et al. (2012) |
| FKBP-INPP5E | mCherry-SGLRSRSAAAGAGGAARAAMG-FKBP1A[3-109]-ARGAAAGAGGAGR-INPP5E[214-644, C641A] | Hammond et al. (2012) |
| FKBP-PJ-Sac1 | mCherry-SGLRSRSAAAGAGGAARAALG-FKBP1A[3-109]-SAGGSAGGSAGGSAGGSAGGPRAQASRSLDA-*S. cerevisiae* Sac1[2-517]-GGTARGAAAGAGGAGR-INPP5E[214-644, D556A C641A] | Hammond et al. (2012) |
| BFP-Tubby(C) | *M. musculus* Tubby protein[243-505]-TVPRARDA-pTagBFP-KRPRL | Quinn et al. (2008) |
| P4Mx1 | *X. laevis* map2k1.L[32-44]-APVAT-mTagBFP2-SGLRSRAQASNSAVDGGSASGLRS- *L. penomophila* SidM[546-647] | Zewe et al. (2020) |
| PM-FRB | Lyn[1-11]-RSANSGAGAGAGAILSR-MTOR[2021-2113]-TSYPYDVPDYAPVAT-iRFP | Hammond et al. (2014) |
| Mito-FRB | iRFP-SGLRSRAGGAGAILSR-MTOR[2021-2113]-GGSGAGGSAQASNSAVDGTA-Fis1[122-152] | Doyle et al. (2024), *Preprint* |
| FKBP-PI4K | mCherry-SGLRSRSAAAGAGGAARAAL-FKBP1A[3-108]-SAGGSAGGSAGGSAGGSAGGPRAQASNSL-PI4KA[1102-2103] | Zewe et al. (2020) |
| FKBP-PIP5K | pTagBFP-SGLRSRSAAAGAGGAARAALG-FKBP1A[3-108]-SAGGSAGGSAGGSAGGSAGGPRAQASNSAVDLQA-PIP5K1C[1-640, D101R, R304D, R445E, K446E] | This study |
| EGFP-PH-PLCδ1 | PLCδ1[1-170]-DPPVAT-EGFP | Várnai and Balla (1998) |
| FKBP-PI-PLC | *B. cereus* PI-PLC[32-329, W78A, W273A]-RILQSTVPMG-FKBP1A[3-108]-RDPPVATM-TagBFP2-SGLRSRSAAATLDHNQPYHICRGFTCFKKPPTPPPEPET | Pemberton et al. (2020) |
| FKBP-DGKa | mRFP-SGLRSRSAAAGAGGAARAAL-FKBP1A[3-108]-SAGGSAGGSAGGSAGGSAGGPRAQASRS-DGKA isoform b[394-773] | This study |
| M3 | pcDNA3.1-HAx3-AchR-CHRM3[2-590] | J. Wess |
| ER-FRB | iRFP713- SGLRSRAQLTMAYPYDVPDYVA-MTOR[2021-2113]-QGSGAGAGAGAILNSRV-SACM1L[418-484] | Zewe et al. (2020) |
| Golgi-FRB | iRFP-SGLRSRAGGAGAILSR-MTOR[2021-2113]-GGSAGGSA-GOLGB1[3096-3224] | Zewe et al. (2020) |
| Rab5-FRB | iRFP713-SGLRSRAGGAGAILSR-MTOR[2021-2113]-GGSAGGSAQASNSAVDGT-*C. lupus* Rab5a[1-215] | Hammond et al. (2012) |
| GFP-Nir2 | EGFP-SGLRSRAQASNS-PITPNM1v2 | Kim et al. (2015) |

Table 2. **Plasmids used in this study (Continued)**

| Shorthand name | Sequence | Reference |
|---|---|---|
| iRFP-PILS-Nir1 | miRFP670-GGSGGM-(PITPNM3) Nir1[613-897] | This study |
| iRFP-Tubby(c) | *M. musculus* Tubby[243-505]-PRARDPPVAT-miRFP670 | Quinn et al. (2008) |
| iRFP-PH-PLCδ1 | iRFP713-CTRDLELKL-*R. norvegicus* PLCD1 isoform X3[2-131] | Idevall-Hagren et al. (2012) |
| TagBFP2-HRAS-CAAX | TagBFP2-SGLRSRAQASNSAVD-HRAS[172-189] | Goulden et al. (2019) |
| mCh-HRAS-CAAX | mCherry-SGLRSRAQASNSAVD-HRAS[172-189] | This study |
| pUC19 | Empty plasmid used to bring total DNA mass up to 1 µg as needed | Yanisch-Perron et al. (1985) |

All genes are *Homo sapiens* unless indicated otherwise. The relevant amino acid positions in the full-length protein are noted. Mutations are described by the position of the residues in the full-length protein. Amino acid sequences written out indicate linkers.

the cell during imaging. However, cells were excluded from analysis if their movement was too large. An additional ROI was drawn in the background of each field.

The background fluorescence was then subtracted from each field, and the resulting intensity within the cell ROI was measured at each time point ($F_t$) and normalized to the intensity within the frames that were taken before stimulation ($F_{pre}$).

### Liposome coflotation assay
Liposomes were prepared in Buffer A as described previously in Methods. Liposomes in Buffer A consisting of 20 mM Tris, pH 7.5, and 150 mM NaCl were incubated for 20 min at room temperature with purified PILS-Nir1 at a final concentration of 1 mM liposomes and 10 µM protein. βME was added to the liposome and protein mixture to 5 mM to prevent unwanted oligomerization. A stock solution of 60% wt/vol Nycodenz in Buffer A was made up, and a gradient reaction was carefully prepared in layers in Beckman Coulter 11 × 34 mm thickwall polycarbonate tubes: 300 µl liposome and protein mixture in 40% Nycodenz solution, 250 µl 30% Nycodenz solution, and 50 µl 0% Nycodenz solution (plain Buffer A). Reactions were ultracentrifuged in a vacuum at 4°C for 4 h at 213,000 *g* (TLS-55 Beckman Coulter rotor at 50,000 rpm). 75 µl samples were carefully taken from the floating top liposome fraction (cloudy appearance), the soluble protein fraction (neither bound to liposomes nor pelleted), and the pellet fraction after it was resuspended in 100 µl Buffer A. Samples were analyzed *via* SDS-PAGE, and gel band intensities were quantified using ImageJ software. Due to the inability to isolate and resuspend the middle fraction in a proportional volume to the liposome and pellet fractions (100 µl) for a gel sample, gel band intensities were multiplied by 6 to determine % protein remaining soluble to account for its 1:6 dilution.

### Circular dichroism
CD spectra of purified PILS-Nir1 were measured on a spectropolarimeter (J-715; Jasco). A final concentration of ~0.1 mg/ml of pure protein was incubated with 0.2 mM liposomes containing 80 mol% POPC and 20 mol% POPA (Avanti Lipids) for 30 min at room temperature prior to

measurement. Liposomes were prepared in plain water as previously described. All samples have the following buffering conditions: 2 mM Tris, pH 7.5, and 15 mM NaCl. CD spectra were measured between 190 and 260 nm in increments of 1 nm, a bandwidth of 1 nm, and an averaging time of 1 min at 25°C. 10 iterations of spectra were averaged and were reported in *m*°, which was converted to molar ellipticity using the following equation:

$$\frac{m° \times 106}{path\ length\ (mm) \times protein\ concentration\ (µM) \times n}$$

where *n* is the number of peptide bonds in the protein, expressed in degree*cm²*dmol$^{-1}$ units.

### Data presentation and statistics
Data analysis, statistics, and graphs were done using GraphPad Prism 9 or later. Details of statistical tests and P values are provided in the figure legends.

### Plasmids and cloning
The plasmids used in this study were obtained from the sources as noted in Table 2 or made by either restriction digest and ligation or PCR and NEBuilder HiFi DNA assembly (E5520S; New England Biolabs). Insert sequences were ordered as custom GeneBlocks from IDT or isolated from existing plasmids. All plasmid sequences were verified over the relevant area by Sanger sequencing or over the full plasmid with long-read nanopore sequencing. Plasmids created in this study are available on Addgene.

### Online supplemental material
Fig. S1 shows PILS-Nir1 does not aggregate or unfold in *in vitro* liposome experiments. Fig. S2 shows PA biosensors can associate with the PM under resting conditions. Table S1 shows P values from the ordinary one-way ANOVA with multiple comparisons for AUC biosensor data presented in Fig. 1 J.

### Data availability
The data underlying Fig. 1, G, H, J, and K; Fig. 2, A–C, E, and F; Fig. 3, A–D; Fig. 4, B–D and F; Fig. 5, A–F; Fig. 6, A–H; Fig. 7, A and B; Fig. S1, A–C; and Fig. S2, B and C are available at https://doi.org/10.5281/zenodo.16782055. Source gel images for Figs. 2 and

S1 are available in the published article and its online supplemental material.

## Acknowledgments

We would like to thank Dr. Shujuan Gao for the initial cloning of the Nir LNS2 domains (Stony Brook University, NY, USA). We are grateful to all members of the Hammond lab for technical assistance with lab maintenance and experiments, especially during the COVID-19 pandemic. A special thanks to Tiernan Swayhoover for her assistance as an undergraduate researcher (University of Pittsburgh, Pittsburgh, PA, USA) and to the Experiments and Logic in Cell Biology class (University of Pittsburgh, Pittsburgh, PA, USA) for their feedback and suggestions.

This work was supported by National Institutes of Health (NIH) grants R35GM119412 (to Gerald R.V. Hammond), R35GM12866 (to Michael V. Airola), and 1F31HL170755-01 (to Claire C. Weckerly). Additional support was given by the Sloan Research Foundation.

Author contributions: Claire C. Weckerly: conceptualization, formal analysis, funding acquisition, investigation, resources, visualization, and writing—original draft, review, and editing. Taylor A. Rahn: conceptualization, formal analysis, investigation, methodology, validation, and writing—review and editing. Max Ehrlich: investigation and visualization. Rachel C. Wills: Formal analysis, investigation, and resources. Joshua G. Pemberton: conceptualization, investigation, methodology, resources, validation, and writing—review and editing. Michael V. Airola: conceptualization, funding acquisition, supervision, and writing—review and editing. Gerald R.V. Hammond: conceptualization, data curation, formal analysis, funding acquisition, methodology, project administration, resources, software, supervision, visualization, and writing—original draft, review, and editing.

Disclosures: The authors declare no competing interests exist.

Submitted: 30 May 2024

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

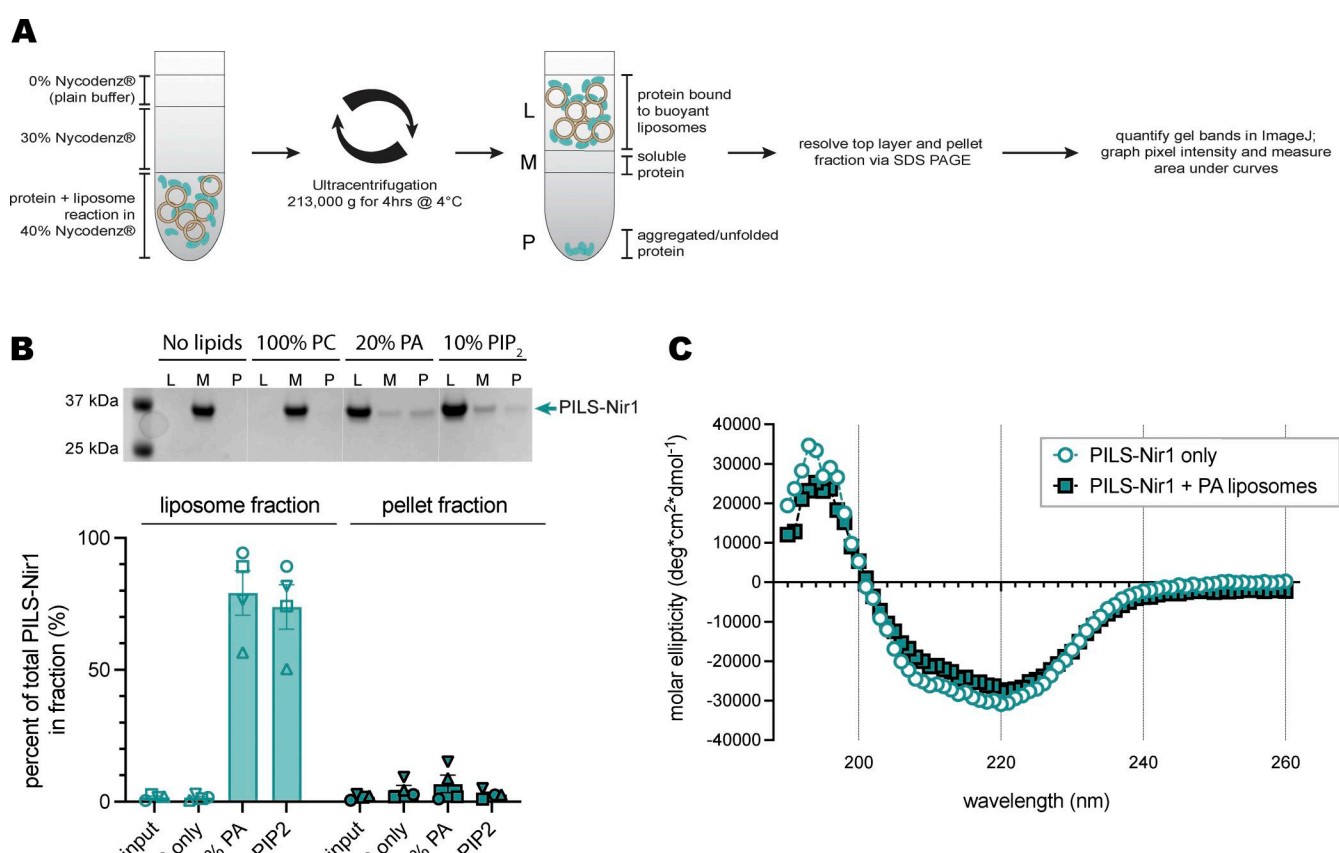

**Figure S1.** **PILS-Nir1 does not aggregate or unfold in *in vitro* liposome experiments. (A)** Schematic of the PILS-Nir1 liposome flotation assay. **(B)** Representative SDS-PAGE gel and quantification of PILS-Nir1 bound to liposomes in the liposome fraction (L), as soluble protein in the middle fraction (M), or as aggregated protein in the pellet fraction (P) after reacting with POPC liposomes of varying compositions. **(C)** CD for PILS-Nir1 with and without PA liposomes.

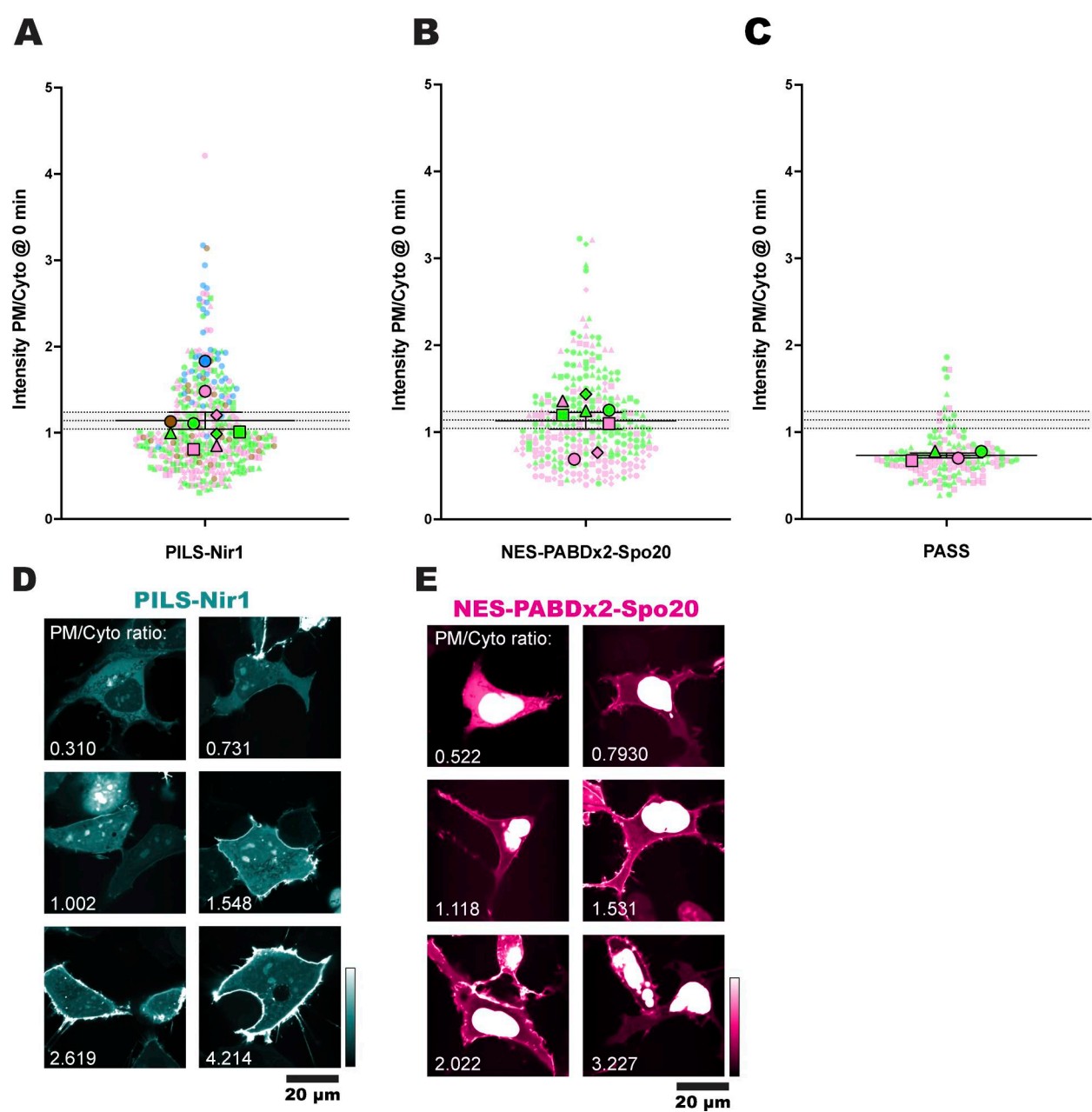

**Figure S2. PA biosensors can associate with the PM under resting conditions. (A–C)** Basal localization of PILS-Nir1 (A), NES-PABDx2-Spo20 (B), and PASS (C). Each small symbol represents the biosensor intensity PM/Cyto ratio of a single cell at time 0 min, before any treatment was added (*n* = 418 PILS-Nir1 cells, 288 NES-PABDx2-Spo20 cells, and 74 PASS cells). The large symbols show the grand means of each experimental replicate (*n* = 3–6 independent experiments). The symbols are color-coded according to the figure where the data can be found. Pink cells are in Fig. 1. Blue cells are in Fig. 2. Green cells are in Fig. 7. Brown cells are cells co-expressing FKBP-PJ-Dead (a catalytically dead PIP phosphatase used as a control in Fig. 3). The shape of the symbol denotes different dishes within each experiment (i.e., cells that were to be treated with PMA or cells that were to be treated with PMA + FIPI). Note that not all treatments shown here were included in their respective figures. Error bars show the mean ± SEM. The gray shaded area shows the PILS-Nir1 grand mean ± SEM to facilitate comparison between graphs. **(D and E)** Representative confocal images of PILS-Nir1 (D) and NES-PABDx2-Spo20 (E) show the range of basal PM/Cyto ratios seen across these experiments, with the given ratio values labeled on each image. Source data are available for this figure: SourceData FS2.

**Provided online is Table S1. Table S1 shows P values from the ordinary one-way ANOVA with multiple comparisons for AUC biosensor data presented in Fig. 1 J.**

