## [Peer Review File · The Journal of Cell Biology]

PILS-Nir1 is a sensitive phosphatidic acid biosensor that reveals mechanisms of lipid production

Claire Weckerly, Taylor Rahn, Max Ehrlich, Rachel Wills, Joshua Pemberton, Michael Airola, and Gerald Hammond

Corresponding Author(s): Gerald Hammond, University of Pittsburgh

Review Timeline:

Submission Date:	2024-05-30
Editorial Decision:	2024-06-04
Revision Received:	2025-07-09
Editorial Decision:	2025-08-03
Revision Received:	2025-08-05

Monitoring Editor: William Prinz

Scientific Editor: Andrea Marat

Transaction Report:

DOI: <https://doi.org/10.1083/jcb.202405174>

Revision 0

Review #1

1. Evidence, reproducibility and clarity:

Evidence, reproducibility and clarity (Required)

****Summary:****

The authors designed, created, and validated a fluorescently tagged sensor protein that binds with high affinity to the signaling phospholipid PA in cells. The LNS2 PA-binding domain used originates from the lipid transfer protein Nir1, and shares conserved features with lipins. The novel sensor outperforms the commonly used Spo20-based PA probes, although it also suffers from binding to PIP2 in vitro (liposomes) and from PIPs affecting its membrane binding in vivo. Importantly, the authors demonstrate that PA but not DAG or PIP2 is sufficient for membrane binding of Nir1-LNS2 in cells, validating Nir1-LNS2 as a PA-sensor in fluorescence microscopy studies.

****Major comments:****

1. CCH treatment of HEK293A cells leads to the PM localization of the DAG sensor C1ab-Prkd1 as well as Nir1-LNS2 (Fig. 5), and the kinetics of these changes - Nir1-LNS2 would lag behind C1ab-Prkd1 fluorescence - is taken as evidence for Nir1-LNS2's specific binding of PA rather than DAG:

Pag. 10: 'When we look at the first 2-minutes after CCh addition, we see that C1ab-Prkd1 moves to the PM much faster than Nir1-LNS2 does (Figure 5D). The delay in Nir1-LNS2 translocation makes sense given DAG is produced first and then converted into PA, again indicating that Nir1-LNS2 is specific for PA.'

Fig. 5 legend: 'The Nir1-LNS2 response to PLC depends on PA and not DAG.(...) (D) Nir1-LNS2 translocation to the PM (data replicated from Figure 5C) lags behind C1ab-Prkd1 (data replicated from Figure 5B) in response to CCh addition.'

The validity of the conclusion from this experiment seems questionable.

The argument relies on the low values of Nir1-LNS2's normalized fluorescence intensity compared to C1ab-Prkd1's, in the first two minutes of stimulation, when DAG is expected to accumulate (Fig. 5D). However, if Nir1-LNS2 would bind DAG, the resulting fluorescence is expected to be low, i.e. compared to the much higher signal resulting from subsequent PA binding. Moreover, in this system, which co-expresses the two sensor proteins, competition in binding may account for the apparent precedence of one sensor over the other. Thus, even if the increase in C1ab-Prkd1 fluorescence would precede Nir1-LNS2 -

following the authors' interpretation - this would not exclude binding of DAG by Nir1-LNS2. Fig. 5D shows the confocal images of cells after 30 sec CCH treatment using the two sensors, next to the respective controls, replicated from Fig. 5B/C. However, in Fig. 5D, different colors are used for Nir1-LNS2 than in Fig. 5C, which makes comparison along the time course difficult.

In conclusion, the data presented in Fig. 5 do not exclude DAG binding by Nir1-LNS2, and modification of the conclusions from this experiment throughout the ms (including the cited sentences) is recommended. Consider removal of Figure 5D.

Despite these considerations, the authors' final conclusion regarding the specificity of Nir1-LNS2 towards PA appears well supported (e.g. by the data presented in Figure 4).

Have the authors considered using the DGK inhibitor R59022 to selectively block the conversion of DAG to PA by DGK? Such an experiment could provide additional evidence for the requirement of DGK activity and consequent PA formation for Nir1-LNS2 membrane localization.

2. P 4 and Fig. 2 mention a 'novel domain structure', responsible for binding of PA and PIP2 (in vitro).

What exactly is this novel domain structure?

Why have the two parts of Nir1-LNS, the AAH and the 263 amino acid domain, not been tested in similar liposome assays as in Fig. 2A? Lipid binding in vivo, as tested in the experiment of Fig. 2D, is confounded by endogenous PA binding proteins.

PA is expected to bind the SIDGS motif, as this is conserved from the Lipin catalytic motif (p. 5). However, experimental evidence of this appears to be lacking. Nevertheless, it is several times presented as a fact in the text.

Pag. 7: 'This suggests that the SIDGS motif alone is not the sole PA binding pocket as the LNS2 domain requires both that motif and the amphipathic helix for sustained binding to membrane embedded PA.'

Pag. 13: '... One reason for this difference could be that the Nir1-LNS2 is not a novel bona fide PA binding pocket. Rather, it requires both the SIDGS motif and an N-terminal amphipathic helix for membrane interactions (Figure 2).

These conclusions therefore appear to be not accurate and would need to be rephrased.

If the authors could show that PA binding by Nir1-LNS can be eliminated by mutating residues in the SIDGS motif, this would not only substantiate the above claims, but also make for a negative control protein, next to Nir1-LNS2. For future applications of Nir1-LNS2 as PA biosensor in other organisms this would be useful.

3. Pag. 7: '...small caveat to consider when using Nir1-LNS2 to study PA, the data also demonstrates that Nir1-LNS2 is not specifically interacting with any of the PM PIPs in

cellular membranes.'

This seems not accurate, since the data in Fig. 3 suggest that PI4P could be involved in membrane localization of Nir1-LNS2. It remains however unresolved whether this is a specific interaction with this PIP.

4. Please note that the presence of PE increases the ionization (and negative charge) of PIP2 (Graber et al., 2012) rather than dilutes the negative charge as stated in the Discussion on p.13. Please revise!

****Minor comments:****

5. P 1 The authors may consider adding a 4th criterium for a lipid biosensor: the sensor should not serve as a sink for the lipid by removing/sequestering it from the active pool, thereby interfering with other interactions/conversions.

6. Nir1 lacks a PITP domain (Fig. 1), yet is referred to as lipid transfer protein: please elaborate/explain.

7. Fig. 2A suggests cooperativity in binding of Nir1-LNS2 to PA-containing liposomes. Please mention/comment! Does binding to PIP2-containing liposomes also exhibit cooperativity?

8. Indicate the concentrations of PC and protein in the legend to Fig. 2 panels A and B. M&M says 2 mM PC, according to the PA-concentrations above panel 2A, this should be 1 mM. Please clarify.

9. In Fig. 2B, PI is missing. Any specific reason?

10. Move Fig. 2C to the Introduction and extend it to illustrate the shared conserved features of Nir1-LNS2 and lipin.

11. P 13. 'While real-time IMPACT does not directly report on PA levels as it does not use the endogenous PLD substrate PC, ...'

It is true that this method doesn't directly report on PA levels, but that is because it uses a click chemistry probe as substrate for PLD's transphosphatidylation reaction. Contrary though to what is stated by the authors, this reaction still uses the endogenous substrate of PLD, PC (Liang et al. 2019; www.pnas.org/cgi/doi/10.1073/pnas.1903949116).

12. Fig. 1K: Control must have been treated with PMA plus vehicle (DMSO); if so, please indicate that vehicle was added.

13. Figure 6C: How is $\Delta F_t/F_{pre}$ defined? Add to legend.

14. P 4: 'The boundaries of the Nir2-LNS2 (Uniprot: O00562) and Nir3-LNS2 (Uniprot: Q9BZ72) were also defined using AlphaFold predictions of the structure of these domains.' How was the extent of each of these domains determined - they are much larger than the previously published sequences of LNS2 domains (Kim et al. 2013; *Embo Rep.* 14:891-899. [doi:10.1038/embor.2013.113](https://doi.org/10.1038/embor.2013.113))?

15. In Fig. 3 legend, specify the starting condition of Nir1-LNS2 binding?? Which fluorescence are we looking at?

16. In legend to Fig. 6 please specify the fluorescent tags used. Have they been shown not to affect protein function?

2. Significance:

Significance (Required)

The development of a new, superior PA-sensor is a significant advance in the fields of lipid signaling and specific lipid-protein interactions, that will benefit research on lipid-mediated cellular signaling and intracellular lipid trafficking.

This reviewer's expertise encompasses lipid metabolism and lipid-protein interactions, not so much fluorescence microscopy.

3. How much time do you estimate the authors will need to complete the suggested revisions:

Estimated time to Complete Revisions (Required)

(Decision Recommendation)

Between 1 and 3 months

No

Review #2

1. Evidence, reproducibility and clarity:

Evidence, reproducibility and clarity (Required)

****Summary:****

The authors reported a new PA-binding probe Nir1-LNS2, which potentially offers advantages over conventional tools with its higher sensitivity for PA. The authors performed

extensive characterization in different cell lines to test the ability of Nir1-LNS2 to selectively bind to PA without disrupting endogenous PA signaling. While the tool is potentially useful as a new PA-binding probe with higher spatiotemporal precision, the data provided in the manuscript are not enough to support their claims and conclusions. Especially, the data do not fully support that the Nir1-LNS2 offers more sensitive and selective binding to PA than conventional PA-binding probes using Spo20.

****Major comments:****

1. Nir1-LNS2 seems to show variable basal localization across different representative images presented in the manuscript. A part of them were justified by the effect of other anionic species by PIP (such as Fig 3 where they co-expressed PIP-degrading enzymes). For example, cells in Fig 1F and those in Fig 4F show quite different basal localization of Nir1-LNS2. Is it due to difference in expression level, cell conditions, or other factors? The significant amount of plasma membrane basal localization seems to indicate that Nir1-LNS2 localization is affected by its binding to PI(4,5)P2. The significant and potentially variable plasma membrane localization of Nir1-LNS1 can limit the utility of this probe.
2. The authors mention high affinity of Nir1-LNS2, but it lacks in vitro characterization that should demonstrate the higher affinity of Nir1-LNS2 compared to conventional probes such as Spo20. The authors should perform side-by-side comparison in Fig 2 to compare the PA affinity and specificity of Nir1-LNS2 compared to Spo20.

****Minor comments:****

1. Fig 1 shows Nir1-LNS2 translocates to plasma membrane upon PMA stimulation in a PLD activity-dependent manner. However, the image in Fig1K is not super convincing since there is already a decent amount of plasma membrane localization of the sensor at t = 0 min, which looks considerably different from the t = 0 min image shown in 1F.
2. Fig 3 and Fig 4 need the validation of PIP depletion/production using PIP-binding probes.
3. In Discussion: "while in vivo it solely binds to PA (Fig 4)" - this claim does not seem to be true according to Fig 4, where the overexpression of PIP-degrading enzymes did affect the Nir1-LNS2 basal localization.

2. Significance:

Significance (Required)

General assessment:

The existing PA-binding probe using Spo20 is indeed quite blunt, which takes minutes to see appreciable accumulation of the probe upon PA production. Nir1-LNS2 can be indeed useful if it offers better spatiotemporal precision. However, the advantage of this tool over existing tools is not convincing without head-to-head comparison of either (1) in vitro characterization of PA binding affinity and selectivity between Nir1-LNS2 and Spo20 or (2) response to PA produced on different subcellular localizations other than plasma membrane and mitochondria (e.g., endosomes, golgi, and endoplasmic reticulum).

Advance:

The key significance of the manuscript, which is the superiority of Nir1-LNS2 over existing PA-binding probes, is not clear from the data provided. Other than that part, the study does not seem to include significant finding, since the binding of Nir1-LNS2 to PA itself is already known (EMBO Rep. 2013 Oct; 14(10): 891-899, Mol Biol Cell. 2022 Mar 1;33(3):br2).

Audience:

The lipid biology community would be highly interested in using the new PA-binding tool to study lipid localization in live cells.

My expertise is PA signaling and development of engineered phospholipase Ds, which can produce PA on demand at various subcellular locations.

3. How much time do you estimate the authors will need to complete the suggested revisions:

Estimated time to Complete Revisions (Required)

(Decision Recommendation)

Between 1 and 3 months

4. Review Commons values the work of reviewers and encourages them to get credit for their work. Select 'Yes' below to register your reviewing activity at Web of Science Reviewer Recognition Service (formerly Publons); note that the content of your review will not be visible on Web of Science.

Yes

Review #3

1. Evidence, reproducibility and clarity:

Evidence, reproducibility and clarity (Required)

Weckerly et al. introduced a fluorescently tagged Nir1-LNS2 construct capable of binding to both PA and PIP2 in vitro, yet selectively targeting PA-enriched membranes in cells. Their findings demonstrate that Nir1-LNS2 exhibits heightened responsiveness to PA, allowing the authors to uncover a modest contribution of PLD to PA production downstream of muscarinic receptors, a phenomenon not visualized with previous Spo20-based biosensors. Thus, Nir1-LNS2 is a sensitive biosensor, potentially providing researchers with a powerful new tool for real-time investigation of PA dynamics in live cells.

The manuscript is well-written, with major conclusions supported by experimental evidence. The tool developed in this study holds significant importance for the field of lipid biology. However, missing controls and weaknesses from the in vitro analysis reduce the overall impact of this work. The authors are encouraged to address the following comments to further strengthen their conclusions:

Major Points:

1. The direct measurement of the binding affinity of Nir1-LNS2 with PA, e.g., K_d , is essential; this information will help the field explore the potential usage of Nir1-LNS2.
2. As mentioned by the authors, there is a confusing inconsistency regarding why Nir1-LNS2 binds to PIP2 in vitro but not in cells. Going beyond what has been discussed in the manuscript, there is a possibility that PIP2 could induce Nir1-LNS2 aggregation, leading to pelleting after centrifugation, among many other possibilities. I recommend the authors perform additional in vitro experiments, including but not limited to the liposome floatation assay to directly examine Nir1-LNS2 binding to the liposomes with varied compositions.
3. In Fig. 2D, it would be beneficial to examine the constructs Nir1-613-630 and Nir1-631-894, comparing them with Nir1-LNS2 using liposome sedimentation and floatation assays to evaluate the contribution of the SIDGS motif and the amphipathic helix in binding PA.
4. Due to PA's versatile biological roles, the evidence provided by the MCS experiment is far from enough to conclude that Nir1-LNS2 does not interfere with PA function. Further examination of various endogenous pathways is warranted before making the statement "Therefore, Nir1-LNS2 can be used without concern of affecting downstream events".

Minor Points:

1. In Fig. 3A-D (Left), it is unclear to what extent PIPs are reduced after treatment with FKBP-tagged PIP phosphatases. The treatment depicted in the illustration should be accompanied by data, e.g., % of PIPs being degraded after treatment.
2. In Fig. 4C, the plasma membrane (PM) localization of Nir1-LNS2 and NES-PABDx2-Spo20, as determined by the "intensity PM/Cyto," should be analyzed following the ectopic production of PI4P and PIP2. Although mitochondria do not apparently recruit Nir1-LNS2 or NES-PABDx2-Spo20 after induced PI4P and PIP2 production, it remains possible that the subsequent trafficking of PI4P and PIP2 from mitochondria might sequester the biosensors away from the PM into the cytoplasm, thereby reducing the "intensity PM/Cyto" of Nir1-LNS2.
3. It would be valuable to determine the half-life (stability) of Nir1-LNS2.

2. Significance:

Significance (Required)

Tracking intracellular phosphatidic acid (PA) in live cells is essential for understanding its cellular functions, leading to the development of genetically encoded lipid biosensors. While several PA biosensors have been developed, they often suffer from limited sensitivity or specificity.

3. How much time do you estimate the authors will need to complete the suggested revisions:

Estimated time to Complete Revisions (Required)

(Decision Recommendation)

Between 3 and 6 months

4. Review Commons values the work of reviewers and encourages them to get credit for their work. Select 'Yes' below to register your reviewing activity at Web of Science Reviewer Recognition Service (formerly Publons); note that the content of your review will not be visible on Web of Science.

Yes

Revision Plan

Manuscript number: RC-2024-02413R
Corresponding author(s): Hammond, Gerald

1. General Statements [optional]

*We are grateful to the three reviewers for such thorough and thoughtful comments. Data or re-writes that we have on hand that address many of these comments have been incorporated already. We also have a comprehensive experimental plan to address all of the remaining major comments. Reviewer’s comments are in light italics, whilst our responses appear in regular font below. We added reviewer numbering for ease of cross-reference to the original comments, with the format: reviewer X’s comment number N as **#X.N***

Overall, we were thrilled that the reviewers agreed that our work is of significance and broad interest:

*“The development of a new, superior PA-sensor is a significant advance in the fields of lipid signaling and specific lipid-protein interactions, that will benefit research on lipid-mediated cellular signaling and intracellular lipid trafficking.” – **reviewer 1.***

*“The lipid biology community would be highly interested in using the new PA-binding tool to study lipid localization in live cells.” – **reviewer 2.***

*“Tracking intracellular phosphatidic acid (PA) in live cells is essential for understanding its cellular functions, leading to the development of genetically encoded lipid biosensors. While several PA biosensors have been developed, they often suffer from limited sensitivity or specificity.” – **reviewer 3***

Table of Contents

- 1. General Statements [optional]..... 1**
- 2. Description of the planned revisions 2**
- 3. Description of the revisions that have already been incorporated in the transferred manuscript 5**
- 4. Description of analyses that authors prefer not to carry out..... 15**

2. Description of the planned revisions

#1.2: P 4 and Fig. 2 mention a 'novel domain structure', responsible for binding of PA and PIP2 (in vitro). What exactly is this novel domain structure? Why have the two parts of Nir1-LNS, the AAH and the 263 amino acid domain, not been tested in similar liposome assays as in Fig. 2A? Lipid binding in vivo, as tested in the experiment of Fig. 2D, is confounded by endogenous PA binding proteins. PA is expected to bind the SIDGS motif, as this is conserved from the Lipin catalytic motif (p. 5). However, experimental evidence of this appears to be lacking. Nevertheless, it is several times presented as a fact in the text. Pag. 7: 'This suggests that the SIDGS motif alone is not the sole PA binding pocket as the LNS2 domain requires both that motif and the amphipathic helix for sustained binding to membrane embedded PA.' Pag. 13: '... One reason for this difference could be that the Nir1-LNS2 is not a novel bona fide PA binding pocket. Rather, it requires both the SIDGS motif and an N-terminal amphipathic helix for membrane interactions (Figure 2). These conclusions therefore appear to be not accurate and would need to be rephrased. If the authors could show that PA binding by Nir1-LNS can be eliminated by mutating residues in the SIDGS motif, this would not only substantiate the above claims, but also make for a negative control protein, next to Nir1-LNS2. For future applications of Nir1-LNS2 as PA biosensor in other organisms this would be useful.

In addition to the already included data on cellular binding of the R784E mutant, we do plan to test this variant in the liposome binding assays to show loss of PA binding abilities as the reviewer has suggested. We also plan to evaluate proper folding of the R784 mutant through circular dichroism.

#1.7. Fig. 2A suggests cooperativity in binding of Nir1-LNS2 to PA-containing liposomes. Please mention/comment! Does binding to PIP2-containing liposomes also exhibit cooperativity?

Using a nonlinear fit, we were able to determine Hill coefficients for PA binding. This has now been included in **Figure 2B (formerly Figure 2A)** and the following text has been added to **page 6, left column, third paragraph**: “In addition, we found that Nir1-LNS2 bound PA-rich liposomes in a concentration dependent manner (Figure 2B). By fitting the binding curve to this data, we found that the interaction of Nir1-LNS2 with PA provided a K_d value of ~19 mol%. Interestingly, Nir1-LNS2 binds to PA in a highly cooperative manner. The Hill coefficient for the interaction of Nir1-LNS2 with PA was calculated to be approximately 4 (Figure 2B).”

However, due to the liposome binding assay used that utilizes a set total lipid concentration but alters mol% lipids, the K_d that we determined is not a “traditional” K_d . Therefore, we plan to repeat this assay using constant PA concentrations but increasing total concentrations of lipid so that we can make a better fit and get a more accurate K_d value and Hill coefficient. We also plan to do the same assay with PIP2 to determine K_d values and Hill coefficients for that interaction.

#2.2. The authors mention high affinity of Nir1-LNS2, but it lacks *in vitro* characterization that should demonstrate the higher affinity of Nir1-LNS2 compared to conventional probes such as Spo20. The authors should perform side-by-side comparison in Fig 2 to compare the PA affinity and specificity of Nir1-LNS2 compared to Spo20.

We plan to take the reviewer’s advice and directly compare Spo20-PABDx2 (and/or the single PABD depending on what we can get to purify correctly) and Nir1-LNS2 in the liposome binding assay.

Additionally, we propose to further characterize these sensors in cells as well. To start, we have added a direct comparison of the Spo20-PABDx2 and Nir1-LNS2 response to PA production at the PM (by PMA stimulation) and at mitochondria (by FKBP-DGKa) in **Figure 4**. The text has been updated to reflect this on **p.10, left paragraph, 3rd paragraph**: “*Importantly, we also observed that Nir1-LNS2 responds to this ectopic PA production quicker and more robustly than NES-PABDx2-Spo20 does, as can be seen when the responses from Figure 4F are plotted together (Figure 4H). When analyzing the responses to PA production at the PM by PMA stimulation in Figure 1D and Figure 1F, we similarly see that the Nir1-LNS2 translocates to the PM more robustly and in a shorter timeframe (Figure 4G). This suggests that the Nir1-LNS2 can serve as a high affinity PA biosensor at various cellular locations.*”

Furthermore, as suggested in the “General Assessment” we propose to use the FKBP-DGKa system to produce PA on other organelles such as the Golgi and ER and then we can directly compare the response of Spo20-PABDx2 and Nir1-LNS2 to the increase in PA at these organelles. This data will be added to Figure 4 for a full comparison of the sensors across cellular locations.

#2.4. Fig 3 and Fig 4 need the validation of PIP depletion/production using PIP-binding probes.

We propose to repeat the experiments using TIRF in figure 3 as it will give us increases sensitivity, and also compare selectivity with the currently used spo20-based biosensors.

#2. “General assessment”: *The existing PA-binding probe using Spo20 is indeed quite blunt, which takes minutes to see appreciable accumulation of the probe upon PA production. Nir1-LNS2 can be indeed useful if it offers better spatiotemporal precision. However, the advantage of this tool over existing tools is not convincing without head-to-head comparison of either (1) in vitro characterization of PA binding affinity and selectivity between Nir1-LNS2 and Spo20 or (2) response to PA produced on different subcellular localizations other than plasma membrane and mitochondria (e.g., endosomes, golgi, and endoplasmic reticulum).*

In order to address the selectivity of Nir1-LNS2 and Spo20, we propose to repeat the experiments in Figure 3 with the PJ enzymes in order to see how the PM PIPs affect Spo20 membrane binding, as described in our response to **#2.4**. Previously published data, as well as our own unpublished observations suggest that Spo20 interacts with the anionic PIPs to a greater extent than Nir1-LNS2 does (Nakanishi et al., 2004 doi: 10.1091/mbc.e03-11-0798; Horchani et al., 2014 doi: 10.1371/journal.pone.0113484). If we can show that Spo20’s interactions with the PM are significantly influenced by the PIPs, then this will add more evidence to the idea that Nir1-LNS2 is more selective for PA.

As described in response to **#2.2**, we are also planning a side-by-side comparison of spo20 based protein binding on liposomes alongside Nir1-LNS2.

Also, as discussed above, we agree with the reviewer that looking at the Nir1-LNS2 and Spo20 responses to PA production at other organelles would increase confidence that Nir1-LNS2 has a higher affinity for PA. **We propose to add these experiments to Figure 4.**

#3.1. *The direct measurement of the binding affinity of Nir1-LNS2 with PA, e.g., K_d , is essential; this information will help the field explore the potential usage of Nir1-LNS2.*

Using a nonlinear fit, we were able to determine Hill coefficients for PA binding. This has now been included in **Figure 2B (formerly Figure 2A)** and the following text has been added to **p.6, left column, 3rd paragraph**: *“In addition, we found that Nir1-LNS2 bound PA-rich liposomes in a concentration dependent manner (Figure 2B). By fitting a nonlinear curve to this data, we found that the interaction of Nir1-LNS2 with PA provided a K_d value of ~19 mol%. Interestingly, Nir1-LNS2 binds to PA in a highly cooperative manner. The Hill coefficient for the interaction of Nir1-LNS2 with PA was calculated to be 4.323 (Figure 2B).... This suggests that the amphipathic helix and the SIDGS-containing domain may both interact with the membrane leading to the cooperative nature of Nir1-LNS2’s binding of PA-rich liposomes (Figure 2B).”*

However, due to the liposome binding assay used that utilizes a set total lipid concentration but alters mol% lipids, the K_d that we determined is not a “traditional” K_d . Therefore, we plan to repeat this assay using higher total lipid concentrations with a fixed PA mol% so that we can make a better fit and get a more accurate K_d value and Hill coefficient. Furthermore, we plan to directly compare Spo20-PABDx2 (and/or the single PABD depending on what we can get to purify correctly) and Nir1-LNS2 in the liposome binding assay to directly compare their affinities for PA in vitro, as described in response to **#2.2**.

#3.2. *As mentioned by the authors, there is a confusing inconsistency regarding why Nir1-LNS2 binds to PIP2 in vitro but not in cells. Going beyond what has been discussed in the manuscript,*

there is a possibility that PIP2 could induce Nir1-LNS2 aggregation, leading to pelleting after centrifugation, among many other possibilities. I recommend the authors perform additional in vitro experiments, including but not limited to the liposome floatation assay to directly examine Nir1-LNS2 binding to the liposomes with varied compositions.

This is an excellent suggestion. We plan to check for aggregation by liposome flotation with Nir1-LNS2 in the presence of high mol% of PA and PIP₂. In addition, we will also perform circular dichroism to see if PA or PIP2 liposomes are inducing any unfolding of Nir1-LNS2.

#3.3. *In Fig. 2D, it would be beneficial to examine the constructs Nir1-613-630 and Nir1-631-894, comparing them with Nir1-LNS2 using liposome sedimentation and floatation assays to evaluate the contribution of the SIDGS motif and the amphipathic helix in binding PA.*

Per our response to **#1.2**, we looked around the SIDGS motif to find the residue that would mediate the binding of membrane embedded PA, which our data suggests is R784 (Figure 2D). We do plan to test the R784 mutant in the liposome binding assays to show loss of PA binding abilities as the reviewer has suggested. We also plan to evaluate proper folding of the R784 mutant through circular dichroism.

3. Description of the revisions that have already been incorporated in the transferred manuscript

#1.1: *CCH treatment of HEK293A cells leads to the PM localization of the DAG sensor C1ab-Prkd1 as well as Nir1-LNS2 (Fig. 5), and the kinetics of these changes - Nir1-LNS2 would lag behind C1ab-Prkd1 fluorescence - is taken as evidence for Nir1-LNS2's specific binding of PA rather than DAG: Pag. 10: 'When we look at the first 2-minutes after CCh addition, we see that C1ab-Prkd1 moves to the PM much faster than Nir1-LNS2 does (Figure 5D). The delay in Nir1-LNS2 translocation makes sense given DAG is produced first and then converted into PA, again indicating that Nir1-LNS2 is specific for PA. 'Fig. 5 legend: 'The Nir1-LNS2 response to PLC depends on PA and not DAG.(...) (D) Nir1-LNS2 translocation to the PM (data replicated from Figure 5C) lags behind C1ab-Prkd1 (data replicated from Figure 5B) in response to CCh addition. 'The validity of the conclusion from this experiment seems questionable. The argument relies on the low values of Nir1-LNS2's normalized fluorescence intensity compared to C1ab-Prkd1's, in the first two minutes of stimulation, when DAG is expected to accumulate (Fig. 5D). However, if Nir1-LNS2 would bind DAG, the resulting fluorescence is expected to be low, i.e. compared to the much higher signal resulting from subsequent PA binding. Moreover, in this system, which co-expresses the two sensor proteins, competition in binding may account for the apparent precedence of one sensor over the other. Thus, even if the increase in C1ab-Prkd1 fluorescence would precede Nir1-LNS2 - following the authors' interpretation - this would not exclude binding of DAG by Nir1-LNS2. Fig. 5D shows the confocal images of cells after 30 sec CCH treatment using the two sensors, next to the respective controls, replicated from Fig. 5B/C. However, in Fig. 5D, different colors are used for Nir1-LNS2 than in Fig. 5C, which makes comparison along the time course difficult. In conclusion, the data presented in Fig. 5 do not exclude DAG binding by Nir1-LNS2, and*

Revision Plan

modification of the conclusions from this experiment throughout the ms (including the cited sentences) is recommended. Consider removal of Figure 5D. Despite these considerations, the authors' final conclusion regarding the specificity of Nir1-LNS2 towards PA appears well supported (e.g. by the data presented in Figure 4).

We agree with the reviewer that the interpretation of the kinetic data is ambiguous and does not fully negate the idea that Nir1-LNS2 may bind to DAG. We have modified the interpretation accordingly. However, we have left the kinetic comparison of the DAG vs Nir1-LNS2 biosensors since these reflect the expected dynamics of the two lipids downstream of PLC. The data are now interpreted as follows on **p. 10, right column, second paragraph**: “*The PM accumulation lagged that of DAG, consistent with conversion of DAG to PA by DGKs (Figure. 5D). Alternatively, in cells treated with CCh and then atropine, Nir1-LNS2 localized to the PM after CCh was added but was then observed returning to the cytoplasm over the 15-minute treatment with atropine as PA levels declined (Figure 5C). Overall, this experiment shows that Nir1-LNS2 binding to the PM follows the expected kinetic profile of DGK-produced PA.*” Likewise, the **legend for figure 5** is now labelled “*Nir1-LNS2 detects PLC stimulated PA production.*” to remove explicit conclusions about PA vs DAG binding.

#1.2: *P 4 and Fig. 2 mention a 'novel domain structure', responsible for binding of PA and PIP2 (in vitro). What exactly is this novel domain structure? Why have the two parts of Nir1-LNS, the AAH and the 263 amino acid domain, not been tested in similar liposome assays as in Fig. 2A? Lipid binding in vivo, as tested in the experiment of Fig. 2D, is confounded by endogenous PA binding proteins. PA is expected to bind the SIDGS motif, as this is conserved from the Lipin catalytic motif (p. 5). However, experimental evidence of this appears to be lacking. Nevertheless, it is several times presented as a fact in the text. Pag. 7: 'This suggests that the SIDGS motif alone is not the sole PA binding pocket as the LNS2 domain requires both that motif and the amphipathic helix for sustained binding to membrane embedded PA.' Pag. 13: '... One reason for this difference could be that the Nir1-LNS2 is not a novel bona fide PA binding pocket. Rather, it requires both the SIDGS motif and an N-terminal amphipathic helix for membrane interactions (Figure 2). These conclusions therefore appear to be not accurate and would need to be rephrased. If the authors could show that PA binding by Nir1-LNS can be eliminated by mutating residues in the SIDGS motif, this would not only substantiate the above claims, but also make for a negative control protein, next to Nir1-LNS2. For future applications of Nir1-LNS2 as PA biosensor in other organisms this would be useful.*

We have clarified that the domain architecture of the Nir1-LNS2 is not a novel domain structure generally, but novel for a PA binding protein which are typically just helices such as that seen in Spo20. **Figure 2** is now titled “*Nir1-LNS2 shows specificity for PA and PIP2 in vitro, based on a novel PA-binding domain.*”

We have also clarified that the SIDGS motif is not the actual location of PA binding, but rather is only the motif conserved with the Lipin/Pah active site. R784 appears to be a PA coordinating residue near the SIDGS, as the positive residue can interact with the negative lipid. Furthermore, we agreed with the reviewer that mutating this residue to perturb PA binding was a much more convincing experiment. We have now included this data in Figure 2 and rewritten the following passages.

Revision Plan

From Page 6, left column, last paragraph: "The putative Lipin catalytic motif DxDxT is partially conserved in

Nir1-LNS2 as a SIDGS motif spanning residues 742-746. We looked for positively charged residues nearby that could bind to the PA in the membrane and coordinate its entrance into the SIDGS site. The active site of the Lipins has a nearby Arg residue which was predicted to perform this role (Khayyo et al., 2020). AlphaFold

(C) Representations of the AlphaFold predicted domain architecture of Nir1-LNS2. It includes an amphipathic alpha helix spanning residues 613-630 and a large, structured domain at residues 631-894, that contains the SIDGS motif that is conserved with the Lipin/Pah active site. The nearby R784 residue is predicted to bind PA at the membrane interface. (D) Isolating either of these domains or introducing a R784E mutation destroys the ability of Nir1-LNS2 to respond to 100 nM PMA in HEK293A cells. The graph shows the grand means \pm SEM of 3-4 experiments (35-44 total cells).

analysis of Nir1-LNS2 showed that this residue was also conserved in Nir1-LNS2 as R784, and that the side chain of the Arg sticks out toward the membrane interface where it would be able to contact the negatively charged PA (Figure 2C).

The conservation of these features between the Lipins and Nir1-LNS2 suggests that PA binds this positively charged residue near the SIDGS pocket within Nir1-LNS2 (Kim et al., 2013; Khayyo et al., 2020). However, for efficient catalytic activity, the Lipins also require an N-terminal amphipathic helix for membrane interaction. This helix is made up of residues 1-18 in *Tetrahymena thermophila* Pah2 (Khayyo et al., 2020), and residues 613-630 in the N-terminus of Nir1-LNS2 are predicted to form a similar amphipathic helix (Figure 2C). We therefore tested whether the N-terminal helix of Nir1-LNS2 was necessary for interaction with PA at the PM. We made two truncations of the Nir1-LNS2 construct: Nir1-613-630 is the isolated amphipathic helix, while Nir1-631-894 is the rest of the domain excluding the helix but including the SIDGS motif. Surprisingly, neither truncated construct responded to PMA by binding the PM, and they even showed reduced basal PM localization (Figure 2D).

Although Figure 2D suggests that the SIDGS motif alone is not sufficient for membrane interactions, we probed into the suspected PA binding residue R784 by mutating it into a negatively charged Glu residue, which should disrupt its interaction with the negatively charged lipid. The R784E mutation completely ablated Nir1-LNS2 interactions at the PM after PMA stimulation and showed reduced association with the PM even before PMA stimulation (Figure 2D).

Altogether, our data suggests that the LNS2 domain requires both the larger SIDGS-containing domain and the amphipathic helix for sustained binding to membrane-embedded PA, but that the PA may directly interact with R784 near the SIDGS motif. Therefore, the Nir1-LNS2 provides

a novel PA binding domain with a tertiary structure beyond the simple amphipathic helices found in Spo20.”

We have also rewritten this sentence in the discussion, **p. 14, right column, second paragraph**: *“As far as the use of Nir1-LNS2 as a biosensor, the one caveat is the discrepancy in its specificity: in vitro PA and PIP2 were sufficient to recruit Nir1-LNS2 to PC liposomes (Figure 2), but in vivo only PA was sufficient for mitochondrial recruitment (Figure 4). One reason for this difference could be that the Nir1-LNS2 requires R784 near the SIDGS pocket and an N-terminal amphipathic helix for membrane interactions (Figure 2).”*

#1.3. *Pag. 7: ‘...small caveat to consider when using Nir1-LNS2 to study PA, the data also demonstrates that Nir1-LNS2 is not specifically interacting with any of the PM PIPs in cellular membranes. This seems not accurate, since the data in Fig. 3 suggest that PI4P could be involved in membrane localization of Nir1-LNS2. It remains however unresolved whether this is a specific interaction with this PIP.*

We have rewritten the text on **Page 8, right column, first paragraph** accordingly: *“This data suggests that decreasing the anionic charge of the membrane through depletion of PIPs slightly reduces Nir1-LNS2’s ability to interact with the PM, but it doesn’t fully re-localize the sensor. Therefore, this is a caveat to consider when using Nir1-LNS2 to study PA.”*

#1.4. *Please note that the presence of PE increases the ionization (and negative charge) of PIP2 (Graber et al., 2012) rather than dilutes the negative charge as stated in the Discussion on p. 13. Please revise!*

We have updated the text on **Page 14, right column, last paragraph**: *“The presence of other lipids such as PI, the formation of PIP₂-rich domains, and even interactions with neighboring proteins can increase hydrogen bonding of PIP₂ and dilute the negative charge (Graber et al., 2012; Borges-Araújo and Fernandes, 2020). Phosphatidylethanolamine on the other hand, increases PIP₂ ionization and its negative charge, though these effects are also thought to be reduced by PIP₂ intramolecular hydrogen bonding which competes for the charges on the lipid (Graber et al., 2012).”*

#1.5. *P 1 The authors may consider adding a 4th criterium for a lipid biosensor: the sensor should not serve as a sink for the lipid by removing/sequestering it from the active pool, thereby interfering with other interactions/conversions.*

We agree that biosensors should not sequester a significant fraction of their cognate lipids and affect downstream pathways by competing with endogenous binding partners. We have rewritten the following text regarding Figure 6 to make this distinction more clear:

Page 13, left column, last paragraph: *“As Nir1-LNS2 shows high affinity for PA across cell lines, this brings up the concern that use of Nir1-LNS2 will sequester PA and inhibit endogenous signaling pathways that depend on PA... Therefore, we conclude that use of Nir1-LNS2 as a PA biosensor does not sequester significant amounts of PA. It is suggested that cellular homeostasis may compensate for the amount of bound lipid by increasing synthesis of free lipid,*

Revision Plan

as this has been seen with the PIP2 biosensor PH-PLCd1 (Traynor-Kaplan et al., 2017). While PA has a plethora of cellular functions, the fact that Nir1-LNS2 expression does not disrupt MCS formation shows promise that the high affinity of Nir1-LNS2 will not inhibit downstream PA signaling.

#1.6. Nir1 lacks a PITP domain (Fig. 1), yet is referred to as lipid transfer protein: please elaborate/explain.

The following text has been added to **Page 2, left column, last paragraph** to clarify this point: “This family of proteins, made up of Nir1, Nir2, and Nir3, form ER-PM membrane contact sites (MCS) to exchange PA and phosphatidylinositol (PI) between the compartments (Cockcroft and Raghu, 2016; Kim et al., 2015). While Nir1 lacks a functional PITP domain, it was initially classified as part of the PITP family based on the homology of its other domains with Nir2 and Nir3. Furthermore, Nir1 has a role in lipid transfer by facilitating Nir2 recruitment to the MCS (Quintanilla et al., 2022).”

#1.8. Indicate the concentrations of PC and protein in the legend to Fig. 2 panels A and B. M&M says 2 mM PC, according to the PA-concentrations above panel 2A, this should be 1 mM. Please clarify.

We have corrected the typo in panel 2A (now panel 2B) and have updated the **Figure 2 legend** as follows, “(A) A representative SDS-PAGE gel is shown for Nir1-LNS2 binding to various PM lipids in POPC liposomes. (B) A representative SDS-PAGE gel is shown for Nir1-LNS2 binding of increasing PA molar concentrations in POPC liposomes. For both A and B, the lipids indicated were mixed with POPC to produce a 2 mM solution, then 50 uL of the resulting liposome mixture was incubated with 50 uL of Nir1-LNS2 at ~1 mg/mL. Supernatant (S) and pellet (P) lanes were quantified using ImageJ to determine percent protein bound. The protein-only control pellet was used as a baseline (input). Nir1-LNS2 appears on the gel at 37 kDa.”

#1.9. In Fig. 2B, PI is missing. Any specific reason?

We have updated the text on **Page 6, left column, second paragraph** to discuss the low levels of PI at the PM, which is why we did not include this lipid. “Using this same PC background, we tested the efficacy of the PM lipids DAG, PA, PS, PI4P and PIP₂ in recruiting Nir1-LNS2 to membranes. While PI serves as a substrate for PI4P and PIP₂ synthesis (collectively referred to as the phosphatidylinositol phosphates (PIPs)) at the PM, levels of PI at the PM are very low compared to the PIPs and therefore PI itself was not tested (Zewe et al., 2020; Pemberton et al., 2020).”

#1.10. Move Fig. 2C to the Introduction and extend it to illustrate the shared conserved features of Nir1-LNS2 and lipin.

We would like to keep the diagram of the Nir1-LNS2 in Figure 2 where the features are discussed in more detail than in the introduction. However, we did add this sentence to the introduction **on p.2, right column, second paragraph** that refers the reader to the cartoon in Figure 2. *“These features are conserved in the Nir LNS2 domains, except for the catalytic Asp in the DxTxT motif and another Mg²⁺-coordinating residue (Figure 2C).”*

#1.11. P 13. *“While real-time IMPACT does not directly report on PA levels as it does not use the endogenous PLD substrate PC, ...It is true that this method doesn't directly report on PA levels, but that is because it uses a click chemistry probe as substrate for PLD's transphosphatidylation reaction. Contrary though to what is stated by the authors, this reaction still uses the endogenous substrate of PLD, PC (Liang et al. 2019; www.pnas.org/cgi/doi/10.1073/pnas.1903949116).*

We have rewritten the aforementioned sentence in the discussion (**Page 15, right column, second paragraph**): *“While real-time IMPACT does not directly report on PA levels as it creates a unique fluorescent lipid, it offers several advantages such as being able to interrogate lipid trafficking over time.”*

#1.12. Fig. 1K: Control must have been treated with PMA plus vehicle (DMSO); if so, please indicate that vehicle was added.

Figure 1K and its legend have been updated. The legend now reads *“Stimulating HEK293A cells with 100 nM PMA and 750 nM of the PLD inhibitor FIPI reduces the Nir1-LNS2 response to PMA and cell media (Veh)”*

#1.13. Figure 6C: How is $\Delta Ft/Fpre$ defined? Add to legend.

We have updated the **Figure 6 legend** to read *“MCS formation was quantified as the change in fluorescence at a given time (Ft) divided by the fluorescence before CCh stimulation (Fpre).”*

#1.14. P 4: *‘The boundaries of the Nir2-LNS2 (Uniprot: O00562) and Nir3-LNS2 (Uniprot: Q9BZ72) were also defined using AlphaFold predictions of the structure of these domains.’ How was the extent of each of these domains determined - they are much larger than the previously published sequences of LNS2 domains (Kim et al. 2013; Embo Rep. 14:891-899. doi:10.1038/embor.2013.113)?*

We have clarified the definition of boundaries by updating the following sentence on **Page 4, right column, 5th paragraph**: *“The boundaries of the Nir2-LNS2 (Uniprot: O00562) and Nir3-LNS2 (Uniprot: Q9BZ72) were also defined using AlphaFold predictions of the structure of these domains. Previous definitions of the Nir2-LNS2 domain have considered the domain smaller than we do here (Kim et al., 2013, 2015) . However, according to AlphaFold, the boundaries set previously exclude a large N-terminal beta barrel that is conserved in the Lipin/Pah PAPs, as*

well as disrupt the domain fold that is homologous to the Lipin active site. Therefore, we are confident that our constructs include the entire LNS2 fold.”

#1.15. *In Fig. 3 legend, specify the starting condition of Nir1-LNS2 binding?? Which fluorescence are we looking at?*

This figure has now been revised in response to **point #3.5**, which hopefully also clarified this point.

#1.16. *In legend to Fig. 6 please specify the fluorescent tags used. Have they been shown not to affect protein function?*

We have updated the figure legend to specify that GFP-Nir2 was used in conjunction with iRFP-Nir1-LNS2, we also changed the text on **Page 13, right column, second paragraph** that refers to this experiment. It now reads *“We co-expressed a GFP-tagged Nir2 and either iRFP-Nir1-LNS2 or iRFP-TubbyC, a PIP2 biosensor that is not expected to affect MCS formation. It should be noted that although we have used the NG-tagged Nir1-LNS2 the most extensively, the iRFP and mCherry-tagged biosensors have behaved the same as the NG-tagged version in the experiments where we utilized them.”*

#2.1. *Nir1-LNS2 seems to show variable basal localization across different representative images presented in the manuscript. A part of them were justified by the effect of other anionic species by PIP (such as Fig 3 where they co-expressed PIP-degrading enzymes). For example, cells in Fig 1F and those in Fig 4F show quite different basal localization of Nir1-LNS2. Is it due to difference in expression level, cell conditions, or other factors The significant amount of plasma membrane basal localization seems to indicate that Nir1-LNS2 localization is affected by its binding to PI(4,5)P2.? The significant and potentially variable plasma membrane localization of Nir1-LNS1 can limit the utility of this probe.*

We have added **Supplemental Figure 1** to show the range of Nir1-LNS2 basal localization compared to NES-PABDx2-Spo20 and PASS. We believe that this localization is due to variable amounts of basal PA combined with some non-selective anionic interactions at the PM. The following paragraph has been added to **page 7, left column, first paragraph** to discuss this point, *“Since the R784E mutant showed reduced basal PM localization, we wanted to further characterize the basal localization of the wild-type Nir1-LNS2. The basal localization of wild-type Nir1-LNS2 varies somewhat between cells, but analysis of all of the cells used throughout this study determines that the basal PM/Cyt ratio of the wild-type Nir1-LNS2 is 1.0644 ± 0.0672 , which suggests that at resting conditions Nir1-LNS2 is slightly enriched at the PM (Supplemental Figure 1A, 1D). When we did the same analysis for all the cells where we expressed NES-PABDx2-Spo20 or PASS, we obtained a basal PM/Cyt ratio of 1.1318 ± 0.0954 for NES-PABDx2-Spo20 and a ratio of 0.6861 ± 0.0143 for PASS (Supplemental Figure 1B, 1C, 1E). We believe that the basal localization of these sensors reflects variable PA levels in the PM at resting conditions. FRET based imaging of PA has indicated that there are detectable levels of PA under basal conditions, and this approach also showed some variability in basal PA levels*

Revision Plan

as we see with the spread of Nir1-LNS2's basal localization (Nishioka et al., 2010). Overall, our data suggests that the high affinity of Nir1-LNS2 for PA is reflected in both its basal localization and its response to stimulations such as PMA."

To address the idea that PIP₂ is responsible for the basal localization of Nir1-LNS, we have added the following to the discussion on **p.15, left column, second paragraph**: "Aside from concerns about specificity, the ability of Nir1-LNS2 to interact with PIP₂ in liposomes could suggest that the basal PM localization of Nir1-LNS2 is due to it binding PIP₂. However, selective depletion of PI(4,5)P₂ did not affect basal Nir1-LNS2 localization to the PM (Figure 3C) and was not able to recruit the probe to mitochondria (Figure 4A-C). We did see FKBP-PJ reduce the association of Nir1-LNS2 with the PM under resting conditions (Figure 3E, 3F), suggesting a possible non-specific ionic interaction with polyanionic inositol lipids. Another mechanism to explain these data would be phosphoinositide-dependence of PA production. Phosphoinositides are well-known to regulate the recruitment of PLD isoforms and type II DGKs to the PM as well as their catalytic activity there (Sciorra et al., 2002; Du et al., 2003; Hodgkin et al., 2000; Liscovitch et al., 1994; Kume et al., 2016). Therefore, we suggest that the effects of FKBP-PJ could be reducing basal PLD and DGK activity and hence lowering resting PA levels. That could explain the loss of both basal Nir1-LNS2 PM association when FKBP-PJ is expressed, and Nir1-LNS2's PM interactions as FKBP-PJ is recruited to the membrane to further deplete phosphoinositides. While this study cannot fully substantiate this hypothesis, the role of PIP₂ in PLD activity and PA production is an interesting hypothesis that warrants further investigation.

Supplemental Figure 1. PA biosensors can associate with the PM under resting conditions. Basal localization of Nir1-LNS2 (A), NES-PABDx2-Spo20 (B), and PASS (C). Each small symbol represents the biosensor intensity PM/cyt of a single cell at time 0 min, before any treatment was added (n = 357 Nir1-LNS2 cells, 288 NES-PABDx2-Spo20 cells, and 74 PASS cells). The large symbols show the grand means of each experimental replicate (n = 3-6 independent experiments). The symbols are color coded according to the figure where the data can be found. Pink are cells in Figure 1. Green are cells in Figure 7. Brown are cells in Figure 3. The shape of the symbol denotes different dishes within each experiment (i.e cells that were to be treated with PMA or cells that were to be treated with PMA + FIPI). Note that not all treatments shown here were included in their respective figures. Error bars show the mean ± SEM. The gray shaded area shows the Nir1-LNS2 mean ± SEM for the experimental replicates (n = 3-6) to facilitate comparison between graphs. Representative confocal images of Nir1-LNS2 (D) and NES-PABDx2-Spo20 (E) show a range of basal PM/cyt ratios seen across these experiments, with the given ratio values labelled on each image.

#2.3. Fig 1 shows Nir1-LNS2 translocates to plasma membrane upon PMA stimulation in a PLD activity-dependent manner. However, the image in Fig1K is not super convincing since there is already a decent amount of plasma membrane localization of the sensor at $t = 0$ min, which looks considerably different from the $t = 0$ min image shown in 1F.

We have updated the images both in **Figure 1F** and **Figure 1K** to best represent the mean basal localization as determined in Supplemental Figure 1.

#2.4. Fig 3 and Fig 4 need the validation of PIP depletion/production using PIP-binding probes.

These controls are shown in Figure 4B. We have only included PH-PLCd1 to show PIP2 levels as the large PIP2 production by a PIP5K also indicates the large elevation of the substrate PI4P.

This control data has now been included in Figure 3, and is referenced by the **Figure 3 legend** and the following text from **p.8, left column, 4th paragraph**: “As a *negative control*, we expressed a doubly catalytically dead mutant of PJ. When PJ-Dead was recruited to the PM, we confirmed that PIP2 and PI4P levels remained unaltered by seeing stable association of the PIP2 biosensor Tubby(c) with the PM (Figure 3A). We observed no loss of the PM localization of Nir1-LNS2 with PJ-Dead recruitment (Figure 3A, 3E). When the active PJ was expressed in HEK293A cells, there was a slight loss of Nir1-LNS2 at the PM even before PJ recruitment (Figure 3B, 3E), although this was not significant as compared to pre-stimulated cells expressing PJ-Dead (Figure 3F). However, Nir1-LNS2 did move off the PM into the cytosol after PJ recruitment, to a similar extent that the PIP2 biosensor Tubby(c) moved off the PM (Figure 3B, 3E). AUC analysis of the Nir1-LNS2 response showed there was a significant reduction of Nir1-LNS2 PM localization (Figure 3G).

Since PJ depletes both PIP2 and PI4P, we examined which of these lipids specifically contribute to Nir1-LNS2 membrane binding. We utilized an FKBP-INPP5E construct that depletes PIP2 but does not deplete PI4P at the PM, as seen by the significant loss of PM-localized Tubby(c) (Figure 3C). Then FKBP-Sac1, an FKBP-PJ construct that has a catalytically dead INPP5E domain, but an active Sac1 domain was used to deplete PI4P without altering PIP2 levels, as seen by removal of the PI4P biosensor P4Mx1 from the PM (Figure 3D). Recruitment of FKBP-INPP5E did not significantly affect Nir1-LNS2 localization (Figure 3C, 3E, 3G). However, recruitment of FKBP-Sac1 slightly, but not significantly affected Nir1-LNS2 localization (Figure 3D, 3E, 3G). This data suggests that decreasing the anionic charge of the membrane through depletion of PIPs slightly reduces Nir1-LNS2’s ability to interact with the PM,

but it doesn't fully re-localize the sensor. Therefore, this is a small caveat to consider when using Nir1-LNS2 to study PA."

#2. "Advance": The key significance of the manuscript, which is the superiority of Nir1-LNS2 over existing PA-binding probes, is not clear from the data provided. Other than that part, the study does not seem to include significant finding, since the binding of Nir1-LNS2 to PA itself is already known (EMBO Rep. 2013 Oct; 14(10): 891-899, Mol Biol Cell. 2022 Mar 1;33(3):br2).

While the Kim et al., paper referenced by the reviewer does show that the LNS2 binds to PA, this same group later published data showing that the LNS2 binds to both PA and DAG. (Kim et al., 2015 doi: 10.1016/j.devcel.2015.04.028). Therefore, we believe our data which unequivocally shows that the LNS2 does not bind DAG, is a significant advancement in the field. Aside from the creation of the new biosensor, it progresses our understanding of the mechanism of the Nir family lipid transfer proteins, which are vital to PM lipid homeostasis.

To highlight this point, we have added the following paragraph to the discussion on **p.14, right column, 1st paragraph**: "The lack of Nir1-LNS2 binding to DAG-rich liposomes (Figure 2), DAG produced at the mitochondria (Figure 4), and DAG analogs (Figure 5) shows that the LNS2 domains only binds to PA rather than to PA and DAG as has been reported previously (Kim et al., 2015). In this study, we redefined the boundaries of the LNS2 domain based on the structure of the Lipin/Pah family domains and the AlphaFold prediction for the Nir1-LNS2. The new boundaries included the entire fold that is conserved between the Lipins/Pahs and the Nirs. Therefore, we suspect that the expansion of the LNS2 domain in our work explains the

differences in our data and the published literature regarding DAG binding. Importantly, the data obtained with our amended Nir1-LNS2 suggests that within the context of the lipid transfer cycle and MCS formation, the Nir family of PITPs translocate to the PM solely based on PA. This information will be important as the field continues to determine the exact mechanism of the Nir PITPs in lipid homeostasis.”

#3.4. *Due to PA's versatile biological roles, the evidence provided by the MCS experiment is far from enough to conclude that Nir1-LNS2 does not interfere with PA function. Further examination of various endogenous pathways is warranted before making the statement "Therefore, Nir1-LNS2 can be used without concern of affecting downstream events".*

We have rewritten the quoted sentence for a more nuanced interpretation on **p.13, right column, second paragraph**: *“While PA has a plethora of cellular functions, the fact that Nir1-LNS2 expression does not disrupt MCS formation shows promise that the high affinity of Nir1-LNS2 will not inhibit downstream PA signaling.”*

#3.5. *In Fig. 3A-D (Left), it is unclear to what extent PIPs are reduced after treatment with FKBP-tagged PIP phosphatases. The treatment depicted in the illustration should be accompanied by data, e.g., % of PIPs being degraded after treatment.*

This comment is addressed in our response to **#2.4**, where we show the addition of control biosensors for PIP₂ and PI4P, and also propose new experiments in TIRFM for more sensitive and precise measurements.

4. Description of analyses that authors prefer not to carry out

#1.1a: *Have the authors considered using the DGK inhibitor R59022 to selectively block the conversion of DAG to PA by DGK? Such an experiment could provide additional evidence for the requirement of DGK activity and consequent PA formation for Nir1-LNS2 membrane localization.*

We did indeed attempt experiments with R59022, and have made several unexpected findings with the compound that go way beyond the scope of the current manuscript. In short, although R59022 reduces DGK catalytic activity, it also potently drives over-expressed or endogenous DGK α to the plasma membrane, and induces large accumulations of PM PA. This complicated interpretation of data obtained with this compound. We are currently preparing a manuscript detailing the novel and unexpected effects.

#3.6. *In Fig. 4C, the plasma membrane (PM) localization of Nir1-LNS2 and NES-PABDx2-Spo20, as determined by the "intensity PM/Cyto," should be analyzed following the ectopic production of*

Revision Plan

PI4P and PIP2. Although mitochondria do not apparently recruit Nir1-LNS2 or NES-PABDx2-Spo20 after induced PI4P and PIP2 production, it remains possible that the subsequent trafficking of PI4P and PIP2 from mitochondria might sequester the biosensors away from the PM into the cytoplasm, thereby reducing the "intensity PM/Cyto" of Nir1-LNS2.

We cannot easily determine the PM/cyt ratio in this experiment as we included a mitochondrial marker rather than a PM marker when imaging. However, based on the images, there is no change in the PM intensity of the Nir1-LNS2 and NES-PABDx2-Spo20 biosensors. The images included in Figure 4 are representative of this localization.

#3.7. *It would be valuable to determine the half-life (stability) of Nir1-LNS2.*

In all of our transient transfections, the Nir1-LNS2 shows good stability where we don't expect degradation to be a major concern. Furthermore, stability has not usually been factor considered in the creation any of the current widely used lipid biosensors.

June 4, 2024

Re: JCB manuscript #202405174T

Dr. Gerald R Hammond
University of Pittsburgh School of Medicine
Department of Cell Biology
BST-South, Room #327 3500 Terrace St
Pittsburgh, PA 15261

Dear Dr. Hammond,

Thank you for submitting your manuscript entitled "Nir1-LNS2 is a novel phosphatidic acid biosensor that reveals mechanisms of lipid production" from Review Commons. We agree that a suitably revised study seems very appropriate as a JCB Tool. Therefore we invite you to submit a manuscript revised as outlined in your revision plan to JCB for further review by the original reviewers from Review Commons, or if they are unavailable suitable alternatives.

GENERAL GUIDELINES:

Text limits: Character count for a Tool is < 40,000, not including spaces. Count includes title page, abstract, introduction, results, discussion, and acknowledgments. Count does not include materials and methods, figure legends, references, tables, or supplemental legends.

Figures: Tools may have up to 10 main text figures. Figures must be prepared according to the policies outlined in our Instructions to Authors, under Data Presentation, <https://jcb.rupress.org/site/misc/ifora.xhtml>. All figures in accepted manuscripts will be screened prior to publication.

*****IMPORTANT:** It is JCB policy that if requested, original data images must be made available. Failure to provide original images upon request will result in unavoidable delays in publication. Please ensure that you have access to all original microscopy and blot data images before submitting your revision. ***

Supplemental information: There are strict limits on the allowable amount of supplemental data. Tools may have up to 5 supplemental figures. Up to 10 supplemental videos or flash animations are allowed. A summary of all supplemental material should appear at the end of the Materials and methods section.

Please note that JCB now requires authors to submit Source Data used to generate figures containing gels and Western blots with all revised manuscripts. This Source Data consists of fully uncropped and unprocessed images for each gel/blot displayed in the main and supplemental figures. File names for Source Data figures should be alphanumeric without any spaces or special characters (i.e., SourceDataF#, where F# refers to the associated main figure number or SourceDataFS# for those associated with Supplementary figures). The lanes of the gels/blots should be labeled as they are in the associated figure, the place where cropping was applied should be marked (with a box), and molecular weight/size standards should be labeled wherever possible. Source Data files will be made available to reviewers during evaluation of revised manuscripts and, if your paper is eventually published in JCB, the files will be directly linked to specific figures in the published article.

The typical timeframe for revisions is three to four months. Please note that papers are generally considered through only one revision cycle, so any revised manuscript will likely be either accepted or rejected.

Thank you for this interesting contribution to Journal of Cell Biology. You can contact us at the journal office with any questions at cellbio@rockefeller.edu.

Sincerely,

William Prinz, PhD
Monitoring Editor

Andrea L. Marat, PhD
Senior Scientific Editor

Journal of Cell Biology

Manuscript number: RC-2024-02413R

Corresponding author(s): Hammond, Gerald

1. General Statements [optional]

We are grateful to the three reviewers for such thorough and thoughtful comments, and pleased to report that we were able to incorporate the majority of their suggestions. Reviewer's comments are *in light italics*, whilst our responses appear in regular font below. We added reviewer numbering for ease of cross-reference to the original comments, with the format: reviewer X's comment number N as **#X.N**

We have also included paragraph numbers in the manuscript that appear as (1) to facilitate the description of where changes have been made.

It should also be noted that we have renamed the Nir1-LNS2 biosensor PILS-Nir1 as it utilizes the phosphatidic acid-interacting Lipin-like sequence of Nir1. Therefore, we will refer to it as PILS-Nir1 throughout this document.

Overall, we were thrilled that the reviewers agreed that our work is of significance and broad interest:

"The development of a new, superior PA-sensor is a significant advance in the fields of lipid signaling and specific lipid-protein interactions, that will benefit research on lipid-mediated cellular signaling and intracellular lipid trafficking." – **reviewer 1.**

"The lipid biology community would be highly interested in using the new PA-binding tool to study lipid localization in live cells." – **reviewer 2.**

"Tracking intracellular phosphatidic acid (PA) in live cells is essential for understanding its cellular functions, leading to the development of genetically encoded lipid biosensors. While several PA biosensors have been developed, they often suffer from limited sensitivity or specificity." – **reviewer 3**

Table of Contents

1. General Statements [optional].....	1
2. Description of revisions that have been addressed.....	2
3. Description of analyses that authors prefer not to carry out.....	21

2. Description of revisions that have been addressed

#1.1: *CCH treatment of HEK293A cells leads to the PM localization of the DAG sensor C1ab-Prkd1 as well as Nir1-LNS2 (Fig. 5), and the kinetics of these changes - Nir1-LNS2 would lag behind C1ab-Prkd1 fluorescence - is taken as evidence for Nir1-LNS2's specific binding of PA rather than DAG: Pag. 10: 'When we look at the first 2-minutes after CCh addition, we see that C1ab-Prkd1 moves to the PM much faster than Nir1-LNS2 does (Figure 5D). The delay in Nir1-LNS2 translocation makes sense given DAG is produced first and then converted into PA, again indicating that Nir1-LNS2 is specific for PA. 'Fig. 5 legend: 'The Nir1-LNS2 response to PLC depends on PA and not DAG.(...) (D) Nir1-LNS2 translocation to the PM (data replicated from Figure 5C) lags behind C1ab-Prkd1 (data replicated from Figure 5B) in response to CCh addition. 'The validity of the conclusion from this experiment seems questionable. The argument relies on the low values of Nir1-LNS2's normalized fluorescence intensity compared to C1ab-Prkd1's, in the first two minutes of stimulation, when DAG is expected to accumulate (Fig. 5D). However, if Nir1-LNS2 would bind DAG, the resulting fluorescence is expected to be low, i.e. compared to the much higher signal resulting from subsequent PA binding. Moreover, in this system, which co-expresses the two sensor proteins, competition in binding may account for the apparent precedence of one sensor over the other. Thus, even if the increase in C1ab-Prkd1 fluorescence would precede Nir1-LNS2 - following the authors' interpretation - this would not exclude binding of DAG by Nir1-LNS2. Fig. 5D shows the confocal images of cells after 30 sec CCh treatment using the two sensors, next to the respective controls, replicated from Fig. 5B/C. However, in Fig. 5D, different colors are used for Nir1-LNS2 than in Fig. 5C, which makes comparison along the time course difficult. In conclusion, the data presented in Fig. 5 do not exclude DAG binding by Nir1-LNS2, and modification of the conclusions from this experiment throughout the ms (including the cited sentences) is recommended. Consider removal of Figure 5D. Despite these considerations, the authors' final conclusion regarding the specificity of Nir1-LNS2 towards PA appears well supported (e.g. by the data presented in Figure 4).*

We agree with the reviewer that the interpretation of the kinetic data is ambiguous and does not fully negate the idea that PILS-Nir1 may bind to DAG. We have modified the interpretation accordingly. However, we have left the kinetic comparison of the DAG vs PILS-Nir1 biosensors since these reflect the expected dynamics of the two lipids downstream of PLC. The data are now interpreted as follows on **page 15 paragraph 59**: *However, in cells treated with CCh and then atropine, PILS-Nir1 localized to the PM after CCh was added but then returned to the cytoplasm over the 15-minute treatment with atropine as PA levels declined (Figure 5C). Additionally, we observed that the PM accumulation of PILS-Nir1 after CCh addition lagged behind the accumulation of DAG, which is consistent with the conversion of DAG to PA by*

DGKs (Figure 5D). Overall, this experiment shows that PILS-Nir1 binding to the PM follows the expected kinetic profile of DGK-produced PA.

Likewise, the **legend for figure 5** is now labelled “PILS-Nir1 detects PA produced downstream of PLC” to remove any overarching conclusions on DAG versus PA binding.

#1.2a: P 4 and Fig. 2 mention a 'novel domain structure', responsible for binding of PA and PIP₂ (in vitro). What exactly is this novel domain structure?

We have clarified that the domain architecture of PILS-Nir1 is not a novel domain structure generally, but novel for a PA binding protein which are typically just helices such as that seen in Spo20. **Page 3 paragraph 11** now reads “As the LNS2 tertiary structure is unique compared to the helical nature of PABD-Spo20, we investigated Nir1-LNS2 as a putative novel PABD.” And **Page 10 paragraph 40** reads “Therefore, PILS-Nir1 demonstrates a novel PA binding domain with a tertiary structure beyond the simple amphipathic helix found in Spo20.”

Figure 2 is now titled “PILS-Nir1 shows specificity for PA and PIP₂ in vitro, based on a novel PABD structure.” **Figure 2D** now also includes the AlphaFold prediction for PILS-Nir1 to make visualization of the architecture easier.

#1.2b: Why have the two parts of Nir1-LNS, the AAH and the 263 amino acid domain, not been tested in similar liposome assays as in Fig. 2A? Lipid binding in vivo, as tested in the experiment of Fig. 2D, is

confounded by endogenous PA binding proteins. PA is expected to bind the SIDGS motif, as this is conserved from the Lipin catalytic motif (p. 5). However, experimental evidence of this appears to be lacking. Nevertheless, it is several times presented as a fact in the text. Pag. 7: 'This suggests that the SIDGS motif alone is not the sole PA binding pocket as the LNS2 domain requires both that motif and the amphipathic helix for sustained binding to membrane embedded PA.' Pag. 13: '... One reason for this difference could be that the Nir1-LNS2 is not a novel bona fide PA binding pocket. Rather, it requires both the SIDGS motif and an N-terminal amphipathic helix for membrane interactions (Figure 2). These conclusions therefore appear to be not accurate and would need to be rephrased. If the authors could show that PA binding by Nir1-LNS can be eliminated by mutating residues in the SIDGS motif, this would not only substantiate the above claims, but also make for a negative control protein, next to Nir1-LNS2. For future applications of Nir1-LNS2 as PA biosensor in other organisms this would be useful.

We have clarified that the SIDGS motif is not the actual location of PA binding, but rather is only the motif conserved with the Lipin/Pah active site. Residue K820 appears to be a PA coordinating residue near the SIDGS, as the positive residue can interact with the negative lipid. Mutating this residue to confirm PA binding was an excellent reviewer suggestion, and we have updated the manuscript and **Figure 2** accordingly:

Page 9, paragraph 36: *We next determined how PILS-Nir1 binds to PA at the structural level. Sequence homology of PILS-Nir1 together with AlphaFold structural predictions showed a high degree of similarity to the Lipin family of enzymes, minus key residues necessary for Mg²⁺ binding and catalysis. The Lipin catalytic motif DxDxT is partially conserved in PILS-Nir1 as a SIDGS motif spanning residues 742-746. We looked for positively charged residues nearby that could bind to the PA in the membrane and stabilize its position in the SIDGS site. The active site of the Lipins has a nearby Lys residue which was predicted to perform this role (Khayyo et al., 2020). AlphaFold analysis of PILS-Nir1 showed that residue K820 similarly projects toward the SIDGS site where it would be able to contact the negatively charged PA (Figure 2D).*

Page 10, paragraph 39: *We probed into the suspected PA binding residue K820 by mutating it into a negatively charged Glu residue, which should disrupt its interaction with the negatively charged PA. The K820E mutation completely ablated PILS-Nir1 localization at the PM under basal conditions, recruitment to the PM after PMA stimulation, and association with PA liposomes (Figure 2E, 2F). Interestingly, the K820E mutation did not alter PILS-Nir1 binding to PIP2 in vitro, demonstrating the specificity of this site for PA and suggesting that the PIP2 binding seen in the wild-type construct is simply due to an electrostatic interaction.*

We have also updated the discussion (**page 21 paragraph 76**): *To interact with membranes, PILS-Nir1 requires both K820 near the SIDGS pocket, which is specific for PA binding, and an N-terminal amphipathic helix, which is thought to generally interact with negatively charged lipids (Kim et al., 2013; Khayyo et al., 2020) (Figure 2).*

#1.3. Pag. 7: *'...small caveat to consider when using Nir1-LNS2 to study PA, the data also demonstrates that Nir1-LNS2 is not specifically interacting with any of the PM PIPs in cellular membranes. This seems not accurate, since the data in Fig. 3 suggest that PI4P could be involved in membrane localization of Nir1-LNS2. It remains however unresolved whether this is a specific interaction with this PIP.*

We have redone the experiment in **Figure 3** using TIRF microscopy to increase the sensitivity of the imaging. We also included control biosensors: Tubby(c) to verify depletion of PIP₂ or P4Mx1

to verify depletion of PI4P from the plasma membrane in the same cells as were analyzed for PILS-Nir1 or NES-PABDx2-Spo20 localization. We now see that neither depletion of PIP₂ or PI4P affects PILS-Nir1's association with the PM.

Page 11 paragraph 46 now reads: *When PJ-Dead was recruited to the PM, we confirmed that PIP2 and PI4P levels remained unaltered by seeing stable association of the PIP2 biosensor Tubby(c) with the PM. Additionally, we observed no loss of the PM localization of PILS-Nir1 or NES-PABDx2-Spo20 with PJ-Dead recruitment (Figure 3A). When the active PJ was recruited in HEK293A cells, we saw that PILS-Nir1 was able to remain associated with the PM, but that NES-PABDx2-Spo20 moved off the PM to a similar extent that the PIP2 biosensor Tubby(c) moved off the PM (Figure 3B).*

Since PJ depletes both PIP2 and PI4P, we examined if either of these lipids specifically contribute to PILS-Nir1 or NES-PABDx2-Spo20 membrane binding. FKBP-INPP5E was used to deplete PIP2, but not PI4P at the PM, as seen by the significant loss of PM-localized Tubby(c). FKBP-PJ-Sac1, an FKBP-PJ construct that has a catalytically dead INPP5E domain, but an active Sac1 domain was used to deplete PI4P without altering PIP2 levels, as seen by removal of the PI4P biosensor P4Mx1 from the PM. The association of PILS-Nir1 was unaffected by either FKBP-INPP5E or FKBP-PJ-Sac1 recruitment (Figure 3C, 3D). However, there was slight loss of NES-PABDx2-Spo20 at the PM upon FKBP-PJ-Sac1 degradation of PI4P (Figure 3D). This data suggests that PILS-Nir1 is specific for PA in cells even though it shows association with the anionic lipid PIP2 in vitro. In contrast, decreasing the anionic charge of the membrane through depletion of PIPs does affect NES-PABDx2-Spo20's ability to associate with the PM. This is not too surprising given the previously reported interactions of Spo20 with PIPs (Nakanishi et al., 2004; Horchani et al., 2014).

#1.4. Please note that the presence of PE increases the ionization (and negative charge) of PIP2 (Graber et al., 2012) rather than dilutes the negative charge as stated in the Discussion on p.13. Please revise!

We have updated the text on **Page 21 Paragraph 76**: “One reason for this discrepancy could be differences in the negative charge of cellular membranes versus that of liposomes. To interact with membranes, PILS-Nir1 requires both K820 near the SIDGS pocket, which is specific for PA binding, and an N-terminal amphipathic helix, which is thought to generally interact with negatively charged lipids (Kim et al., 2013; Khayyo et al., 2020) (Figure 2). The negative charge of PIP2 and therefore its ability to recruit the N-terminal helix of PILS-Nir1 depends on its protonation state (Kooijman et al., 2009). PIP2 can hydrogen bond with other lipids such as PI, with itself inside of PIP2-rich domains, or even with neighboring proteins, all of which would attenuate its charge. Phosphatidylethanolamine on the other hand, increases PIP2 ionization and its negative charge (Graber et al., 2012; Borges-Araújo and Fernandes, 2020). Therefore, the degree of charge on PIP2 is greatly dependent on the local protein and lipid environment. Due to this, PIP2 molecules in cellular membranes may possess less charge compared to PIP2 molecules in liposomes. Thus, in cells, the N-terminal helix’s interaction with PIP2 may be very weak compared to the interaction of K820 with PA, so that only PA levels influence membrane binding.”

#1.5. P 1 The authors may consider adding a 4th criterium for a lipid biosensor: the sensor should not serve as a sink for the lipid by removing/sequestering it from the active pool, thereby interfering with other interactions/conversions.

We agree that biosensors should not sequester a significant fraction of their cognate lipids or affect downstream pathways by competing with endogenous binding partners. We have rewritten the following text regarding Figure 6 to make this distinction more clear:

Page 19 paragraph 68: “As PILS-Nir1 shows high affinity for PA across cell lines, this brings up the concern that use of PILS-Nir1 will sequester PA and inhibit endogenous signaling pathways that depend on PA, an effect that has been seen with other biosensors for low abundance lipids (Holmes et al., 2025).”

Page 19 paragraph 70: “It is suggested that cellular homeostasis may compensate for the amount of bound lipid by increasing synthesis of free lipid, as this has been seen with the PIP₂ biosensor PH-PLCδ1 (Traynor-Kaplan et al., 2017). While PA has a plethora of cellular functions, the fact that PILS-Nir1 expression does not disrupt MCS formation shows promise that the high affinity of PILS-Nir1 will not inhibit downstream PA signaling.”

#1.6. Nir1 lacks a PITP domain (Fig. 1), yet is referred to as lipid transfer protein: please elaborate/explain.

The following text has been added to **Page 3 paragraph 9** to clarify this point: *While Nir1 lacks a functional PITP domain, it was initially classified as part of the PITP family based on the homology of its other domains with Nir2 and Nir3. Furthermore, Nir1 has a role in lipid transfer by facilitating Nir2 recruitment to the MCS (Quintanilla et al., 2022).*

#1.7. *Fig. 2A suggests cooperativity in binding of Nir1-LNS2 to PA-containing liposomes. Please mention/comment! Does binding to PIP2-containing liposomes also exhibit cooperativity?*

We performed an additional, quantitative lipid binding experiment as described in our **response to #2.2** (now shown in **Figure 2B**). We found that PA binding was fit with a model without cooperativity.

#1.8. *Indicate the concentrations of PC and protein in the legend to Fig. 2 panels A and B. M&M says 2 mM PC, according to the PA-concentrations above panel 2A, this should be 1 mM. Please clarify.*

For ease of comparison, we have updated **Figure 2** to report all lipid mixtures in terms of percent of lipid species, and we have updated the **Figure 2 legend** as follows, “Lipids indicated were mixed with POPC to produce a 2 mM solution, then 50 μL of the resulting liposome mixture was incubated with 50 μL of protein at ~1 mg/mL to produce 1 mM lipids in the assay.”

#1.9. *In Fig. 2B, PI is missing. Any specific reason?*

We have updated the text on **Page 7 paragraph 31** to discuss the low levels of PI at the PM, which is why we did not include this lipid. “Using this same PC background, we tested the efficacy of the PM lipids DAG, PA, phosphatidylserine (PS), PI4P and PIP2

in recruiting PILS-Nir1 to membranes. While PI serves as a substrate for PI4P and PIP2 synthesis (collectively referred to as the phosphatidylinositol phosphates (PIPs)) at the PM, levels of PI at the PM are very low compared to the PIPs and therefore PI itself was not tested (Zewe et al., 2020; Pemberton et al., 2020).

#1.10. Move Fig. 2C to the Introduction and extend it to illustrate the shared conserved features of Nir1-LNS2 and lipin.

We would like to keep the diagram of the PILS-Nir1 in Figure 2 where the features are discussed in more detail than in the introduction. However, we did add the actual AlphaFold-predicted structure along with the cartoon to make the diagram more helpful (**see response to #1.2**). We also edited the introduction on **page 3 paragraph 10** that refers the reader to the cartoon in Figure 2. *“The Nir proteins all contain a C-terminal Lipin/Nde1/Smp2 (LNS2) domain (Figure 1I). The AlphaFold predicted structure of this domain shows similarity to the Lipin/Pah family of phosphatidic acid phosphatases (PAPs). Lipin/Pah PAPs interact with the membrane through an N-terminal amphipathic helix and catalyze the dephosphorylation of PA through a DxDxT-containing Mg²⁺-binding active site (Khayyo et al., 2020). These features are conserved in the Nir LNS2 domains, except for the catalytic Asp in the DxDxT motif and another Mg²⁺-coordinating residue (Figure 2D).*

#1.11. P 13. While real-time IMPACT does not directly report on PA levels as it does not use the endogenous PLD substrate PC, ...'It is true that this method doesn't directly report on PA levels, but that is because it uses a click chemistry probe as substrate for PLD's transphosphatidylation reaction. Contrary though to what is stated by the authors, this reaction still uses the endogenous substrate of PLD, PC (Liang et al. 2019; www.pnas.org/cgi/doi/10.1073/pnas.1903949116).

We have rewritten the aforementioned sentence in the discussion (**Page 22 paragraph 80**): *“However, real-time IMPACT does not directly report on PA levels as it creates a bio-orthogonal fluorescent lipid. Instead, it offers several advantages such as being able to interrogate lipid trafficking over time.”*

#1.12. Fig. 1K: Control must have been treated with PMA plus vehicle (DMSO); if so, please indicate that vehicle was added.

Unfortunately, we did not include the appropriate vehicle control in this experiment. However, we note that in many other experiments, the presence of the equivalent concentration of DMSO (0.1% v/v) does not affect PILS-Nir1 localization; examples can be seen in control experiments in **Figures 3 and 4**, where cells expressing inactive control enzymes treated with 1 μ M rapamycin (also in 0.1% DMSO) do not exhibit a change in PILS-Nir1.

We have updated **Figure 1K and its legend** to accurately state the control conditions. The legend now reads “Stimulating HEK293A cells with 100 nM PMA and 750 nM of the PLD1/2 inhibitor FIPI diminished the PM translocation of PILS-Nir1 seen with PMA and cell media.”

#1.13. *Figure 6C: How is $\Delta F_t/F_{pre}$ defined? Add to legend.*

The F_t/F_{pre} analysis is now introduced in Figure 3, so the **Figure 3 legend** explains this ratio as “The fluorescence at a given time (F_t) was divided by the fluorescence before rapamycin stimulation (F_{pre}).”

This is also now explained in the text on **page 11 paragraph 44**: “The fluorescence at a given time (F_t) was divided by the fluorescence before rapamycin stimulation (F_{pre}), so that as biosensors moved off of the PM, the fluorescence ratio decreased.”

The **Figure 6 legend** now also reiterates this: “Nir2 MCS formation was quantified as the change in fluorescence at a given time (F_t) divided by the fluorescence before 100 μ M CCh stimulation (F_{pre})”

#1.14. *P 4: 'The boundaries of the Nir2-LNS2 (Uniprot: O00562) and Nir3-LNS2 (Uniprot: Q9BZ72) were also defined using AlphaFold predictions of the structure of these domains.' How was the extent of each of these domains determined - they are much larger than the previously published sequences of LNS2 domains (Kim et al. 2013; Embo Rep. 14:891-899. doi:10.1038/embor.2013.113)?*

We have clarified the definition of our boundaries by updating **page 7 paragraph 26**: “We then tested the LNS2 domains of the other two Nir family members, Nir2 and Nir3, to determine their sensitivity to PA. The boundaries of the Nir2-LNS2 (Uniprot: O00562) and Nir3-LNS2 (Uniprot: Q9BZ72) were also defined using AlphaFold predictions of these domains. Previous definitions of the Nir2-LNS2 domain considered the domain smaller than we do here (Kim et al., 2013, 2015). However, according to AlphaFold, the boundaries set previously excluded a large N-terminal immunoglobulin domain that is conserved in the Lipin/Pah PAPs, as well as disrupted the domain fold that is homologous to the Lipin active site. Therefore, we set boundaries in our constructs to include the entire predicted LNS2 fold.”

#1.15. *In Fig. 3 legend, specify the starting condition of Nir1-LNS2 binding?? Which fluorescence are we looking at?*

In the updated **Figure 3** now using the TIRF data, we have updated the labels and the figure legend to make it more clear which color corresponds to which biosensor. **See response to #1.3.**

#1.16. In legend to Fig. 6 please specify the fluorescent tags used. Have they been shown not to affect protein function?

The **legend to Figure 6H** now reads “GFP-Nir2 was co-expressed with either iRFP-PILS-Nir1 or a control biosensor iRFP-Tubby(c), which binds PIP₂. The graph shows the grand means ± SEM for 4-5 experimental replicates (n=44-48 cells).”

Page 19 paragraph 69 was updated to read: We co-expressed a GFP-tagged Nir2 and either iRFP-PILS-Nir1 or iRFP-Tubby(c), a PIP₂ biosensor that is not expected to affect MCS formation. It should be noted that although we have used the NG-tagged PILS-Nir1 throughout this work, iRFP and mCherry-tagged PILS-Nir1 sensors have behaved the same as the NG-tagged version in the experiments where we utilized them (data not shown).

#2.1. Nir1-LNS2 seems to show variable basal localization across different representative images presented in the manuscript. A part of them were justified by the effect of other anionic species by PIP (such as Fig 3 where they co-expressed PIP-degrading enzymes). For example, cells in Fig 1F and those in Fig 4F show quite different basal localization of Nir1-LNS2. Is it due to difference in expression level, cell conditions, or other factors? The significant amount of plasma membrane basal localization seems to indicate that Nir1-LNS2 localization is affected by its binding to PI(4,5)P₂.? The significant and potentially variable plasma membrane localization of Nir1-LNS1 can limit the utility of this probe.

We have added **Supplemental Figure 2** to show the range of PILS-Nir1 basal localization compared to NES-PABDx2-Spo20 and PASS. We believe that this localization is due to variable amounts of basal PA. The following paragraphs have been added on **page 10 paragraph 41** to discuss this, *Since the truncated and mutated PILS-Nir1 constructs showed reduced basal PM localization, we wanted to further characterize the basal localization of the wild-type PILS-Nir1. PILS-Nir1 localization varies between resting cells, but analysis of all the cells used throughout this study determined that the basal PM/Cyto ratio of the wild-type PILS-Nir1 is 1.141 ± 0.097 (mean ± SEM), which suggests that at resting conditions PILS-Nir1 is slightly enriched at the PM (**Supplemental Figure 2A, 2D**). When we did the same analysis for all the cells where we expressed NES-PABDx2-Spo20 or PASS, we observed that NES-PABDx2-Spo20 had a similar basal PM/Cyto ratio and spread of data to that of PILS-Nir1. PASS had a lower ratio than NES-PABDx2-Spo20 and PILS-Nir1, presumably due to its single PABD limiting its affinity for PA (**Supplemental Figure 2B, 2C, 2E**).*

Page 10 paragraph 42: *As the K820E mutation disrupted PILS-Nir1 PM association at rest, this suggests that the spread in the basal localization of these sensors reflects variable PA levels in the PM at resting conditions. Mass spectrometry data has estimated that PA comprises 2 mol% of the inner leaflet of the PM in resting red blood cells (Lorent et al., 2020). Furthermore, FRET-based imaging of PA has indicated that there are detectable levels of PA under basal conditions, and this approach also showed that there was variability in basal PA levels within individual cells of a population (Nishioka et al., 2010). Overall, our data suggests that the high*

affinity of PILS-Nir1 for PA is reflected in both its basal association with the PM and its response to stimulations such as PMA.

To address the idea that PIP₂ is responsible for the basal localization of PILS-Nir1, we have redone the experiment in **Figure 3** using TIRF microscopy to increase the sensitivity of the imaging. We also included control biosensors: Tubby(c) to verify depletion of PIP₂ or P4Mx1 to verify depletion of PI4P from the plasma membrane in the same cells as were analyzed for PILS-Nir1 or NES-PABDx2-Spo20 localization. We now see that neither depletion of PIP₂ or PI4P affects PILS-Nir1's association with the PM. **See response to #1.3.** Therefore, this data suggests that the basal localization of PILS-Nir1 is not mediated by the PIPs, although NES-PABDx2-Spo20's basal localization could be influenced by these lipids.

#2.2. The authors mention high affinity of Nir1-LNS2, but it lacks in vitro characterization that should demonstrate the higher affinity of Nir1-LNS2 compared to conventional probes such as Spo20. The authors should perform side-by-side comparison in Fig 2 to compare the PA affinity and specificity of Nir1-LNS2 compared to Spo20.

We have taken the reviewer's excellent suggestion and directly compared PILS-Nir1 and PABD-Spo20 with liposome binding assays, now in **Figure 2C**, which shows that PILS-Nir1 binds PA liposomes significantly better than PABD-Spo20 does. We have also performed a binding assay with PABD-Spo20 to compare Spo20's binding affinity (**Rebuttal Figure 1**) to that

of PILS-Nir1 (**Figure 2B**). However, the maximum specific PABD-Spo20 that bound to any of the PA liposomes was 6%, so the binding curve was not included in the manuscript as it did not provide any further information than that already shown in **Figure 2C**. Note that these liposomes were a standard 80 mol% PC and 20 mol% PA and then the total lipid concentration (mM) was increased. Our observation of poor PABD-Spo20 binding to 20 mol% PA liposomes is consistent with previous literature, where significant binding to PA-containing liposomes was only observed above 50 mol% PA (see figure 6 in Nakanishi et al., 2004, <https://doi.org/10.1091/mbc.e03-11-0798>).

#2.3. Fig 1 shows Nir1-LNS2 translocates to plasma membrane upon PMA stimulation in a PLD activity-dependent manner. However, the image in Fig1K is not super convincing since

there is already a decent amount of plasma membrane localization of the sensor at $t = 0$ min, which looks considerably different from the $t = 0$ min image shown in 1F.

We have updated the images both in **Figure 1F** and **Figure 1K** to best represent the mean basal localization as determined in **Supplemental Figure 2**.

#2.4. Fig 3 and Fig 4 need the validation of PIP depletion/production using PIP-binding probes.

We have redone the experiment in **Figure 3** now using TIRF for increased sensitivity, and we have included the data for PIP₂ depletion as monitored by the biosensor Tubby(c) and PI4P depletion monitored by P4Mx1. **See response to #1.3.**

In **Figure 4**, PIP₂ production is shown by recruitment of PH-PLCδ1 to mitochondria in **panel 4B**.

#2.5. In Discussion: "while in vivo it solely binds to PA (Fig 4)" - this claim does not seem to be true according to Fig 4, where the overexpression of PIP-degrading enzymes did affect

the Nir1-LNS2 basal localization.

We have redone this experiment in **Figure 3** using TIRF and now do not see that PIP degradation affects PILS-Nir1 membrane association. **See response to #1.3.** We have also updated the discussion on **page 21 paragraph 75** to make a more clear distinction, “As far as the use of PILS-Nir1 as a biosensor, one caveat is the discrepancy in its specificity: *in vitro* PA and PIP2 are sufficient to recruit PILS-Nir1 to PC liposomes (Figure 2), but *in vivo* only PA is sufficient for mitochondrial recruitment (**Figure 4**).

#2. “Advance”: *The key significance of the manuscript, which is the superiority of Nir1-LNS2 over existing PA-binding probes, is not clear from the data provided. Other than that part, the study does not seem to include significant finding, since the binding of Nir1-LNS2 to PA itself is already known (EMBO Rep. 2013 Oct; 14(10): 891-899, Mol Biol Cell. 2022 Mar 1;33(3):br2).*

While the Kim et al., paper referenced by the reviewer does show that the LNS2 binds to PA, this same group later published data showing that the LNS2 binds to both PA and DAG. (Kim et al., 2015 doi: [10.1016/j.devcel.2015.04.028](https://doi.org/10.1016/j.devcel.2015.04.028)). Therefore, we believe our data which unequivocally shows that the LNS2 does not bind DAG, is a significant advancement in the field. Aside from the creation of the new biosensor, this work progresses our understanding of the mechanism of the Nir family lipid transfer proteins, which are vital to PM lipid homeostasis.

To highlight this point, we have added the following paragraph to the discussion on **page 21 paragraph 74**: “The lack of PILS-Nir1 binding to DAG-rich liposomes (**Figure 2**), DAG produced at the mitochondria (**Figure 4**), and DAG analogs (**Figure 5**) shows that the Nir family LNS2 domain only binds to PA rather than to PA and DAG as has been reported previously (Kim et al., 2015). In this study, we redefine the boundaries of the Nir family LNS2 domain based on the structure of the Lipin/Pah family domains and the AlphaFold prediction for Nir1. The new boundaries include the entire fold that is conserved between the Lipin/Pah family and the Nir family (**Figure 2D**). Therefore, we suspect that the extended boundaries of the LNS2 domain in our work explain the differences in our data and the published literature regarding DAG binding. Importantly, the data obtained with our amended LNS2 suggests that within the context of the lipid transfer cycle and MCS formation, the Nir family of PITPs translocate to the PM solely based on PA. This information will be important as the field continues to determine the exact mechanism of the Nir PITPs in lipid homeostasis.

#2. “General assessment”: *The existing PA-binding probe using Spo20 is indeed quite blunt, which takes minutes to see appreciable accumulation of the probe upon PA production. Nir1-LNS2 can be indeed useful if it offers better spatiotemporal precision. However, the advantage of this tool over existing tools is not convincing without head-to-head comparison of either (1) *in vitro* characterization of PA binding affinity and selectivity*

between Nir1-LNS2 and Spo20 or (2) response to PA produced on different subcellular localizations other than plasma membrane and mitochondria (e.g., endosomes, golgi, and endoplasmic reticulum).

As described in **response to #2.2**, we have done a side-by-side comparison of PABD-Spo20 protein binding to liposomes alongside PILS-Nir1. This data is now in **Figure 2C** and shows that PILS-Nir1 binds PA significantly better than PABD-Spo20 does, but both sensors bind to PIP₂.

In addition, we have repeated the experiments in **Figure 3** to include NES-PABDx2-Spo20 and have found that this sensor is much more sensitive to changes in PM PIPs than PILS-Nir1 is. This is in agreement with previously published data (Nakanishi et al., 2004 doi: 10.1091/mbc.e03-11-0798; Horchani et al., 2014 doi: 10.1371/journal.pone.0113484). It also suggests that PILS-Nir1 is more selective for PA. **See response to #1.3.**

We agreed with the reviewer that looking at the PILS-Nir1 and Spo20 responses to PA production at other organelles would increase confidence that PILS-Nir1 has a higher affinity for PA. In **Figure 6A, 6B**, we have replicated previous data to facilitate the comparison between the sensors at the PM and mitochondria, and then also tested the two sensors at the Golgi, endosomes, and the ER in **Figure 6C-E**. The manuscript has been updated accordingly:

Page 17, paragraph 64: *In **Figure 1**, we saw that PILS-Nir1 bound PMA-stimulated PA at the PM with higher affinity than NES-PABDx2-Spo20 did. Then in **Figure 4**, we saw that PILS-Nir1 was recruited to the mitochondria more robustly than NES-PABDx2-Spo20 after PA was produced by FKBP-PI-PLC and FKBP-DGK. These data have been replicated in **Figure 6A and Figure 6B** to facilitate comparison between the biosensors.*

Page 17, paragraph 65: *To see if PILS-Nir1 showed higher affinity for PA in other organelles, we modified the FKBP-PI-PLC and FKBP-DGK α system by using FRB fragments that are targeted to other organelles: Golgi-FRB, Rab5-FRB, and ER-FRB. After recruitment of the FKBP constructs to these organelles, we saw that PILS-Nir1 responded to PA production in the Golgi and Rab5 endosomes with higher affinity than NES-PABDx2-Spo20 did (**Figure 6C, 6D**). When it came to PA produced at the ER, both biosensors responded only transiently, presumably due to quick metabolism of PA in this compartment. However, we did see that NES-PABDx2-Spo20 showed a higher peak response in this organelle. It seems that the localization of this sensor in the nucleus helped it to respond to PA made in the ER that is continuous with the nuclear envelope, while PILS-Nir1 tended to label ER structures that were more distal (**Figure 6E**). Altogether, this data suggests that PILS-Nir1 can serve as a high affinity PA biosensor at various cellular locations, although NES-PABDx2-Spo20 has some advantages when it comes to PA production in specific regions of the ER.*

#3.1. The direct measurement of the binding affinity of Nir1-LNS2 with PA, e.g., K_d, is essential; this information will help the field explore the potential usage of Nir1-LNS2.

The original liposome binding assay included in the manuscript used an assay with a set total lipid concentration, but with altered mol% PA. To more quantitatively address binding, we used varying concentrations of liposomes containing 20 mol% PA, increasing total [PA] to 800 μ M. We fit the data with a one-site specific binding with background model, producing a K_d of 422 μ M. The results are shown in **Figure 2B**. **See response to #2.2.**

#3.2. As mentioned by the authors, there is a confusing inconsistency regarding why Nir1-LNS2 binds to PIP2 in vitro but not in cells. Going beyond what has been discussed in the manuscript, there is a possibility that PIP2 could induce Nir1-LNS2 aggregation, leading to pelleting after centrifugation, among many other possibilities. I recommend the authors perform additional in vitro experiments, including but not limited to the liposome flotation assay to directly examine Nir1-LNS2 binding to the liposomes with varied compositions.

This was an excellent suggestion. We have included liposome flotation assays and circular dichroism in **Supplemental Figure 1**, and updated the manuscript accordingly.

Page 9 paragraph 34: To confirm that the PILS-Nir1 associated with liposomes was folded properly and not aggregated protein that had been pelleted, we performed a liposome flotation assay. This assay showed that little PILS-Nir1 was aggregated after incubation with PA or PIP2-containing liposomes (**Supplemental Figure 1A, 1B**). Circular dichroism (CD) analysis also showed that incubation of PILS-Nir1 with PA-containing liposomes did not change the CD spectra of PILS-Nir1. The spectra of PILS-Nir1 with and without PA liposomes both showed characteristic features of secondary structures, suggesting that membranes do not induce unfolding of PILS-Nir1 (**Supplemental Figure 1C**).

#3.3. In Fig. 2D, it would be beneficial to examine the constructs Nir1-613-630 and Nir1-631-894, comparing them with Nir1-LNS2 using liposome sedimentation and floatation assays to evaluate the contribution of the SIDGS motif and the amphipathic helix in binding PA.

Per our **response to #1.2b**, we looked around the SIDGS motif and found a K820 residue that would mediate the binding of membrane embedded PA. Swapping the charge of this residue did abolish PA binding in cells and *in vitro*.

#3.4. Due to PA's versatile biological roles, the evidence provided by the MCS experiment is far from enough to conclude that Nir1-LNS2 does not interfere with PA function. Further examination of various endogenous pathways is warranted before making the statement

Supplemental Figure 1. PILS-Nir1 does not aggregate or unfold in *in vitro* liposome experiments. (A) Schematic of PILS-Nir1 liposome flotation assay. **(B)** Representative SDS-PAGE gel and quantification of PILS-Nir1 bound to liposomes in the liposome fraction (L), as soluble protein in the middle fraction (M), or as aggregated protein in the pellet fraction (P) after reacting with POPC liposomes of varying compositions. **(C)** Circular dichroism for PILS-Nir1 with and without PA liposomes.

"Therefore, Nir1-LNS2 can be used without concern of affecting downstream events".

We have rewritten the quoted sentence for a more nuanced interpretation on **page 19 paragraph 70**: *"While PA has a plethora of cellular functions, the fact that PILS-Nir1 expression does not disrupt MCS formation shows promise that the high affinity of PILS-Nir1 will not inhibit downstream PA signaling."*

#3.5. In Fig. 3A-D (Left), it is unclear to what extent PIPs are reduced after treatment with FKBP-tagged PIP phosphatases. The treatment depicted in the illustration should be accompanied by data, e.g., % of PIPs being degraded after treatment.

This comment is addressed in our **response to #1.3**, where we now show the addition of control biosensors for PIP₂ and PI4P.

3. Description of analyses that authors prefer not to carry out

#1.1a: Have the authors considered using the DGK inhibitor R59022 to selectively block the conversion of DAG to PA by DGK? Such an experiment could provide additional evidence for the requirement of DGK activity and consequent PA formation for Nir1-LNS2 membrane localization.

We did indeed attempt experiments with R59022 and have made several unexpected findings with the compound that go way beyond the scope of the current manuscript. In short, although R59022 reduces DGK catalytic activity, it also potently drives over-expressed and/or endogenous DGK α to the plasma membrane. It induces large accumulations of PM PA that we see with our biosensor as well as in lipidomics studies. This complicated interpretation of the data obtained with this compound. We are currently preparing a manuscript detailing the novel and unexpected effects.

#3.6. In Fig. 4C, the plasma membrane (PM) localization of Nir1-LNS2 and NES-PABDx2-Spo20, as determined by the "intensity PM/Cyto," should be analyzed following the ectopic production of PI4P and PIP2. Although mitochondria do not apparently recruit Nir1-LNS2 or NES-PABDx2-Spo20 after induced PI4P and PIP2 production, it remains possible that the subsequent trafficking of PI4P and PIP2 from mitochondria might sequester the biosensors away from the PM into the cytoplasm, thereby reducing the "intensity PM/Cyto" of Nir1-LNS2.

We cannot easily determine the PM/Cyto ratio in this experiment as we included a mitochondrial marker rather than a PM marker when imaging. However, the images shown in Figure 4 are

seemingly comparable to the images shown in **Supplemental Figure 2 (see response to #2.1)**, suggesting that basal PILS-Nir1 localization is not significantly disrupted in this experiment.

#3.7. It would be valuable to determine the half-life (stability) of Nir1-LNS2.

While a definitive half-life value could be useful, in all of our transient transfections, the PILS-Nir1 shows good stability, and we don't expect degradation to be a major concern.

August 3, 2025

RE: JCB Manuscript #202405174R

Gerald Hammond
University of Pittsburgh

Dear Dr. Hammond:

Thank you for submitting your revised manuscript entitled "PILS-Nir1 is a novel phosphatidic acid biosensor that reveals mechanisms of lipid production". The reviewers now all support publication therefore we would be happy to publish your paper in JCB pending final revisions necessary to meet our formatting guidelines (see details below).

A. MANUSCRIPT ORGANIZATION AND FORMATTING:

- 1) Text limits: Character count for Articles is < 40,000, not including spaces. Count includes abstract, introduction, results, discussion, and acknowledgments. Count does not include title page, figure legends, materials and methods, references, tables, or supplemental legends.
- 2) Figures limits: Articles may have up to 10 main text figures.
- 3) Figure formatting: Scale bars must be present on all microscopy images, including inset magnifications. Molecular weight or nucleic acid size markers must be included on all gel electrophoresis. Aspect ratios of images may not be altered. Please consider the comment of reviewer 1 regarding colors.
- 4) Statistical analysis: Error bars on graphic representations of numerical data must be clearly described in the figure legend. The number of independent data points (n) represented in a graph must be indicated in the legend. Statistical methods should be explained in full in the materials and methods. For figures presenting pooled data the statistical measure should be defined in the figure legends. Please also be sure to indicate the statistical tests used in each of your experiments (either in the figure legend itself or in a separate methods section) as well as the parameters of the test (for example, if you ran a t-test, please indicate if it was one- or two-sided, etc.). Also, if you used parametric tests, please indicate if the data distribution was tested for normality (and if so, how). If not, you must state something to the effect that "Data distribution was assumed to be normal but this was not formally tested."
- 5) Abstract and title: The abstract should be no longer than 160 words and should communicate the significance of the paper for a general audience. The title should be less than 100 characters including spaces. Make the title concise but accessible to a general readership.

Please do not include "novel" in your title
- 6) Materials and methods: Should be comprehensive and not simply reference a previous publication for details on how an experiment was performed. Please provide full descriptions in the text for readers who may not have access to referenced manuscripts.
- 7) All antibodies, cell lines, animals, and tools used in the manuscript should be described in full, including accession numbers for materials available in a public repository such as the Resource Identification Portal. Please be sure to provide the sequences for all of your primers/oligos and RNAi constructs in the materials and methods. You must also indicate in the methods the source, species, and catalog numbers (where appropriate) for all of your antibodies. Please also indicate the acquisition and quantification methods for immunoblotting/western blots.
- 8) Microscope image acquisition: The following information must be provided about the acquisition and processing of images:
 - a. Make and model of microscope
 - b. Type, magnification, and numerical aperture of the objective lenses
 - c. Temperature
 - d. Imaging medium
 - e. Fluorochromes
 - f. Camera make and model

g. Acquisition software

h. Any software used for image processing subsequent to data acquisition. Please include details and types of operations involved (e.g., type of deconvolution, 3D reconstitutions, surface or volume rendering, gamma adjustments, etc.).

10) Supplemental materials: There are strict limits on the allowable amount of supplemental data. Articles may have up to 5 supplemental figures. Please also note that tables, like figures, should be provided as individual, editable files. A summary of all supplemental material should appear at the end of the Materials and methods section.

13) ORCID IDs: ORCID IDs are unique identifiers allowing researchers to create a record of their various scholarly contributions in a single place. Please note that ORCID IDs are now *required* for all authors. At resubmission of your final files, please be sure to provide your ORCID ID and those of all co-authors.

Please note that JCB now requires authors to submit Source Data used to generate figures containing gels and Western blots with all revised manuscripts. This Source Data consists of fully uncropped and unprocessed images for each gel/blot displayed in the main and supplemental figures. For assays performed using capillary electrophoresis and/or immunoassay-based detection, authors should instead provide the electropherogram graph(s) for each experiment, plotting fluorescence/chemiluminescence intensity vs. molecular weight/size. Please be sure to provide one Source Data file for each figure gels, blots, and/or capillary electrophoresis assays along with your revised manuscript files. File names for Source Data figures should be alphanumeric without any spaces or special characters (i.e., SourceDataF#, where F# refers to the associated main figure number or SourceDataFS# for those associated with Supplementary figures). For traditional gels and blots, the lanes of the gels/blots should be labeled as they are in the associated figure, the place where cropping was applied should be marked (with a box), and molecular weight/size standards should be labeled wherever possible. For capillary electrophoresis assays, each trace in the graph should be color-coded and labeled to indicate which protein, gene, or sample is being measured (please try to avoid red/green combinations to accommodate our color-blind readers).

Journal of Cell Biology now requires a data availability statement for all research article submissions. These statements will be published in the article directly above the Acknowledgments. The statement should address all data underlying the research presented in the manuscript. Please visit the JCB instructions for authors for guidelines and examples of statements at (<https://rupress.org/jcb/pages/editorial-policies#data-availability-statement>).

B. FINAL FILES:

-- Cover images: If you have any striking images related to this story, we would be happy to consider them for inclusion on the

journal cover. Submitted images may also be chosen for highlighting on the journal table of contents or JCB homepage carousel. Images should be uploaded as TIFF or EPS files and must be at least 300 dpi resolution.

****It is JCB policy that if requested, original data images must be made available to the editors. Failure to provide original images upon request will result in unavoidable delays in publication. Please ensure that you have access to all original data images prior to final submission.****

****The license to publish form must be signed before your manuscript can be sent to production. A link to the electronic license to publish form will be sent to the corresponding author only. Please take a moment to check your funder requirements before choosing the appropriate license.****

Thank you for your attention to these final processing requirements. Please revise and format the manuscript and upload materials within 7 days. If you need an extension for whatever reason, please let us know and we can work with you to determine a suitable revision period.

Thank you for this interesting contribution, we look forward to publishing your paper in Journal of Cell Biology.

Sincerely,

William Prinz, PhD
Monitoring Editor

Andrea L. Marat, PhD
Deputy Editor

Journal of Cell Biology

Reviewer #1 (Comments to the Authors (Required)):

1. The authors designed, created, and thoroughly validated a fluorescently tagged sensor protein that binds specifically and with high affinity to the signaling phospholipid PA in cells. The LNS2 PA-binding domain used originates from the lipid transfer protein Nir1, and shares conserved features with lipins. The novel sensor outperforms the commonly used Spo20-based PA probes. Importantly, the authors elegantly and unambiguously demonstrate that PA but not DAG or PIP2 is required and sufficient for membrane binding of Nir1-LNS2 in cells. The development of a new PA biosensor with superior specificity and sensitivity in vivo is a significant advance in, and will benefit the research areas of lipid signaling and specific lipid-protein interactions.

2. Comments to authors in response to their rebuttal; numbers of points follows the authors' numbering.

1.1 The modified interpretation and description of the results is appropriate.

Earlier, we also noted that the pictures of controls in Fig. 5D have been copied from the series in Fig. 5B and 5C, but that Fig. 5D uses a different color gradient for PILS-NIR1 than Fig. 5C.

[This is undesirable because the different color gradient used for the PILS-NIR1 control in Fig. 5D results in a lower visibility of cytosolic fluorescence compared to the same picture in Fig. 5C. The decrease in cytosolic fluorescence in Fig. 5C after 6 minutes of stimulation appears to be the clearest sign of probe relocalization. However, in the color gradient of Fig. 5C such a decrease would probably hardly be discernable.]

Consistent use of color gradients is recommended.

1.2 The new experimental data and discussion that have been added to the ms in response to our second point is most satisfactory. The experiments show - in vivo - a convincing dependency of the probe's binding to PA on K820, leading to a very low PM association of PILS-NIR1-K820E, also under control conditions. Also, the fact that PIP2 binding in liposomes is not affected by the mutation is very interesting. The new data take away concerns about the biosensor's specificity.

1.3 The new results strongly suggest that the probe's localization is not significantly influenced by PIP and PIP2 depletion at the

PM. This satisfies our critical concerns.

Points 1.4 through 1.16 have been satisfactorily dealt with. (Although, point 1.7 does raise questions about the experiment shown in Figure 2A of the original ms...)

In conclusion, we think the newly developed PA biosensor reported in the ms presents an important new tool to study PA signaling, with potential use in a wide variety of cells and organisms.

Reviewer #2 (Comments to the Authors (Required)):

Weckerly and Rahn et al. reported a new biosensor, PILS-Nir1, that binds to and reports on phosphatidic acid (PA) levels in live cells. The new data presented by the authors clearly demonstrate that PILS-Nir1 achieves superior sensitivity and specificity compared to the existing PA biosensors currently used in the field. The authors sufficiently addressed the concerns I previously raised, and this manuscript should be suitable for publication in JCB.

Reviewer #3 (Comments to the Authors (Required)):

The authors have addressed all my major concerns. I suggest publication of the revised manuscript by JCB.